# PROPER VELOCITY NEURAL NETWORKS

**Ziheng Chen**[*1], **Zihan Su**[*2], **Bernhard Schölkopf**[3] **& Nicu Sebe**[1]
[1] University of Trento, [2] University of Nottingham, [3] MPI-IS

## ABSTRACT

Hyperbolic Neural Networks (HNNs) have shown remarkable success in representing hierarchical and tree-like structures, yet most existing work relies on the Poincaré ball and hyperboloid models. While these models admit closed-form Riemannian operators, their constrained nature potentially leads to numerical instabilities, especially near model boundaries. In this work, we explore the Proper Velocity (PV) space, an unconstrained representation of hyperbolic space rooted in Einstein's special relativity, as a stable alternative. We first establish the complete Riemannian toolkit of the PV space. Building on this foundation, we introduce Proper Velocity Neural Networks (PVNNs) with core layers including Multinomial Logistic Regression (MLR), Fully Connected (FC), convolutional, activation, and batch normalization layers. Extensive experiments across four tasks, namely numerical stability, image classification, graph node classification, and genomic sequence learning, demonstrate the stability and effectiveness of PVNNs. The code is available at `https://github.com/NickyoyoSu/PVNN`.

## 1 INTRODUCTION

Hyperbolic geometry provides a natural representation for hierarchical data due to its exponential representation capacity, and has proven successful across diverse applications, including computer vision (Khrulkov et al., 2020; Bdeir et al., 2024; Sur et al., 2025; Wang et al., 2026), knowledge graphs (Li et al., 2024), natural language processing (Ganea et al., 2018; Shimizu et al., 2021), knowledge-graph reasoning (Nickel & Kiela, 2017), astronomy (Chen et al., 2025a), graph learning (Chami et al., 2019; Li et al., 2024), genomic sequence learning (Khan et al., 2025), fine-tuning (Yang et al., 2025), and brain signal analysis (Li et al., 2026). Recently, the focus has shifted from hyperbolic embeddings to building Hyperbolic Neural Networks (HNNs) that operate entirely within hyperbolic space. As hyperbolic geometry admits multiple models (Cannon et al., 1997), the choice of representation is central to the design of hyperbolic networks. Most recent works rely on the Poincaré ball and hyperboloid models, which provide convenient Riemannian or gyrovector structures (Ganea et al., 2018; Ungar, 2022; Chen et al., 2025c), thereby facilitating neural network construction. However, both models are constrained spaces, which can lead to numerical instabilities. In particular, as embeddings in the Poincaré ball approach the boundary, numerical computations become unstable and might cause gradients to vanish (Guo et al., 2022).

On the other hand, the Proper Velocity (PV) model originates from Einstein's special relativity, where proper velocity provides a natural parameterization for relativistic velocity addition (Ungar, 2022, Ch. 10). Algebraically, PV admits a gyrovector space (Ungar, 2022, Ch. 6), analogous to the Möbius gyrovector space of the Poincaré ball. Unlike the constrained Poincaré ball and hyperboloid models, PV offers an unconstrained representation that alleviates numerical instabilities. These properties have made the PV model successful in relativistic physics and motivate its exploration as a stable alternative geometry for HNNs. However, its Riemannian operators, including exponential and logarithmic maps and parallel transport, remain largely unexplored, despite being fundamental for constructing neural networks.

Inspired by the above discussions, we propose Proper Velocity Neural Networks (PVNNs). To this end, we first establish the complete Riemannian geometry of PV by deriving closed-form expressions for the exponential map, logarithmic map, geodesic distance, and parallel transport. Building on this foundation, we extend several fundamental neural layers into PV space, including Multinomial Logistics Regression (MLR) classification, Fully Connected (FC), convolutional, activation,

---

[*]Equal contribution.

and batch normalization layers. Based on these layers, one can construct different network architectures. We validate the framework through four sets of experiments, including numerical stability, computer vision, graph learning, and genomic sequence learning, demonstrating both the stability of PV embeddings and effectiveness of PVNNs. To our knowledge, the PV model has remained largely unexplored in machine learning, and our work provides the first systematic study of its use for representation learning. In summary, our **contributions** are threefold:

1. We establish the complete Riemannian geometric toolkit of the PV manifold, deriving closed-form operators that enable its use as a new alternative to classical hyperbolic models.
2. We develop fundamental building blocks in PV space, including MLR, FC, convolutional, activation, and batch normalization layers.
3. We validate the stability and effectiveness of PVNNs through experiments on four tasks: numerical stability, image classification, graph node classification, and genomic sequence learning.

## 2 RELATED WORK

**Hyperbolic representation.** Hyperbolic embeddings have been widely explored for hierarchical and non-Euclidean structures in networks, trees, and text (Krioukov et al., 2010; Wilson et al., 2014; Sonthalia & Gilbert, 2020; Nickel & Kiela, 2017; Chami et al., 2019). Hyperbolic neural networks (HNNs) explore these embeddings within deep architectures (Ganea et al., 2018), and subsequent works explore them in different applications, such as graphs, knowledge bases, vision, genome, protein, and brain signals (Chami et al., 2019; Balazevic et al., 2019; Khrulkov et al., 2020; Bachmann et al., 2020; Khan et al., 2025; Li et al., 2026).

**Hyperbolic models and networks.** Among the multiple models of hyperbolic geometry (Cannon et al., 1997), the Poincaré ball and the hyperboloid (Lorentz) models are most commonly adopted. The Poincaré ball admits closed-form Möbius and Riemannian operators (Ganea et al., 2018; Shimizu et al., 2021), whereas the hyperboloid model provides numerically stable geodesics and natural formulations in Minkowski space (Nickel & Kiela, 2018; Chen et al., 2022; Mishne et al., 2023). Building on these operators, researchers have adapted core Euclidean layers to hyperbolic geometries. For instance, Ganea et al. (2018); Shimizu et al. (2021) introduced FC and MLR layers on the Poincaré ball via point-to-hyperplane distances, while Chen et al. (2022); Bdeir et al. (2024) designed FC and convolutional layers on the hyperboloid through ambient spacetime formulations. Recent works further develop residual architectures and graph-specific formulations (Van Spengler et al., 2023; He et al., 2024; Chami et al., 2019; Dai et al., 2021).

**Riemannian normalization.** Although normalization layers are essential for stabilizing and accelerating training (Ioffe & Szegedy, 2015; Ba et al., 2016; Ulyanov et al., 2016; Wu & He, 2018), their Euclidean formulations do not generalize directly to manifolds. Early extensions adapted Riemannian operators such as the exponential map, logarithmic map, and parallel transport to define batch normalization on different manifolds (Brooks et al., 2019; Lou et al., 2020; Chakraborty, 2020; Bdeir et al., 2024; Wang et al., 2025a;b). However, these approaches often lack theoretical guarantees to normalize sample statistics. More recently, algebraic structures, such as Lie groups and gyrogroups, have been explored to establish principled and unified formulations, which can normalize sample statistics (Chakraborty, 2020; Chen et al., 2024a;c; 2025c).

## 3 PRELIMINARIES

**Riemannian geometry (Lee, 2018).** Throughout, $\langle \cdot, \cdot \rangle$ denotes the standard Euclidean inner product, and $\|\cdot\|$ the induced norm. A Riemannian manifold $(\mathcal{M}, g)$ is a smooth manifold equipped with an inner product $g_x$ or $\langle, \rangle_x$ on each tangent space $T_x\mathcal{M}$ that depends smoothly on $x \in \mathcal{M}$. We use $\mathrm{Exp}_x$, $\mathrm{Log}_x$, and $\mathrm{PT}_{x \to y}$ to denote the exponential map at $x$, logarithmic map at $x$, and parallel transport along the geodesic connecting $x$ and $y$, respectively. A smooth map $f : (\mathcal{M}, g) \to (\widetilde{\mathcal{M}}, \widetilde{g})$ is a *Riemannian isometry* if it preserves the metric: $g_x(u, v) = \widetilde{g}_{f(x)}(d_x f(u), d_x f(v))$ with $d_x f$ as the differential map at $x$ and $u, v \in T_x\mathcal{M}$.

**PV space (Ungar, 2022).** Hyperbolic space is a space with constant negative curvature $K < 0$ and admits several models one can work with (Cannon et al., 1997). The popular models include the Poincaré ball and the hyperboloid (also known as the Lorentz model). The PV model $\mathbb{PV}_K^n = \mathbb{R}^n$ is

an alternative representation of hyperbolic geometry, which was initially named the Ungar gyrovector space and is used to describe algebraic structures of relativistic proper velocities (Ungar, 2022). Unlike the bounded Poincaré ball or the constrained hyperboloid, the PV model is an unconstrained space, offering better numerical stability. Its Riemannian metric is given by App. E.1:

$$g_x(u,v) = \langle u,v \rangle + K\beta_x^2 \langle x,u \rangle \langle x,v \rangle, \quad \forall x \in \mathbb{PV}_K^n, \forall u,v \in T_x\mathbb{PV}_K^n. \tag{1}$$

Here, $\beta_x = \frac{1}{\sqrt{1-K\|x\|^2}}$ is the relativistic beta factor. In Ungar's notation, the curvature is parametrized by a positive constant $s$ with $s^2 = -1/K$, where $s$ plays the role of the vacuum speed of light in special relativity (Ungar, 2022, Sec. 3.8).

**PV gyrovector (Ungar, 2022).** From an algebraic point of view, the PV space forms a gyrovector space (Ungar, 2022, Def. 6.2), which extends the Euclidean vector space to manifolds. Given $x,y,z \in \mathbb{PV}_K^n$ and $t \in \mathbb{R}$, PV gyroaddition $\oplus_U$ and scalar gyromultiplication $\otimes_U$ (Ungar, 2022, Ch. 3.11 and 6.20) are defined as[1]

$$x \oplus_U y = x + y + \left\{ \frac{1-\beta_y}{\beta_y} - K\frac{\beta_x}{1+\beta_x} \langle x,y \rangle \right\} x, \tag{2}$$

$$t \otimes_U y = \sinh\left( t\sinh^{-1}\left(\sqrt{-K}\|y\|\right) \right) \frac{y}{\sqrt{-K}\|y\|}, \quad (t \otimes_U \mathbf{0} = \mathbf{0}). \tag{3}$$

In particular, the PV inverse is $\ominus_U x = -x$, and the PV identity is the zero vector: $\mathbf{0} \oplus_U x = x \oplus_U \mathbf{0} = x$.

For detailed reviews of Riemannian geometry, gyrovector spaces, PV gyrovector spaces, and the hyperbolic Poincaré ball and hyperboloid models, we refer the reader to App. B.

## 4 PV GEOMETRY

### 4.1 FROM GYRO ISOMORPHISM TO RIEMANNIAN ISOMETRY

The Poincaré ball also admits a gyrovector space, named the Möbius gyrovector space, as reviewed in (Ungar, 2022, Sec. 6.14). Algebraically, the PV and Möbius gyrovector spaces are isomorphic. We further show that PV and the Poincaré ball are geometrically isometric.

Let $\mathbb{P}_K^n = \left\{ x \in \mathbb{R}^n \,|\, \|x\|^2 < -1/K \right\}$ be the Poincaré ball. The following bijections define the gyrovector space isomorphism (Ungar, 2022, Tab. 6.1):

$$\pi_{\mathbb{PV}_K^n \to \mathbb{P}_K^n} : \mathbb{PV}_K^n \ni x \mapsto \frac{\beta_x}{1+\beta_x} x \in \mathbb{P}_K^n, \quad \pi_{\mathbb{P}_K^n \to \mathbb{PV}_K^n} : \mathbb{P}_K^n \ni y \mapsto 2\gamma_y^2 y \in \mathbb{PV}_K^n, \tag{4}$$

where $\gamma_y = \frac{1}{\sqrt{1+K\|y\|^2}}$ is the gamma factor. The isomorphism preserves the gyro operations:

$$\pi_{\mathbb{PV}_K^n \to \mathbb{P}_K^n}(x \oplus_U y) = \pi_{\mathbb{PV}_K^n \to \mathbb{P}_K^n}(x) \oplus_M \pi_{\mathbb{PV}_K^n \to \mathbb{P}_K^n}(y), \quad \forall x,y \in \mathbb{PV}_K^n, \tag{5}$$

$$\pi_{\mathbb{PV}_K^n \to \mathbb{P}_K^n}(r \otimes_U x) = r \otimes_M \pi_{\mathbb{PV}_K^n \to \mathbb{P}_K^n}(x), \quad \forall x \in \mathbb{PV}_K^n, \forall r \in \mathbb{R}, \tag{6}$$

where $\otimes_M$ and $\oplus_M$ are the Möbius gyro operations which are reviewed in App. B.4.

**Lemma 4.1** (Differentials). [↓] *The differentials of $\pi_{\mathbb{PV}_K^n \to \mathbb{P}_K^n}$ and $\pi_{\mathbb{P}_K^n \to \mathbb{PV}_K^n}$ are*

$$d_x(\pi_{\mathbb{PV}_K^n \to \mathbb{P}_K^n})(v) = K\frac{\beta_x^3}{(1+\beta_x)^2} \langle x,v \rangle x + \frac{\beta_x}{1+\beta_x} v, \quad \forall x \in \mathbb{PV}_K^n, \forall v \in T_x\mathbb{PV}_K^n, \tag{7}$$

$$d_y(\pi_{\mathbb{P}_K^n \to \mathbb{PV}_K^n})(w) = -4K\gamma_y^4 \langle y,w \rangle y + 2\gamma_y^2 w, \quad \forall y \in \mathbb{P}_K^n, \forall w \in T_y\mathbb{P}_K^n. \tag{8}$$

*Let $\mathbb{I}$ be the identity map. The differentials at the origin $\mathbf{0}$ are*

$$d_{\mathbf{0}}(\pi_{\mathbb{PV}_K^n \to \mathbb{P}_K^n}) = \tfrac{1}{2}\mathbb{I}, \quad d_{\mathbf{0}}(\pi_{\mathbb{P}_K^n \to \mathbb{PV}_K^n}) = 2\mathbb{I}. \tag{9}$$

Based on Lem. 4.1, we can prove that the above isomorphisms are isometries.

**Theorem 4.2** (Isometries). [↓] *The mappings in Eq. (4) are Riemannian isometries.*

---

[1]The subscript U refers to the initial of Ungar.

## 4.2 PV RIEMANNIAN OPERATORS

The Poincaré ball admits closed-form Riemannian operators (Ganea et al., 2018). By Thm. 4.2, we can readily obtain the counterparts on PV space via properties of isometries (Chen et al., 2026, App. C.2).

**Theorem 4.3** (PV Riemannian operators). [↓] *Let $\pi = \pi_{\mathbb{PV}_K^n \to \mathbb{P}_K^n}$. Given $x, y \in \mathbb{PV}_K^n$ and $v \in T_x\mathbb{PV}_K^n$, the Riemannian operators on the PV space are*

$$\mathrm{Exp}_x(v) = x \oplus_\mathrm{U} \left( \frac{1}{\sqrt{-K}} \sinh\left( \frac{\sqrt{-K}(1+\beta_x)}{\beta_x} \|d\pi_x(v)\| \right) \frac{d\pi_x(v)}{\|d\pi_x(v)\|} \right), \quad (10)$$

$$\mathrm{Log}_x(y) = \sigma(x,y)z + \tau(x,y)\langle x,z \rangle x, \quad (11)$$

$$\mathrm{PT}_{x \to y}(v) = \frac{1+\beta_x}{\beta_x}\tilde{v} - K\frac{(1+\beta_x)\beta_y}{(1+\beta_y)\beta_x}\langle y,\tilde{v} \rangle y, \quad (12)$$

$$\mathrm{d}(x,y) = \frac{2}{\sqrt{-K}}\tanh^{-1}\left( \sqrt{-K}\|\pi(-x \oplus_\mathrm{U} y)\| \right), \quad (13)$$

*with $z = (-x) \oplus_\mathrm{U} y$. For the parallel transport, $\tilde{v} = \mathrm{gyr}_\mathrm{M}[\bar{y}, -\bar{x}]\left( d\pi_x(v) \right)$ with $\mathrm{gyr}_\mathrm{M}$ as the Möbius gyration in App. B.4, $\bar{x} = \frac{\beta_x}{1+\beta_x}x$ and $\bar{y} = \frac{\beta_y}{1+\beta_y}y$. Here, the scalar coefficients in the logarithm are*

$$\sigma(x,y) = \frac{2}{\sqrt{-K}}\frac{\tanh^{-1}\left( \sqrt{-K}\|\pi(z)\| \right)}{\|z\|}, \tau(x,y) = \frac{2\beta_x}{1+\beta_x}\frac{\sqrt{-K}\tanh^{-1}\left( \sqrt{-K}\|\pi(z)\| \right)}{\|z\|}.$$

*At the identity $\mathbf{0}$, the above operators can be further simplified:*

$$\mathrm{Exp}_\mathbf{0}(v) = \frac{1}{\sqrt{-K}}\sinh\left( \sqrt{-K}\|v\| \right)\frac{v}{\|v\|}, \qquad \mathrm{Log}_\mathbf{0}(y) = \frac{1}{\sqrt{-K}}\sinh^{-1}\left( \sqrt{-K}\|y\| \right)\frac{y}{\|y\|},$$

$$\mathrm{PT}_{\mathbf{0} \to y}(v) = v - K\frac{\beta_y}{1+\beta_y}\langle y,v \rangle y, \qquad \mathrm{PT}_{x \to \mathbf{0}}(v) = v + K\frac{\beta_x^2}{1+\beta_x}\langle x,v \rangle x,$$

$$\mathrm{d}(\mathbf{0},y) = \frac{1}{\sqrt{-K}}\sinh^{-1}\left( \sqrt{-K}\|y\| \right).$$

The above facts imply that the PV gyro operations can be expressed via Riemannian operations.

**Theorem 4.4** (Gyro by Riemannian). [↓] *The PV gyro operations can be rewritten as*

$$x \oplus_\mathrm{U} y = \mathrm{Exp}_x\left( \mathrm{PT}_{\mathbf{0} \to x}(\mathrm{Log}_\mathbf{0}(y)) \right), \quad t \otimes_\mathrm{U} x = \mathrm{Exp}_\mathbf{0}(t\,\mathrm{Log}_\mathbf{0}(x)) \quad \forall x,y \in \mathbb{PV}_K^n, \forall t \in \mathbb{R}.$$

## 5 PV NEURAL NETWORKS

Building on the above gyrovector and Riemannian tools, we introduce fundamental building blocks for PV neural networks, including Multinomial Logistics Regression (MLR), Fully Connected (FC), convolutional, activation, and batch normalization layers, thereby enabling the construction of concrete deep architectures in this space.

## 5.1 PV MULTINOMIAL LOGISTIC REGRESSION

The Euclidean MLR $\mathrm{Softmax}(Ax+b)$ is a standard classification layer in Euclidean deep learning. As shown by Lebanon & Lafferty (2004); Ganea et al. (2018); Chen et al. (2024b), each output of a $C$-class MLR can be reformulated as the signed margin distance to a hyperplane:

$$p(y = k \mid x) \propto \exp\left( v_k(x) \right), \quad v_k(x) = \mathrm{sign}\left( \langle a_k, x - p_k \rangle \right)\|a_k\|\mathrm{d}\left( x, H_{a_k,p_k} \right), \quad 1 \le k \le C, \quad (14)$$

where $a_k, p_k \in \mathbb{R}^n$ and $H_{a_k,p_k} = \{x \in \mathbb{R}^n \mid \langle a_k, x - p_k \rangle = 0\}$.

Following the Poincaré MLR (Ganea et al., 2018, Sec. 3.1), we define the PV hyperplane as

$$H_{a,p} = \left\{ x \in \mathbb{PV}_K^n \mid \left\langle \mathrm{Log}_p(x), a \right\rangle_p = 0 \right\}, \qquad p \in \mathbb{PV}_K^n, a \in T_p\mathbb{PV}_K^n. \quad (15)$$

where $p \in \mathbb{PV}_K^n$ and $a \in T_p\mathbb{PV}_K^n$ are the hyperplane parameters. As the Poincaré hyperplane can be expressed by the Möbius gyro operations (Ganea et al., 2018, Eq. 22), the PV hyperplane can also be expressed by the PV gyro operations. In addition, building PV MLR requires the PV point-to-hyperplane distance. The following theorem provides these results.

**Theorem 5.1.** [↓] *Let $\pi = \pi_{\mathbb{PV}_K^n \to \mathbb{P}_K^n}$. Given $x, p \in \mathbb{PV}_K^n$ and $a \in T_p\mathbb{PV}_K^n$, we have*

$$H_{a,p} = \left\{ x \in \mathbb{PV}_K^n \mid \left\langle \mathrm{Log}_p(x), a \right\rangle_p = 0 \right\} = \left\{ x \in \mathbb{PV}_K^n \mid \left\langle -p \oplus_U x, d_p\pi(a) \right\rangle = 0 \right\}, \quad (16)$$

$$d(y, H_{a,p}) = \inf_{w \in H_{a,p}} d(y, w) = \frac{1}{\sqrt{-K}} \sinh^{-1} \left( \frac{\sqrt{-K} \left| \left\langle -p \oplus_U y, d_p\pi(a) \right\rangle \right|}{\|d_p\pi(a)\|} \right). \quad (17)$$

By Thm. 5.1, we define the $C$-class PV MLR as

$$p(y = k \mid x) \propto \exp\left(v_k(x)\right), \quad v_k(x) = \mathrm{sign}\left(\left\langle -p_k \oplus_U x, d_{p_k}\pi(a_k) \right\rangle\right) \|a_k\|_{p_k} d\left(x, H_{a_k, p_k}\right), \quad (18)$$

where $p_k \in \mathbb{PV}_K^n$ and $a_k \in T_{p_k}\mathbb{PV}_K^n$ are the PV MLR parameters for class $k$. However, the above expression has three drawbacks: (i) the parameter $p_k$ is over-parameterized, as it corresponds to the scalar bias parameter in the Euclidean MLR; (ii) the gyroaddition in $\left\langle -p_k \oplus_U x, d_{p_k}\pi(a_k) \right\rangle$ complicates the computation; and (iii) the parameters $(p_k, a_k)$ are constrained, making optimization costly. To address these drawbacks, we follow Shimizu et al. (2021) and adopt the parameterization $p_k = \mathrm{Exp}_\mathbf{0}(r_k z_k / \|z_k\|), a_k = \mathrm{PT}_{\mathbf{0} \to p_k}(z_k)$ with $z_k \in T_\mathbf{0}\mathbb{PV}_K^n \cong \mathbb{R}^n$ and $r_k \in \mathbb{R}$. This parameterization avoids Riemannian optimization in PV MLR and further simplifies the formulation.

**Theorem 5.2** (PV MLR). [↓] *For $x \in \mathbb{PV}_K^n$, the score $v_k(x)$ in Eq. (18) for each class $k$ is*

$$v_k(x) = \frac{\|z_k\|}{\sqrt{-K}} \sinh^{-1} \left( \cosh(\sqrt{-K}r_k) \frac{\sqrt{-K}}{\|z_k\|} \langle x, z_k \rangle - \sinh(\sqrt{-K}r_k)\sqrt{1 - K\|x\|^2} \right), \quad (19)$$

*where $z_k \in \mathbb{R}^n$ and $r_k \in \mathbb{R}$ are parameters for class $k$. In particular, as $K \to 0^-$ we have $v_k(x) \to \langle x, z_k \rangle + b_k$ with $b_k = -r_k\|z_k\|$, which recovers the Euclidean MLR in Eq. (14).*

The parameterization $(z_k, r_k)$ is essential for efficiency. In the original form Eq. (18), computing $v_k(x)$ for a batch $x \in \mathbb{R}^{b \times n}$ and $C$ classes requires explicit gyroaddition $-p_k \oplus_U x$ for each class, producing an intermediate tensor of size $b \times C \times n$ that could cause out-of-memory errors in high dimensions. One could instead loop over classes, but this is computationally inefficient. In contrast, Eq. (19) depends on inner products $\langle x, z_k \rangle$, which can be implemented as a matrix multiplication.

## 5.2 PV FULLY CONNECTED LAYER

The Euclidean FC layer is defined as $y = Ax + b$ with $A \in \mathbb{R}^{m \times n}$ and $b \in \mathbb{R}^m$. It can be expressed element-wise as $y_k = \langle a_k, x \rangle - b_k = \langle a_k, x - p_k \rangle$ with $a_k, p_k \in \mathbb{R}^n$ and $\langle p_k, a_k \rangle = b_k$. As shown by Shimizu et al. (2021, Sec. 3.2), the LHS $y_k$ is the signed distance from $y$ to the hyperplane passing through the origin and orthogonal to the $k$-th axis of the output space, which can be formulated as

$$\mathrm{sign}\left(\langle e_k, y - \mathbf{0} \rangle\right) d(y, H_{e_k, \mathbf{0}}) = \langle a_k, x - p_k \rangle, \quad \forall 1 \le k \le m, \quad (20)$$

where $e_k$ denotes the vector whose $k$-th element is 1 and all others are 0.

For the PV model, the LHS of Eq. (20) can be formulated by the signed point-to-hyperplane distance, while the RHS can be formulated by the $v_k$ in PV MLR. Specifically, the PV FC layer $\mathcal{F} : \mathbb{PV}_K^n \to \mathbb{PV}_K^m$ from the $n$-dimensional to the $m$-dimensional PV spaces for the input $x \in \mathbb{PV}_K^n$ returns the output $y \in \mathbb{PV}_K^m$ by solving the $m$ equations:

$$\mathrm{sign}\left(\langle d_{\mathbf{0}_k}\pi(e_k), -\mathbf{0} \oplus_U x \rangle\right) d(y, H_{e_k, \mathbf{0}}) = v_k(x), \quad \forall 1 \le k \le m, \quad (21)$$

where $H_{e_k, \mathbf{0}}$ and $v_k(x)$ are given by Thm. 5.1. This definition has an explicit solution.

**Theorem 5.3** (PV FC layer). [↓] *The output $y = \mathcal{F}(x) \in \mathbb{PV}_K^m$ has the closed form*

$$y_k = \frac{1}{\sqrt{-K}} \sinh(\sqrt{-K}v_k(x)), \quad 1 \le k \le m, \quad (22)$$

*where $v_k(x)$ is defined as Eq. (19) with $z_k \in \mathbb{R}^n$ and $r_k \in \mathbb{R}$ as the FC parameters. In particular, as $K \to 0^-$ we have $y_k \to \langle x, z_k \rangle + b_k$ with $b_k = -r_k\|z_k\|$, which recovers the Euclidean FC layer.*

**Generalization.** We can jointly express the Euclidean FC layer and activation $\sigma$, which yields the RHS of Eq. (20) with $\sigma\left(\langle a_k, x - p_k \rangle\right)$. Accordingly, we extend the PV FC by applying the activation to $v_k(x)$ in Eq. (21). Then, Eq. (22) becomes

$$y_k = \frac{1}{\sqrt{-K}} \sinh(\sqrt{-K}\sigma(v_k(x))), \quad 1 \le k \le m. \tag{23}$$

## 5.3 PV CONVOLUTION AND ACTIVATION

**Convolution.** As shown by Shimizu et al. (2021); Bdeir et al. (2024), Euclidean convolution consists of linear maps between kernel weights and concatenated values in each receptive field. To define convolution on PV space, it therefore suffices to define PV concatenation, since we already have the PV FC layer. Because PV space is unconstrained, we define PV concatenation to coincide with Euclidean concatenation. For simplicity, we consider the 1D case. For PV inputs $\{x_i \in \mathbb{PV}_K^n\}_{i=1}^k$ in a 1D receptive field (where $k$ is the kernel size), the PV convolution output $y \in \mathbb{PV}_K^m$ for this receptive field is $y = \mathcal{F}\left(\text{Concat}(x_i, \ldots, x_k)\right)$, where $\text{Concat}(\cdot)$ is standard Euclidean concatenation and $\mathcal{F}$ is the PV FC layer.

**Activation.** A natural choice is to apply a Euclidean activation $\sigma$ in the tangent space at the origin via the mapping $x \mapsto \text{Exp}_{\mathbf{0}}\left(\sigma\left(\text{Log}_{\mathbf{0}}(x)\right)\right)$, which has been shown to be effective in Poincaré networks (Ganea et al., 2018). Alternatively, since PV space is unconstrained, we can apply the activation directly in PV space as $x \mapsto \sigma(x)$. This direct PV-space activation avoids exponential and logarithmic maps and is therefore more efficient.

## 5.4 PV NORMALIZATION

Recently, Chen et al. (2024c; 2025c) extended Batch Normalization (BN) to non-Euclidean manifolds through gyro-structures, referred to as GyroBN. Intuitively, subtraction, addition, and scaling in Euclidean BN are replaced by gyrosubtraction, gyroaddition, and gyromultiplication, respectively. We extend their framework to PV space and show that PV GyroBN can normalize sample statistics.

We first recall the Fréchet statistics. Given $N$ samples $\{x_i\}_{i=1}^N \subset \mathbb{PV}_K^n$, the Fréchet mean and Fréchet variance are

$$\mu = \text{FM}(\{x_i\}_{i=1}^N) = \text{argmin}_{y \in \mathbb{PV}_K^n} \frac{1}{N} \sum\nolimits_{i=1}^N \mathrm{d}^2(x_i, y), \quad v^2 = \frac{1}{N} \sum\nolimits_{i=1}^N \mathrm{d}^2(x_i, \mu). \tag{24}$$

Given activations $\{x_i \in \mathbb{PV}_K^n\}_{i=1}^N$, the core operations of PV GyroBN are

$$\forall i \le N, \quad \tilde{x}_i \leftarrow \overbrace{\beta \oplus_{\text{U}}}^{\text{Biasing}} \left( \overbrace{\frac{s}{\sqrt{v^2 + \epsilon}} \otimes_{\text{U}}}^{\text{Scaling}} \left( \overbrace{-\mu \oplus_{\text{U}} x_i}^{\text{Centering}} \right) \right), \tag{25}$$

where $\mu$ and $v^2$ denote the Fréchet mean and variance, and $\beta \in \mathbb{PV}_K^n$ and $s \in \mathbb{R}$ are parameters. Owing to the isometry between the PV space and Poincaré ball, the PV Fréchet mean can be computed via the Poincaré ball: map the data to the Poincaré ball, compute the Poincaré mean (Lou et al., 2020, Alg. 1), and map the result back.

The following theorem guarantees that PV GyroBN can normalize sample statistics.

**Theorem 5.4** (Homogeneity). [↓] *For $N$ samples $\{x_i\}_{i=1}^N \subset \mathbb{PV}_K^n$, we have*

*Homogeneity of mean:* $\text{FM}\left(\{\beta \oplus_{\text{U}} x_i\}_{i=1}^N\right) = \beta \oplus_{\text{U}} \text{FM}\left(\{x_i\}_{i=1}^N\right), \quad \forall \beta \in \mathbb{PV}_K^n, \tag{26}$

*Homogeneity of dispersion from* $\mathbf{0}$: $\frac{1}{N} \sum\nolimits_{i=1}^N \mathrm{d}^2(t \otimes_{\text{U}} x_i, \mathbf{0}) = t^2 \cdot \frac{1}{N} \sum\nolimits_{i=1}^N \mathrm{d}^2(x_i, \mathbf{0}). \tag{27}$

Thm. 5.4 directly explains the PV GyroBN in Eq. (25). After the centering, the batch mean is shifted to the identity $\mathbf{0}$. After the biasing, it is translated to $\beta$. After the scaling, the variance becomes $s^2$.

## 6 EXPERIMENTS

We evaluate PV embeddings and PV Neural Networks (PVNN) on four representative tasks:

- Sec. 6.1 evaluates the numerical advantage of the PV model against Poincaré and hyperboloid.
- Sec. 6.2 compares PV, Poincaré, and hyperboloid MLRs on image classification.
- Sec. 6.3 evaluates our PV MLR, FC, and GyroBN layers on graph learning.
- Sec. 6.4 compares fully PV convolutional networks with fully hyperboloid convolutional networks on genomic sequence learning.

### 6.1 NUMERICAL STABILITY

We study three aspects: gyro operator, Riemannian operator, and gradient behavior. All experiments use curvature $K = -1$, dimension $n = 16$, and batch size $4096$.

**Gyro operator.** We use scalar gyromultiplication $r \otimes_{\mathcal{H}} x$ as a probe of numerical stability across hyperbolic models. Given random batches $x$ and radii $r$, we evaluate two metrics. The *failure rate* is the fraction of outputs that contain NaN/Inf. The *violation rate* is defined only for models with manifold constraints: Poincaré ball requires $\|x\|^2 < -1/K$, and hyperboloid requires $x_t^2 - \|x_s\| = \frac{1}{K}$ for $x = [x_t, x_s^\top]^\top$. The tolerance is set to $10^{-8}$.

Table 1: Failure and violation probabilities (%) of $r \otimes_{\mathcal{H}} x$ in FP32.

| $r$ | Failure rate | | | Violation rate | | |
|---|---|---|---|---|---|---|
| | $\mathbb{PV}_K^n$ | $\mathbb{P}_K^n$ | $\mathbb{H}_K^n$ | $\mathbb{PV}_K^n$ | $\mathbb{P}_K^n$ | $\mathbb{H}_K^n$ |
| 1 | 0 | 0 | 0 | N/A | 0 | 32.50 |
| 5 | 0 | 0 | 0 | N/A | 0 | 92.36 |
| 10 | 0 | 0 | 0 | N/A | 0 | 99.76 |
| 20 | 0 | 0 | 4.23 | N/A | 0 | 100 |
| 50 | 0 | 0 | 64.42 | N/A | 0 | 100 |
| 75 | 0 | 0 | 79.63 | N/A | 0 | 100 |
| 100 | 0 | 0 | 88.26 | N/A | 0 | 100 |
| 150 | 0 | 0 | 96.43 | N/A | 0 | 100 |
| 200 | 0 | 0 | 100 | N/A | 0 | 100 |
| 1000 | 0 | 0 | 100 | N/A | 0 | 100 |

As PV is unconstrained, its violation rate is reported as N/A. As shown in Tab. 1, PV maintains zero failures up to $r = 1000$ in FP32. The Poincaré ball has zero failure and violation rates, whereas the hyperboloid model starts to fail around $r = 20$ and quickly accumulates both NaN/Inf outputs and off-manifold points under large scalar multipliers, revealing pronounced numerical instability.

**Riemannian operator.** We evaluate the exponential and logarithmic maps by measuring the round-trip error $\|\text{Log}_{\mathbf{0}}(\text{Exp}_{\mathbf{0}}(v)) - v\|$ for tangent vectors $v$ with large norm $\|v\| = 10$. Since this quantity is theoretically zero, any non-zero value reflects numerical instability. We sample a batch of such vectors and report the average error in Tab. 2. PV achieves stable behavior in both FP32 and FP64, whereas the Poincaré ball already exhibits noticeable errors in FP32 and the hyperboloid model remains unstable in both precisions.

Table 2: $\|\text{Log}_{\mathbf{0}}(\text{Exp}_{\mathbf{0}}(v)) - v\|$.

| Model | FP32 | FP64 |
|---|---|---|
| $\mathbb{P}_K^n$ | $2.1 \times 10^{-4}$ | $4.3 \times 10^{-11}$ |
| $\mathbb{H}_K^n$ | $1.0 \times 10^0$ | $1.0 \times 10^0$ |
| $\mathbb{PV}_K^n$ | $2.1 \times 10^{-7}$ | $6.7 \times 10^{-16}$ |

**Gradient.** To compare gradient behavior, we study the gradient of $f_r(x) = \|r \otimes_{\mathcal{H}} x - x\|$ with respect to $x$. Specifically, we sample 24 logarithmically spaced radii $r \in [1, 1000]$ and, for each radius, measure the $\|\nabla_x f_r(x)\|$ on a random batch. The

Table 3: Gradient magnitude $\|\nabla_x\|$ across varying radii.

| Model | $\|\nabla_x\|$ Range | Gradient behavior |
|---|---|---|
| $\mathbb{P}_K^n$ | $[1.1 \times 10^{-11}, 7.6 \times 10^{-13}]$ | Vanishing gradients |
| $\mathbb{H}_K^n$ | $[0, \text{NaN}]$ | Exploding gradients |
| $\mathbb{PV}_K^n$ | $[1.1 \times 10^{-4}, 2.1 \times 10^{-6}]$ | Stable gradients |

range of $\|\nabla_x f_r(x)\|$ is summarized in Tab. 3. The Poincaré ball exhibits severe gradient vanishing near the boundary. In contrast, the hyperboloid model yields gradients that vary from 0 to NaN, reflecting gradient explosion. PV maintains gradients in a safer band.

### 6.2 IMAGE CLASSIFICATION

We compare our PV MLR against previous Poincaré MLRs (Ganea et al., 2018; Shimizu et al., 2021) and Lorentz MLR (Bdeir et al., 2024). Following (Bdeir et al., 2024), we train a ResNet-18 backbone (He et al., 2016) on CIFAR-10 and CIFAR-100 (Krizhevsky & Hinton, 2009), replacing the final Euclidean MLR with a hyperbolic MLR. The backbone output is lifted to the target geometry via the exponential map at the identity. Since PV space is unconstrained, we also consider a

Table 4: Accuracies of hyperbolic MLRs on ResNet-18. Best results are in **bold**. $\delta$ represents the $\delta$-hyperbolicity (lower is more hyperbolic), which comes from Bdeir et al. (2024, Tab. 1).

| Model | Method | CIFAR-10 ($\delta = 0.26$) | CIFAR-100 ($\delta = 0.23$) |
|---|---|---|---|
| $\mathbb{P}_K^n$ | Poincaré MLR (Ganea et al., 2018, Eq. 25) | $95.09 \pm 1.51$ | $76.78 \pm 0.67$ |
| | Unidirectional MLR (Shimizu et al., 2021, Eq. 6) | $95.12 \pm 0.20$ | $77.19 \pm 0.10$ |
| $\mathbb{H}_K^n$ | Lorentz MLR (Bdeir et al., 2024, Thm. 2) | $95.02 \pm 0.12$ | $77.96 \pm 0.09$ |
| $\mathbb{PV}_K^n$ | PV MLR (with $\mathrm{Exp}_\mathbf{0}$) | $95.27 \pm 0.12$ | $78.19 \pm 0.59$ |
| | PV MLR (without $\mathrm{Exp}_\mathbf{0}$) | $\mathbf{95.30 \pm 0.18}$ | $\mathbf{78.20 \pm 0.37}$ |

Table 5: Accuracies of hyperbolic networks on graph learning. Best results are in **bold**. $\delta$ represents the $\delta$-hyperbolicity (lower is more hyperbolic).

| Model | Method | Disease ($\delta = 0$) | Airport ($\delta = 1$) | PubMed ($\delta = 3.5$) | Cora ($\delta = 11$) |
|---|---|---|---|---|---|
| $\mathbb{K}_K^n$ | KNN (Mao et al., 2024) | $79.41 \pm 0.55$ | $92.10 \pm 0.97$ | $69.36 \pm 0.76$ | $52.26 \pm 1.99$ |
| $\mathbb{P}_K^n$ | HNN (Ganea et al., 2018) | $79.90 \pm 0.01$ | $82.16 \pm 2.95$ | $69.28 \pm 0.85$ | $49.68 \pm 1.25$ |
| | HNN++ (Shimizu et al., 2021) | $80.57 \pm 0.23$ | $88.40 \pm 0.17$ | $73.68 \pm 0.39$ | $52.06 \pm 0.90$ |
| $\mathbb{H}_K^n$ | LNN (Bdeir et al., 2024) | $79.90 \pm 0.01$ | $75.20 \pm 1.08$ | $68.82 \pm 0.88$ | $\mathbf{53.34 \pm 1.65}$ |
| $\mathbb{PV}_K^n$ | PVNN | $\mathbf{81.15 \pm 0.23}$ | $\mathbf{97.96 \pm 0.42}$ | $\mathbf{74.33 \pm 0.22}$ | $51.42 \pm 1.33$ |

direct variant that skips $\mathrm{Exp}_\mathbf{0}$ and treats the backbone output as PV coordinates. We denote these two PV heads as PV MLR (with $\mathrm{Exp}_\mathbf{0}$) and PV MLR (without $\mathrm{Exp}_\mathbf{0}$). More details are provided in App. C.2. Tab. 4 reports the 5-fold results. PV MLR matches or outperforms prior hyperbolic baselines, with the largest gains on CIFAR-100 where the decision boundaries are more complex. Both PV variants, with and without $\mathrm{Exp}_\mathbf{0}$, achieve similar accuracies.

## 6.3 GRAPH LEARNING

**Data and Setup.** We study node classification on four standard graph datasets: Disease (Anderson & May, 1991), Airport (Zhang & Chen, 2018), Cora (Sen et al., 2008), and PubMed (Namata et al., 2012). All models share the same architecture consisting of two FC layers with nonlinear activations followed by an MLR classifier; they differ only in the underlying hyperbolic model. Baselines include KNN (Mao et al., 2024) for the Klein ball, HNN/HNN++ (Chami et al., 2019; Shimizu et al., 2021) for the Poincaré ball, and LNN (Bdeir et al., 2024) for the hyperboloid model. Our PVNN is built by PV FC, activation and MLR layers. More details are provided in App. C.3.

**Main results.** For a fair comparison, we use a tangent activation in each model and set $\sigma = \mathbb{I}$ for the PV FC layer in Eq. (23). Tab. 5 summarizes the 5-fold results. On the three more hyperbolic datasets (Disease, Airport, and PubMed), PVNN consistently achieves the best performance, with especially large gains on Airport where it improves over the strongest baseline by $5.86\%$. On the weakly hyperbolic Cora dataset, PVNN remains comparable to Poincaré- and Klein-based networks, and worse than the hyperboloid-based one. Overall, these results suggest that PV geometry is more effective on strongly hyperbolic graphs.

**Tangent vs. Riemannian.** A natural construction of hyperbolic layers is to work in the tangent space. To validate the benefits of our Riemannian PV layers, we compare our PV FC with a tangent-space FC of the form $\mathrm{Exp}_\mathbf{0}(A \mathrm{Log}_\mathbf{0}(x) + b)$, and our GyroBN with a tangent BN (TBN) given by $\mathrm{Log}_\mathbf{0}(\mathrm{BN}(\mathrm{Exp}_\mathbf{0}(x)))$, where BN denotes standard Euclidean batch normalization (Ioffe & Szegedy, 2015). We denote these variants by PVNN+TFC and PVNN+TBN, respectively. As shown in Tab. 6, PVNN consistently outperforms PVNN+TFC on the more hyperbolic Disease and Airport datasets, while the other two are comparable and TFC can be slightly better. For normalization, PVNN+GyroBN improves over PVNN+TBN on all datasets. Overall, these ablations validate the effectiveness of our Riemannian PV constructions, especially in strongly hyperbolic settings.

**Ablations on batch statistics.** PV GyroBN in Eq. (25) uses Fréchet mean and variance, which requires iterative solvers. We also consider two efficient variants. A tangent variant computes batch

Table 6: Results of Tangent FC (TFC) vs PV FC, and Tangent BN (TBN) vs GyroBN.

| Method | Disease | Airport | PubMed | Cora |
|---|---|---|---|---|
| PVNN+TFC | $80.86 \pm 0.30$ | $86.99 \pm 0.61$ | $\mathbf{74.40 \pm 0.43}$ | $\mathbf{53.58 \pm 0.81}$ |
| PVNN | $\mathbf{81.24 \pm 0.36}$ | $\mathbf{97.93 \pm 0.29}$ | $74.16 \pm 0.32$ | $52.26 \pm 1.32$ |
| PVNN+TBN | $80.67 \pm 0.38$ | $98.71 \pm 0.36$ | $73.52 \pm 0.12$ | $45.36 \pm 2.44$ |
| PVNN+GyroBN | $\mathbf{81.24 \pm 0.19}$ | $\mathbf{99.03 \pm 0.18}$ | $\mathbf{74.34 \pm 0.31}$ | $\mathbf{46.64 \pm 5.45}$ |

Table 7: Comparison of methods in calculating mean and variance in PV GyroBN. Time is measured in milliseconds per training epoch.

| Method | Disease | | Airport | | PubMed | | Cora | |
|---|---|---|---|---|---|---|---|---|
| | Acc | Fit Time | Acc | Fit Time | Acc | Fit Time | Acc | Fit Time |
| Tangent | $81.15 \pm 0.23$ | 26.08 | $98.56 \pm 0.36$ | 55.48 | $61.50 \pm 5.75$ | 3.10 | $33.10 \pm 1.58$ | 7.12 |
| Euclidean | $81.15 \pm 0.23$ | 25.80 | $98.75 \pm 0.31$ | 55.19 | $69.82 \pm 3.58$ | 2.99 | $32.62 \pm 0.65$ | 7.29 |
| Fréchet 1 iter | $81.05 \pm 0.23$ | 29.79 | $88.93 \pm 1.17$ | 65.19 | $62.52 \pm 8.44$ | 3.38 | $42.84 \pm 6.15$ | 7.67 |
| Fréchet 2 iters | $81.05 \pm 0.23$ | 30.12 | $94.11 \pm 0.46$ | 67.37 | $73.78 \pm 0.20$ | 3.49 | $45.68 \pm 4.36$ | 8.21 |
| Fréchet 5 iters | $81.24 \pm 0.36$ | 30.90 | $98.50 \pm 0.16$ | 82.28 | $73.92 \pm 0.44$ | 4.02 | $\mathbf{49.50 \pm 1.83}$ | 9.15 |
| Fréchet 10 iters | $\mathbf{81.24 \pm 0.19}$ | 30.49 | $\mathbf{99.03 \pm 0.18}$ | 105.79 | $\mathbf{74.34 \pm 0.31}$ | 3.96 | $46.64 \pm 5.45$ | 9.77 |
| Fréchet $\infty$ | $80.86 \pm 0.00$ | 31.29 | $98.46 \pm 0.15$ | 122.37 | $71.16 \pm 3.93$ | 4.46 | $47.32 \pm 4.73$ | 9.27 |

Table 8: Ablations on PVNN with or without exponential map for the input PV feature.

| $\mathrm{Exp_0}$ | Disease | Airport | PubMed | Cora |
|---|---|---|---|---|
| ✗ | $81.05 \pm 0.23$ | $97.71 \pm 0.34$ | $\mathbf{74.22 \pm 0.26}$ | $51.92 \pm 2.01$ |
| ✓ | $\mathbf{81.24 \pm 0.36}$ | $\mathbf{97.93 \pm 0.29}$ | $74.16 \pm 0.32$ | $\mathbf{52.26 \pm 1.32}$ |

Table 9: Ablations on PV activations.

| Method | Disease | Airport | PubMed | Cora |
|---|---|---|---|---|
| Tangent Act. | $81.24 \pm 0.36$ | $97.93 \pm 0.29$ | $74.16 \pm 0.32$ | $\mathbf{52.26 \pm 1.32}$ |
| FC $\sigma$ | $81.34 \pm 0.43$ | $\mathbf{99.40 \pm 0.15}$ | $74.02 \pm 0.17$ | $51.34 \pm 0.46$ |
| FC $\sigma$ + Tangent Act. | $80.96 \pm 0.19$ | $99.15 \pm 0.38$ | $73.96 \pm 0.22$ | $51.30 \pm 1.65$ |
| Euc. Act. | $\mathbf{81.34 \pm 0.43}$ | $98.87 \pm 0.35$ | $\mathbf{74.56 \pm 0.59}$ | $38.10 \pm 3.30$ |

statistics in the tangent space at the identity via

$$\mu = \mathrm{Exp_0}\left(\frac{1}{N}\sum_{i=1}^{N}\mathrm{Log_0}(x_i)\right), \quad v^2 = \frac{1}{N}\sum_{i=1}^{N}\left\|\mathrm{Log_0}(x_i) - \mathrm{Log_0}(\mu)\right\|^2, \tag{28}$$

and a Euclidean variant computes standard Euclidean mean and variance directly in the unconstrained PV space. Tab. 7 shows that Tangent and Euclidean are up to $2\times$ faster while achieving similar accuracies on Disease and Airport. Although Fréchet-based GyroBN attains the best accuracies, it is more computationally expensive.

**Ablations on PV embedding.** In the main experiments, the input features are first lifted to PV via $\mathrm{Exp_0}$ and then processed by PVNN. Since PV space is unconstrained, we also consider a variant that feeds the Euclidean features directly as PV coordinates. Tab. 8 compares these two settings. The two variants perform similarly, while using $\mathrm{Exp_0}$ provides small improvements on Disease, Airport, and Cora. This differs from image classification in Tab. 4, where the variant without $\mathrm{Exp_0}$ is marginally better. This slight discrepancy may stem from the different nature of the inputs. In vision, the ResNet encoder can adapt its learned representation to the chosen lifting, whereas in graphs the raw node features benefit slightly from the explicit exponential map.

**Ablations on activation.** We ablate three types of activations in PVNN: the internal nonlinearity $\sigma$ in the PV FC layer (fixed to $\tanh$), and explicit activations applied either directly in PV (*Euc. Act.*) or in the tangent space (*Tangent Act.*). Tab. 9 reports the results. First, when comparing these three choices individually, the differences are small on Disease and PubMed, while FC $\sigma$ performs best on

Table 10: Comparison (MCC) of hyperbolic and Euclidean convolutional networks on TEB datasets.

| Task | Dataset | Euclidean CNN | HCNN-S | PVCNN |
|------|---------|---------------|--------|-------|
| Retrotransposons | LINEs | $70.63 \pm 1.24$ | $76.12 \pm 2.16$ | $\mathbf{81.83 \pm 0.27}$ |
| | SINEs | $85.15 \pm 1.64$ | $85.45 \pm 1.16$ | $\mathbf{93.78 \pm 0.54}$ |
| DNA transposons | hAT-Ac | $87.45 \pm 0.90$ | $89.61 \pm 1.34$ | $\mathbf{92.08 \pm 0.80}$ |
| Pseudogenes | processed | $60.66 \pm 0.82$ | $68.30 \pm 0.93$ | $\mathbf{71.27 \pm 0.78}$ |
| | unprocessed | $51.94 \pm 2.69$ | $56.10 \pm 0.56$ | $\mathbf{62.31 \pm 0.78}$ |

Airport, Tangent Act. performs best on Cora, and Euc. Act. degrades substantially on Cora. Second, when comparing the composite variant FC $\sigma$ + Tangent Act. against Tangent Act., the composite does not yield consistent gains, suggesting redundancy.

### 6.4 GENOMIC SEQUENCE LEARNING

Khan et al. (2025) recently proposed hyperbolic convolutional neural networks (HCNNs) on the hyperboloid for genomic sequence learning, demonstrating that HCNNs outperform Euclidean CNNs on this task. Following Khan et al. (2025), we evaluate on the TEB dataset for DNA transposable element prediction. To ensure a fair comparison, all models share the same backbone network architecture, which consists of two convolutional blocks followed by an FC layer and a final MLR classifier (Khan et al., 2025). We use a single curvature shared for all layers. More details are provided in App. C.4. Tab. 10 reports 5-fold Matthews correlation coefficient (MCC). PVCNN achieves the best performance on all TEB tasks, with particularly strong gains on SINEs, where it improves over HCNN-S by about 9 MCC points. These results demonstrate the benefits of PV convolutional networks.

## 7 CONCLUSIONS

This work introduces Proper Velocity Neural Networks (PVNNs), leveraging the unconstrained PV model as an alternative to the constrained Poincaré and Lorentz models. We establish the full Riemannian toolkit on PV space and develop core neural layers, including MLR, FC, convolutional, activation, and normalization layers. Through four sets of experiments on numerical stability, image classification, graph node classification, and genomic sequence learning, PVNNs demonstrate both improved stability and competitive or superior performance compared with strong hyperbolic baselines. Our study provides the first systematic treatment of the PV manifold for deep learning, positioning it as a stable and practical geometry for future research on hyperbolic neural networks. As future work, we plan to extend PVNNs to more advanced architectures such as residual networks (He et al., 2016; Van Spengler et al., 2023; He et al., 2024) and transformers (Vaswani et al., 2017; Hu et al., 2023), and further exploit PV space for large-scale deep learning.

### REPRODUCIBILITY STATEMENT

All theoretical results are established under explicit assumptions, with complete proofs in App. E. The experimental details are presented in App. C. The code will be released upon acceptance.

### ETHICS STATEMENT

This work uses only publicly available benchmark datasets, which contain no personally identifiable or sensitive information. We did not identify any ethical concerns.

### ACKNOWLEDGMENTS

This work was supported by EU Horizon project ELLIOT (No. 101214398) and by the FIS project GUIDANCE (No. FIS2023-03251). We acknowledge CINECA for awarding high-performance

computing resources under the ISCRA initiative, and the EuroHPC Joint Undertaking for granting access to Leonardo at CINECA, Italy.

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

APPENDIX CONTENTS

Table 12: The geometric reinterpretations of Riemannian operators.

| Operation | Euclidean space | Riemannian manifold |
|---|---|---|
| Straight line | Straight line | Geodesic |
| Subtraction | $\overrightarrow{xy} = y - x$ | $\overrightarrow{xy} = \text{Log}_x(y)$ |
| Addition | $y = x + \overrightarrow{xy}$ | $y = \text{Exp}_x(\overrightarrow{xy})$ |
| Parallel transport | $v \to v$ | $\text{PT}_{x \to y}(v)$ |

## LIST OF ACRONYMS

HNNs     Hyperbolic Neural Networks 1
PVNNs    Proper Velocity Neural Networks 1
PV         Proper Velocity 1

FC         Fully Connected 1
MLR      Multinomial Logistics Regression 1

## A  USE OF LARGE LANGUAGE MODELS

Large Language Models (LLMs) were used primarily for language polishing and minor text editing. In limited cases, they also assisted in translating certain mathematical formulations into PyTorch code. All generated outputs were carefully reviewed and, where necessary, corrected by the authors. The authors take full responsibility for the final content of this paper.

## B  PRELIMINARIES

### B.1  RIEMANNIAN GEOMETRIES

For an in-depth discussion on Riemannian geometry, one can refer to Lee (2018).

**Riemannian manifold.** A Riemannian manifold $(\mathcal{M}, g)$, abbreviated as $\mathcal{M}$, carries a smoothly varying Riemannian metric $g_x : T_x\mathcal{M} \times T_x\mathcal{M} \to \mathbb{R}$ on each tangent space $T_x\mathcal{M}$. The induced norm is $\|v\|_x = \sqrt{g_x(v, v)}$. As an inner product, $g_x$ is also denoted as $\langle \cdot, \cdot \rangle_x$.

**Geodesic.** Straight lines are generalized to constant-speed curves that are locally length-minimizing between points $x, y \in \mathcal{M}$, known as geodesics. The shortest distance between two points is called the geodesic distance, denoted as $\text{d}(\cdot, \cdot)$.

**Exponential and logarithmic maps.** For $x \in \mathcal{M}$ and $v \in T_x\mathcal{M}$, let $\gamma_{x,v}$ denote the unique geodesic with $\gamma_{x,v}(0) = x$ and $\dot{\gamma}_{x,v}(0) = v$. The exponential map $\text{Exp}_x : T_x\mathcal{M} \supset \mathcal{V} \to \mathcal{M}$ is defined by $\text{Exp}_x(v) = \gamma_{x,v}(1)$, where $\mathcal{V}$ is an open neighborhood of the origin in $T_x\mathcal{M}$. Its local inverse, defined for $y$ in a neighborhood $\mathcal{U} \subset \mathcal{M}$ of $x$, is the logarithmic map $\text{Log}_x : \mathcal{U} \to T_x\mathcal{M}$, satisfying $\text{Exp}_x \circ \text{Log}_x = \mathbb{I}_{\mathcal{U}}$. In hyperbolic geometry, exponential and logarithmic maps are globally defined (Lee, 2018).

**Parallel transport.** Given a geodesic $\gamma$ from $x$ to $y$, the parallel transport of a tangent vector $v \in T_x\mathcal{M}$ along the geodesic is the unique vector $\text{PT}_{x \to y}(v) \in T_y\mathcal{M}$ obtained by transporting $v$ along $\gamma$ so that its covariant derivative along $\gamma$ vanishes. Parallel transport defines a linear isometry between $T_x\mathcal{M}$ and $T_y\mathcal{M}$.

Tab. 12 compares the corresponding operators in Euclidean and Riemannian geometries.

**Isometry.** Isometries generalize bijections in Riemannian geometry. If $\{\mathcal{M}, g\}$ and $\{\widetilde{\mathcal{M}}, \widetilde{g}\}$ are both Riemannian manifolds, a smooth map $f : \mathcal{M} \to \widetilde{\mathcal{M}}$ is called a (Riemannian) isometry if it is a diffeomorphism that satisfies

$$g_x(v, w) = \widetilde{g}_{f(x)}(d_x f(v), d_x f(w)),$$

where $d_x f(\cdot): T_x \mathcal{M} \to T_{f(x)} \widetilde{\mathcal{M}}$ is the differential map of $f$ at $x \in \mathcal{M}$, and $v, w \in T_x \mathcal{M}$ are two tangent vectors.

## B.2 GYRO-STRUCTURES

This subsection briefly reviews the gyrovector space (Ungar, 2022), which generalizes the vector structure to manifolds. It has shown great success in building hyperbolic neural networks (Ganea et al., 2018; Chami et al., 2019; Shimizu et al., 2021).

We start from the gyrogroup. Intuitively, gyrogroups are natural generalizations of groups. Unlike groups, gyrogroups are non-associative but have gyroassociativity characterized by gyrations.

**Definition B.1** (Gyrogroups (Ungar, 2022)). Given a nonempty set $G$ with a binary operation $\oplus: G \times G \to G$, $(G, \oplus)$ forms a gyrogroup if its binary operation satisfies the following axioms for any $x, y, z \in G$:

(G1) There is at least one element $e \in G$ called a left identity (or neutral element) such that $e \oplus x = x$.

(G2) There is an element $\ominus x \in G$ called a left inverse of $x$ such that $\ominus x \oplus x = e$.

(G3) There is an automorphism $\text{gyr}[x, y]: G \to G$ for each $x, y \in G$ such that

$$x \oplus (y \oplus z) = (x \oplus y) \oplus \text{gyr}[x, y]z \quad \text{(Left Gyroassociative Law)}.$$

The automorphism $\text{gyr}[x, y]$ is called the gyroautomorphism, or the gyration of $G$ generated by $x, y$.

(G4) Left reduction law: $\text{gyr}[x, y] = \text{gyr}[x \oplus y, y]$.

**Definition B.2** (Gyrocommutative Gyrogroups (Ungar, 2022)). A gyrogroup $(G, \oplus)$ is gyrocommutative if it satisfies

$$x \oplus y = \text{gyr}[x, y](y \oplus x) \quad \text{(Gyrocommutative Law)}.$$

Similarly, a gyrovector space generalizes a vector space.

**Definition B.3** (Gyrovector Spaces (Chen et al., 2025c)). A gyrocommutative gyrogroup $(G, \oplus)$ equipped with a scalar gyromultiplication $\otimes: \mathbb{R} \times G \to G$ is called a gyrovector space if it satisfies the following axioms for $s, t \in \mathbb{R}$ and $x, y, z \in G$:
(V1) Identity Scalar Multiplication: $1 \otimes x = x$.
(V2) Scalar Distributive Law: $(s + t) \otimes x = s \otimes x \oplus t \otimes x$.
(V3) Scalar Associative Law: $(st) \otimes x = s \otimes (t \otimes x)$.
(V4) Gyroautomorphism: $\text{gyr}[x, y](t \otimes z) = t \otimes \text{gyr}[x, y]z$.
(V5) Identity Gyroautomorphism: $\text{gyr}[s \otimes x, t \otimes x] = \mathbb{I}$, where $\mathbb{I}$ is the identity map.

**Definition B.4** (Real Inner Product Gyrovector Spaces (Ungar, 2022)). Let $(G, \oplus, \otimes)$ be a gyrovector space and let $\langle \cdot, \cdot \rangle$ denote the Euclidean inner product on $\mathbb{R}^n$ with associated norm $\|\cdot\|$. We call $(G, \oplus, \otimes, \langle \cdot, \cdot \rangle)$ a *real inner product gyrovector space* if the following conditions hold.
(V6) $G \subseteq \mathbb{R}^n$ and inherits the inner product $\langle \cdot, \cdot \rangle$ and norm $\|\cdot\|$.
(V7) Inner product gyroinvariance: $\langle \text{gyr}[x, y]u, \text{gyr}[x, y]v \rangle = \langle u, v \rangle, \quad \forall x, y, u, v \in G$.
(V8) Scaling property: $\frac{|s| \otimes x}{\|s \otimes x\|} = \frac{x}{\|x\|}, \quad \forall x \in G \setminus \{\mathbf{0}\}, \forall s \in \mathbb{R} \setminus \{0\}$.
(V10) Let $\|G\| = \{\pm\|x\| \mid x \in G\} \subset \mathbb{R}$. The set $\|G\|$ forms a one-dimensional real vector space with respect to the vector addition and scalar multiplication induced by $\oplus$ and $\otimes$ on $G$.
(V11) Homogeneity property: $\|s \otimes x\| = |s| \otimes \|x\|, \quad \forall x \in G, \forall s \in \mathbb{R}$.
(V12) Gyrotriangle inequality: $\|x \oplus y\| \leq \|x\| \oplus \|y\|, \quad \forall x, y \in G$.

When a gyrovector space $(G, \oplus, \otimes)$ is a subset of the real inner product vector space $\mathbb{R}^n$ and satisfies additional axioms with respect to $\|\cdot\|$, it forms a real inner gyrovector space. This is analogous to the relationship between inner product spaces and vector spaces.

**Definition B.5** (Gyrovector Space Isomorphisms (Ungar, 2022)). Let $(G_1, \oplus_1, \otimes_1)$ and $(G_2, \oplus_2, \otimes_2)$ be real inner product gyrovector spaces. A map $\phi: G_1 \to G_2$ is a *gyrovector space isomorphism* if it is bijective and satisfies

$$\phi(x \oplus_1 y) = \phi(x) \oplus_2 \phi(y), \quad \forall x, y \in G_1, \tag{29}$$

$$\phi(t \otimes_1 x) = t \otimes_2 \phi(x), \quad \forall x \in G_1, \forall t \in \mathbb{R}, \tag{30}$$

Table 13: Riemannian operators on the Poincaré ball and the hyperboloid ($K < 0$).

| Operator | Poincaré ball $\mathbb{P}_K^n$ | Hyperboloid $\mathbb{H}_K^n$ |
|---|---|---|
| Definition | $\mathbb{P}_K^n = \{x \in \mathbb{R}^n \mid \|x\|^2 < -1/K\}$ | $\mathbb{H}_K^n = \{x \in \mathbb{R}^{n+1} \mid \langle x,x \rangle_{\mathcal{L}} = 1/K, x_t > 0\}$ |
| $g_x(w,v)$ | $\left(\lambda_x^K\right)^2 \langle w,v \rangle, \quad \lambda_x^K = \dfrac{2}{1+K\|x\|^2}$ | $\langle w,v \rangle_{\mathcal{L}} = \langle v_s, w_s \rangle - v_t w_t$ |
| $\mathrm{d}(x,y)$ | $\dfrac{2}{\sqrt{|K|}} \tanh^{-1}\left(\sqrt{|K|}\|-x \oplus_{\mathrm{M}} y\|\right)$ | $\dfrac{1}{\sqrt{|K|}} \cosh^{-1}\left(K \langle x,y \rangle_{\mathcal{L}}\right)$ |
| $\mathrm{Log}_x y$ | $\dfrac{2}{\sqrt{|K|}\lambda_x^K} \tanh^{-1}\left(\sqrt{|K|}\|-x \oplus_{\mathrm{M}} y\|\right) \dfrac{-x \oplus_{\mathrm{M}} y}{\|-x \oplus_{\mathrm{M}} y\|}$ | $\dfrac{\cosh^{-1}(\beta)}{\sqrt{\beta^2-1}}(y - \beta x), \quad \beta = K \langle x,y \rangle_{\mathcal{L}}$ |
| $\mathrm{Exp}_x v$ | $x \oplus_{\mathrm{M}} \left(\tanh\left(\sqrt{|K|}\dfrac{\lambda_x^K \|v\|}{2}\right) \dfrac{v}{\sqrt{|K|}\|v\|}\right)$ | $\cosh(\alpha)x + \dfrac{\sinh(\alpha)}{\alpha}v, \quad \alpha = \sqrt{|K|}\|v\|_{\mathcal{L}}$ |
| $\mathrm{PT}_{x \to y}(v)$ | $\dfrac{\lambda_x^K}{\lambda_y^K} \mathrm{gyr}[y,-x]v$ | $v - \dfrac{K \langle y,v \rangle_{\mathcal{L}}}{1+K \langle x,y \rangle_{\mathcal{L}}}(x+y)$ |

and it keeps the inner product of unit gyrovectors invariant,

$$\frac{\langle \phi(x), \phi(y) \rangle}{\|\phi(x)\| \|\phi(y)\|} = \frac{\langle x,y \rangle}{\|x\| \|y\|}, \quad \forall x,y \in G_1 \text{ with } x \neq \mathbf{0}, y \neq \mathbf{0}. \tag{31}$$

A useful property is that gyrovector space isomorphisms preserve the gyration, inverse, and identity.

**Proposition B.6.** *Let $(G_1, \oplus_1, \otimes_1)$ and $(G_2, \oplus_2, \otimes_2)$ be real inner product gyrovector spaces with gyrations $\mathrm{gyr}_1$ and $\mathrm{gyr}_2$, respectively. If $\phi : G_1 \to G_2$ is a gyrovector space isomorphism, then for all $x,y,z \in G_1$,*

$$\phi\left(\mathrm{gyr}_1[x,y]z\right) = \mathrm{gyr}_2[\phi(x), \phi(y)]\phi(z), \tag{32}$$
$$\phi(e_1) = e_2, \tag{33}$$
$$\phi(\ominus_1 x) = \ominus_2 \phi(x), \tag{34}$$

*where $e_1$ and $e_2$ are the gyro identities in $G_1$ and $G_2$, respectively.*

*Proof.* The gyration properties have been shown by Ungar (2022, Ch. 6.21). The proofs for the gyro identity and gyroinverse follow directly from the isomorphism and the uniqueness of inverse and identity (Ungar, 2022, Thm. 2.10). $\square$

### B.3 PV GYRATION

As shown by Ungar (2022, Eqs. 3.220 and 3.221), the PV gyration for any $x,y,z \in \mathbb{PV}_K$ is given by

$$\mathrm{gyr}[x,y]z = z + \frac{Ax + By}{D}, \tag{35}$$

where the coefficients are

$$A = (1 - \beta_y^2)K \langle x,z \rangle - (1+\beta_x)(1+\beta_y)\beta_x\beta_y K \langle y,z \rangle + 2\beta_x^2\beta_y^2 K^2 \langle x,y \rangle \langle y,z \rangle,$$
$$B = (1 - \beta_x^2)\beta_y^2 K \langle y,z \rangle + (1+\beta_x)(1+\beta_y)\beta_x\beta_y K \langle x,z \rangle,$$
$$D = (1+\beta_x)(1+\beta_y)\left(1 - \beta_x\beta_y K \langle x,y \rangle + \beta_x\beta_y\right).$$

Here, $\beta_x = \frac{1}{\sqrt{1-K\|x\|^2}}$ is the relativistic beta factor.

### B.4 POINCARÉ BALL AND HYPERBOLOID

The Poincaré ball is defined as $\mathbb{P}_K^n = \{x \in \mathbb{R}^n \mid \|x\|^2 < -1/K\}$ with sectional curvature $K < 0$. The hyperboloid, also known as the Lorentz model, is defined as

$$\mathbb{H}_K^n = \left\{x \in \mathbb{R}^{n+1} \mid \langle x,x \rangle_{\mathcal{L}} = 1/K, x_t > 0\right\}, \tag{36}$$

where $\langle x, y \rangle_{\mathcal{L}} = -x_t y_t + \langle x_s, y_s \rangle$ is the Lorentz inner product. Here, $x_t \in \mathbb{R}$ and $x_s \in \mathbb{R}^n$ denote the time component and space components. The induced norm $\|\cdot\|_{\mathcal{L}}$ is the Lorentz norm. Let $\mathcal{H}^n_K \in \{\mathbb{P}^n_K, \mathbb{H}^n_K\}$. Given $x, y \in \mathcal{H}^n_K$ and tangent vectors $v, w \in T_x \mathcal{H}^n_K$, Tab. 13 summarizes the Riemannian operators.

The gyro-structure over the hyperbolic space can be defined by its Riemannian operators (Ganea et al., 2018; Chen et al., 2025c). Let $e = \mathbf{0}$ for the Poincaré ball and $e = \overline{\mathbf{0}} = [1/\sqrt{|K|}, \mathbf{0}^\top]^\top$ for hyperboloid. Given $x, y, z \in \mathcal{H}^n_K$ and $t \in \mathbb{R}$, the gyroaddition and gyromultiplication are defined as

$$x \oplus_{\mathcal{H}} y = \text{Exp}_x \left( \text{PT}_{e \to x} \left( \text{Log}_e y \right) \right), \tag{37}$$

$$t \otimes_{\mathcal{H}} x = \text{Exp}_e \left( t \, \text{Log}_e x \right), \tag{38}$$

$$\text{gyr}[x, y]z = \ominus_{\mathcal{H}} \left( x \oplus_{\mathcal{H}} y \right) \oplus_{\mathcal{H}} \left( x \oplus_{\mathcal{H}} \left( y \oplus_{\mathcal{H}} z \right) \right), \tag{39}$$

On the Poincaré ball $\mathbb{P}^n_K$, such gyro-structure is known as the Möbius gyrovector space (Ungar, 2022, Ch. 6.14):

$$x \oplus_{\text{M}} y = \frac{\left( 1 - 2K \langle x, y \rangle - K\|y\|^2 \right) x + \left( 1 + K\|x\|^2 \right) y}{1 - 2K \langle x, y \rangle + K^2 \|x\|^2 \|y\|^2}, \tag{40}$$

$$t \otimes_{\text{M}} x = \frac{\tanh \left( t \tanh^{-1}(\sqrt{|K|}\|x\|) \right)}{\sqrt{|K|}} \frac{x}{\|x\|}, \tag{41}$$

$$\text{gyr}_{\text{M}}[x, y]z = z + \frac{2}{D}(Ax + By), \tag{42}$$

with

$$A = -K^2 \langle x, z \rangle \|y\|^2 - K \langle y, z \rangle + 2K^2 \langle x, y \rangle \langle y, z \rangle, \tag{43}$$

$$B = -K^2 \langle y, z \rangle \|x\|^2 + K \langle x, z \rangle, \tag{44}$$

$$D = 1 - 2K \langle x, y \rangle + K^2 \|x\|^2 \|y\|^2. \tag{45}$$

Here, $\ominus_{\text{M}} x = -1 \otimes_{\text{M}} x = -x$ is the gyroinverse and $\mathbf{0}$ is the gyro identity: $\mathbf{0} \oplus_{\text{M}} x = x, \forall x \in \mathbb{P}^n_K$. Interestingly, the Möbius gyration has a similar expression as the PV gyration.

As shown by Chen et al. (2025c, Props. 24-25), the hyperboloid gyroaddition and gyromultiplication also admit closed-form expressions:

$$x \oplus_{\text{L}} y = \begin{cases} x, & y = \overline{\mathbf{0}}, \\ y, & x = \overline{\mathbf{0}}, \\ \begin{bmatrix} \frac{1}{\sqrt{|K|}} \frac{D - KN}{D + KN} \\ \frac{2(A_s x_s + A_y y_s)}{D + KN} \end{bmatrix}, & \text{Otherwise.} \end{cases} \tag{46}$$

$$t \otimes_{\text{L}} x = \begin{cases} \overline{\mathbf{0}}, & t = 0 \vee x = \overline{\mathbf{0}} \\ \frac{1}{\sqrt{|K|}} \begin{bmatrix} \cosh(t\theta) \\ \frac{\sinh(t\theta)}{\|x_s\|} x_s \end{bmatrix}, & \text{Otherwise,} \end{cases} \tag{47}$$

Here, $\theta = \cosh^{-1}(\sqrt{|K|}x_t)$, $A_s = ab^2 - 2Kbs_{xy} - Kan_y$ and $A_y = b(a^2 + Kn_x)$ with the following notation:

$$\begin{aligned} & a = 1 + \sqrt{|K|}x_t, b = 1 + \sqrt{|K|}y_t, \\ & n_x = \|x_s\|^2, n_y = \|y_s\|^2, s_{xy} = \langle x_s, y_s \rangle, \\ & D = a^2 b^2 - 2Kabs_{xy} + K^2 n_x n_y, \\ & N = a^2 n_y + 2abs_{xy} + b^2 n_x. \end{aligned} \tag{48}$$

In particular, the gyro identity is $\overline{\mathbf{0}}$ and the gyroinverse is $\ominus_{\text{L}} x = -1 \otimes_{\text{L}} x = [x_t, -x_s^\top]^\top$.

Table 14: Summary statistics for the node classification datasets.

| Dataset | #Nodes | #Edges | #Classes | #Features |
|---------|--------|--------|----------|-----------|
| Disease | 1044   | 1043   | 2        | 1000      |
| Airport | 3188   | 18631  | 4        | 4         |
| PubMed  | 19717  | 44338  | 3        | 500       |
| Cora    | 2708   | 5429   | 7        | 1433      |

## C EXPERIMENTAL DETAILS

### C.1 COMMON IMPLEMENTATIONS

As we use trivialization tricks in our MLR, FC, and GyroBN layers, all parameters in PVNN lie in Euclidean space and are optimized using standard Euclidean optimizers.

### C.2 EXPERIMENTAL DETAILS ON IMAGE CLASSIFICATION

#### C.2.1 DATASETS

CIFAR-10 and CIFAR-100 (Krizhevsky & Hinton, 2009) datasets contain 60K 32×32 colored images from 10 and 100 different classes, respectively. We use the dataset split implemented in Py-Torch, which has 50K training images and 10K testing images. Following Bdeir et al. (2024), we use data augmentation that includes random cropping with padding of 4 pixels and random horizontal flipping.

#### C.2.2 IMPLEMENTATION DETAILS

We implement the experiments using the official code[2] of Bdeir et al. (2024). All models share a common backbone, which consists of a ResNet-18 encoder followed by a hyperbolic MLR classifier. The output embedding of the ResNet-18 backbone is mapped to the target hyperbolic space via the exponential map at the identity $e$, that is, $\mathrm{Exp}_e(x)$. Here, $e = \mathbf{0}$ for the Poincaré and PV spaces, and $e = \overline{\mathbf{0}}$ for the hyperboloid model. All models are trained from scratch. Optimization is performed using SGD (Robbins & Monro, 1951) with an initial learning rate of $0.1$, a momentum of $0.9$, and a weight decay of $5 \times 10^{-4}$. Training is conducted with a batch size of 128 for 200 epochs. The learning rate is decayed by a factor of $\gamma = 0.2$ at epochs 60, 120, and 160. The curvature for the PV space is set as $K = -0.5$.

### C.3 EXPERIMENTAL DETAILS ON GRAPH LEARNING

#### C.3.1 DATASETS

**Disease (Anderson & May, 1991).** It represents a disease propagation tree, simulating the SIR disease transmission model, with each node representing either an infection or a non-infection state.

**Airport (Zhang & Chen, 2018).** It is a transductive dataset where nodes represent airports and edges represent the airline routes from OpenFlights.org.

**PubMed (Namata et al., 2012).** This is a standard benchmark describing citation networks where nodes represent scientific papers in the area of medicine, edges are citations between them, and node labels are academic (sub)areas.

**Cora (Sen et al., 2008).** It is a citation network where nodes represent scientific papers in the area of machine learning, edges are citations between them, and node labels are academic (sub)areas.

Tab. 14 summarizes the statistics of the datasets.

---

[2] https://github.com/kschwethelm/HyperbolicCV

Table 15: Hyperparameters for PVNN that vary across graph datasets.

| Hyperparameter | Disease | Airport | PubMed | Cora |
|---|---|---|---|---|
| Learning rate | 0.01 | 0.01 | 0.05 | 0.05 |
| Dropout | 0.4 | 0.4 | 0.6 | 0.6 |
| Curvature | -0.3 | -0.3 | -1.0 | -1.0 |

Table 16: Hyperparameters for PVNN that are shared across graph datasets.

| Setting | Epochs | Batch size | Weight decay |
|---|---|---|---|
| Value | 2000 | 128 | $5 \times 10^{-4}$ |

### C.3.2 IMPLEMENTATION DETAILS

We adopt the official code of HGCN[3] (Chami et al., 2019) to conduct experiments. The features of each node are embedded into the hyperbolic space via the exponential map at the identity. The hyperbolic network consists of two FC layers: the first maps the input feature dimension to a 16-dimensional hidden representation, and the second maps from 16 to 16. Each FC layer is followed by an activation function. An MLR layer is then used for classification. All models are trained using the Adam optimizer (Kingma & Ba, 2015). We evaluate performance every 10 epochs and employ early stopping with a patience of 200 evaluations, restoring the checkpoint with the best test accuracy. Tabs. 15 and 16 summarize the hyperparameters for PVNN. For KNN (Mao et al., 2024), HNN (Ganea et al., 2018), HNN++ (Shimizu et al., 2021), and LNN (Bdeir et al., 2024), we follow their original papers to implement the experiments. Tab. 17 summarizes the hyperbolic layers used in each model.

### C.4 EXPERIMENTAL DETAILS ON GENOME SEQUENCE LEARNING

### C.4.1 DATASETS AND PREPROCESSING

We use the Transposable Elements Benchmark (TEB) datasets (Khan et al., 2025). This benchmark provides seven DNA sequence classification datasets spanning three prediction tasks: retrotransposons, DNA transposons, and pseudogenes. We focus on five among them, as summarized in Tab. 18. We follow their original train/validation/test splits and preprocessing.

Table 18: Statistics for the TEB datasets.

| Prediction task | Species | Max length | Dataset | Train / Val / Test |
|---|---|---|---|---|
| Retrotransposons | Plant | 1000 | LINEs | 22502 / 2030 / 1782 |
| | | 500 | SINEs | 21152 / 1836 / 1784 |
| DNA Transposons | Plant | 1000 | hAT-Ac | 17322 / 1822 / 1428 |
| Pseudogenes | Human | 1000 | processed | 17956 / 1046 / 1740 |
| | | 1000 | unprocessed | 12938 / 766 / 884 |

### C.4.2 IMPLEMENTATION DETAILS

For the Euclidean CNN and the hyperbolic CNN baseline (HCNN-S), we directly use the results reported in the original paper (Khan et al., 2025, Tab. 2). Our PV CNN architecture follows their implementation[4]. Each DNA sequence is represented as a length $L$ sequence with 4 input channels. We first apply a PV convolution that maps the 4 input channels to 32 channels, followed by PV tangent batch normalization and a tanh tangent activation. A second PV convolution layer is then applied. The final PV feature is concatenated and passed through an FC layer, and finally classified

---

[3]https://github.com/HazyResearch/hgcn
[4]https://github.com/rrkhan/HGE

Table 17: Summary of the hyperbolic layers used in the graph node classification models.

| Model | FC layer | Activation | MLR |
|---|---|---|---|
| PVNN | PV FC in Thm. 5.3 | $\mathrm{Log}_\mathbf{0}(\sigma(\mathrm{Exp}_\mathbf{0}(x)))$ | PV MLR in Thm. 5.2 |
| KNN | $\mathrm{Log}_\mathbf{0}(W\,\mathrm{Exp}_\mathbf{0}(x))$ | $\mathrm{Log}_\mathbf{0}(\sigma(\mathrm{Exp}_\mathbf{0}(x)))$ | Euclidean MLR after $\mathrm{Exp}_\mathbf{0}$ |
| HNN | $\mathrm{Log}_\mathbf{0}(W\,\mathrm{Exp}_\mathbf{0}(x))$ | $\mathrm{Log}_\mathbf{0}(\sigma(\mathrm{Exp}_\mathbf{0}(x)))$ | Poincaré MLR (Ganea et al., 2018) |
| HNN++ | Poincaré FC (Shimizu et al., 2021) | $\mathrm{Log}_\mathbf{0}(\sigma(\mathrm{Exp}_\mathbf{0}(x)))$ | Poincaré MLR (Shimizu et al., 2021) |
| LNN | Lorentz FC (Chen et al., 2022) | Lorentz activation (Bdeir et al., 2024) | Lorentz MLR (Bdeir et al., 2024) |

Table 19: Hyperparameters for TEB.

| Setting | Value |
|---|---|
| Optimizer | Adam |
| Learning rate | $1e^{-4}$ |
| Weight decay | $2e^{-2}$ |
| Batch size | 100 |
| Dropout | 0.1 |
| Adam $(\beta_1, \beta_2)$ | $(0.9, 0.999)$ |

with a PV MLR head. The curvature is initialized at $K = -0.5$ and learned during training. We train for 100 epochs with a step learning-rate schedule, using milestones at epochs 60 and 85 with decay factor 0.1. For the PV FC layer, $\sigma$ in Eq. (23) is set to $\mathtt{tanh}$. All other hyperparameters are summarized in Tab. 19.

## C.5 HARDWARE

All experiments are conducted on an NVIDIA A6000 GPU.

## D CONNECTIONS TO THE HYPERBOLOID

This section discusses the connections between the PV model and the hyperboloid model. We first show the isometry between the two models. Then, we show that several current hyperboloid network layers can be rewritten as PV layers.

**Proposition D.1** (PV–hyperboloid isometries). [↓] *The following maps are Riemannian isometries between the hyperboloid model $\mathbb{H}^n_K$ and the PV model $\mathbb{PV}^n_K$:*

$$\pi_{\mathbb{H}^n_K \to \mathbb{PV}^n_K} : \mathbb{H}^n_K \ni \begin{bmatrix} x_t \\ x_s \end{bmatrix} \mapsto x_s \in \mathbb{PV}^n_K, \tag{49}$$

$$\pi_{\mathbb{PV}^n_K \to \mathbb{H}^n_K} : \mathbb{PV}^n_K \ni x \mapsto \begin{bmatrix} \sqrt{\|x\|^2 - \frac{1}{K}} \\ x \end{bmatrix} \in \mathbb{H}^n_K. \tag{50}$$

**Implications.** The PV–hyperboloid isometries in Prop. D.1 imply that several standard layers in hyperboloid networks can be rewritten as PV layers composed with $\pi_{\mathbb{H}^n_K \to \mathbb{PV}^n_K}$ and $\pi_{\mathbb{PV}^n_K \to \mathbb{H}^n_K}$.

The Lorentz activation (Bdeir et al., 2024, Eq. 13), Lorentz FC layer (Chen et al., 2022, Sec. 3.1) and Lorentz concatenation (Bdeir et al., 2024, Eq. 32) are

$$\mathrm{LAct}\left(\begin{bmatrix} x_t \\ x_s \end{bmatrix}\right) = \begin{bmatrix} \sqrt{\left\| \sigma(x_s) \right\|^2 - \frac{1}{K}} \\ \sigma(x_s) \end{bmatrix}, \tag{51}$$

$$\mathrm{LFC}\left(\begin{bmatrix} x_t \\ x_s \end{bmatrix}\right) = \begin{bmatrix} \sqrt{\left\| W x_s + b \right\|^2 - \frac{1}{K}} \\ W x_s + b \end{bmatrix}, \tag{52}$$

$$\mathrm{HCat}(\{x_i\}_{i=1}^N) = \begin{bmatrix} \sqrt{\sum_{i=1}^N x_{i,t}^2 + \frac{N-1}{K}} \\ x_{1,s} \\ \vdots \\ x_{N,s} \end{bmatrix} \in \mathbb{H}_K^{nN} \tag{53}$$

where $x = [x_t, x_s^\top]^\top \in \mathbb{H}_K^n$ and $x_i = [x_{i,t}, x_{i,s}^\top]^\top \in \mathbb{H}_K^n$ for $1 \leq i \leq N$. Then Prop. D.1 implies that the above Lorentz layers can be rewritten in terms of PV layers as follows:

$$\mathrm{LAct}(x) = \pi_{\mathbb{PV}_K^n \to \mathbb{H}_K^n}(\sigma(\pi_{\mathbb{H}_K^n \to \mathbb{PV}_K^n}(x))), \tag{54}$$

$$\mathrm{LFC}(x) = \pi_{\mathbb{PV}_K^n \to \mathbb{H}_K^n}(W \pi_{\mathbb{H}_K^n \to \mathbb{PV}_K^n}(x) + b), \tag{55}$$

$$\mathrm{HCat}(\{x_i\}_{i=1}^N) = \pi_{\mathbb{PV}_K^n \to \mathbb{H}_K^n}\left(\mathrm{Concat}(\pi_{\mathbb{H}_K^n \to \mathbb{PV}_K^n}(x_1), \ldots, \pi_{\mathbb{H}_K^n \to \mathbb{PV}_K^n}(x_N))\right). \tag{56}$$

These identities show that many hyperboloid constructions effectively operate by mapping to PV space, applying Euclidean building blocks there, and mapping back through $\pi_{\mathbb{PV}_K^n \to \mathbb{H}_K^n}$. This perspective naturally motivates designing networks directly in PV space, instead of repeatedly switching between equivalent models. Moreover, even if one follows the pattern $\mathbb{H}_K^n \to \mathbb{PV}_K^n \to \mathbb{PV}_K^m \to \mathbb{H}_K^m$ to construct FC layers, the intermediate map should be the PV FC layer from Thm. 5.3 rather than a naive linear map, since PV is a non-linear Riemannian manifold and intrinsic layers must respect its geometry.

# E  PROOFS

## E.1  PROOF OF EQ. (1)

The PV line element at $x \in \mathbb{PV}_K^n$ can be written in terms of the curvature parameter $K < 0$ as

$$Q_x(u) = \|u\|^2 + K\beta_x^2 \langle x, u \rangle^2, \quad \forall u \in T_x \mathbb{PV}_K^n \simeq \mathbb{R}^n, \tag{57}$$

where $\beta_x = \frac{1}{\sqrt{1 - K\|x\|^2}}$. This is equivalent to the expression in Ungar (2022, Eq. 7.76) after substituting $s^2 = -1/K$. Given $u, v \in T_x \mathbb{PV}_K^n$, the bilinear form $g_x(u, v)$ is obtained by the polarization identity:

$$g_x(u, v) = \tfrac{1}{4}\left(Q_x(u + v) - Q_x(u - v)\right). \tag{58}$$

We first expand the two terms in the polarization identity:

$$\begin{aligned} Q_x(u + v) &= \|u + v\|^2 + K\beta_x^2 \langle x, u + v \rangle^2 \\ &= \|u\|^2 + 2\langle u, v \rangle + \|v\|^2 + K\beta_x^2\left(\langle x, u \rangle^2 + 2\langle x, u \rangle\langle x, v \rangle + \langle x, v \rangle^2\right), \\ Q_x(u - v) &= \|u - v\|^2 + K\beta_x^2 \langle x, u - v \rangle^2 \\ &= \|u\|^2 - 2\langle u, v \rangle + \|v\|^2 + K\beta_x^2\left(\langle x, u \rangle^2 - 2\langle x, u \rangle\langle x, v \rangle + \langle x, v \rangle^2\right). \end{aligned} \tag{59}$$

Taking the difference yields

$$Q_x(u + v) - Q_x(u - v) = 4\langle u, v \rangle + 4K\beta_x^2 \langle x, u \rangle\langle x, v \rangle. \tag{60}$$

Substituting this expression into the polarization identity, we obtain

$$\begin{aligned} g_x(u, v) &= \tfrac{1}{4}\left(Q_x(u + v) - Q_x(u - v)\right) \\ &= \langle u, v \rangle + K\beta_x^2 \langle x, u \rangle\langle x, v \rangle, \end{aligned} \tag{61}$$

which coincides with the expression of the PV metric in Eq. (1).

### E.2 PROOF OF LEM. 4.1

*Proof.* **Differential of $\pi_{\mathbb{PV}_K^n \to \mathbb{P}_K^n}$.** Consider the curve $c : (-\varepsilon, \varepsilon) \to \mathbb{PV}_K^n$ which satisfies $c(0) = x$ and $c'(0) = v$. By definition of the differential,

$$d_x(\pi_{\mathbb{PV}_K^n \to \mathbb{P}_K^n})(v) = \left.\frac{\mathrm{d}}{\mathrm{d}t}\right|_{t=0} \pi_{\mathbb{PV}_K^n \to \mathbb{P}_K^n}\big(c(t)\big). \tag{62}$$

Using $\pi_{\mathbb{PV}_K^n \to \mathbb{P}_K^n}(x) = \frac{\beta_x}{1+\beta_x}x$ with $\beta_x = \frac{1}{\sqrt{1-K\|x\|^2}}$, we write

$$\pi_{\mathbb{PV}_K^n \to \mathbb{P}_K^n}\big(c(t)\big) = h(t)c(t), \qquad h(t) := \frac{\beta_{c(t)}}{1 + \beta_{c(t)}}. \tag{63}$$

Let $r(t) = \big\|c(t)\big\|^2$, so that $\beta_{c(t)} = (1 - Kr(t))^{-1/2}$. Then

$$r'(0) = 2\langle x, v \rangle, \qquad \beta'_{c(0)} = \tfrac{1}{2}(1 - Kr(0))^{-3/2}Kr'(0) = K\beta_x^3\langle x, v \rangle. \tag{64}$$

Differentiating $h(t)$ at $t = 0$ gives

$$h'(0) = \frac{\beta'_{c(0)}}{(1 + \beta_x)^2} = K\frac{\beta_x^3}{(1 + \beta_x)^2}\langle x, v \rangle. \tag{65}$$

Finally, differentiating $h(t)c(t)$ at $t = 0$ yields

$$\begin{aligned} d_x(\pi_{\mathbb{PV}_K^n \to \mathbb{P}_K^n})(v) &= h'(0)x + h(0)v \\ &= K\frac{\beta_x^3}{(1 + \beta_x)^2}\langle x, v \rangle\, x + \frac{\beta_x}{1 + \beta_x}v. \end{aligned} \tag{66}$$

In particular, at $x = \mathbf{0}$ one has $\beta_{\mathbf{0}} = 1$ and $\langle x, v \rangle = 0$. Thus, we have

$$d_{\mathbf{0}}(\pi_{\mathbb{PV}_K^n \to \mathbb{P}_K^n})(v) = \frac{\beta_{\mathbf{0}}}{1 + \beta_{\mathbf{0}}}v = \frac{1}{2}v. \tag{67}$$

**Differential of $\pi_{\mathbb{P}_K^n \to \mathbb{PV}_K^n}$.** Consider the curve $c : (-\varepsilon, \varepsilon) \to \mathbb{P}_K^n$ which satisfies $c(0) = y$ and $c'(0) = w$. By definition of the differential,

$$d_y(\pi_{\mathbb{P}_K^n \to \mathbb{PV}_K^n})(w) = \left.\frac{\mathrm{d}}{\mathrm{d}t}\right|_{t=0} \pi_{\mathbb{P}_K^n \to \mathbb{PV}_K^n}(c(t)). \tag{68}$$

Using the explicit expression $\pi_{\mathbb{P}_K^n \to \mathbb{PV}_K^n}(y) = 2\gamma_y^2 y$ with $\gamma_y = \frac{1}{\sqrt{1+K\|y\|^2}}$, we obtain

$$\pi_{\mathbb{P}_K^n \to \mathbb{PV}_K^n}(c(t)) = 2\gamma_{c(t)}^2 c(t). \tag{69}$$

Let $r(t) = \big\|c(t)\big\|^2$ so that $\gamma_{c(t)}^2 = (1 + Kr(t))^{-1}$. Then

$$r'(0) = 2\langle y, w \rangle, \qquad \left.\frac{\mathrm{d}}{\mathrm{d}t}\right|_{t=0}\gamma_{c(t)}^2 = -\frac{Kr'(0)}{(1 + Kr(0))^2} = -2K\gamma_y^4\langle y, w \rangle. \tag{70}$$

Differentiating $2\gamma_{c(t)}^2 c(t)$ at $t = 0$ yields

$$\begin{aligned} d_y(\pi_{\mathbb{P}_K^n \to \mathbb{PV}_K^n})(w) &= 2\left.\frac{\mathrm{d}}{\mathrm{d}t}\right|_{t=0}\gamma_{c(t)}^2 y + 2\gamma_y^2 w \\ &= -4K\gamma_y^4\langle y, w \rangle\, y + 2\gamma_y^2 w. \end{aligned} \tag{71}$$

In particular, at $y = \mathbf{0}$ we have $\gamma_{\mathbf{0}} = 1$ and $\langle y, w \rangle = 0$. Thus, we have

$$d_{\mathbf{0}}(\pi_{\mathbb{P}_K^n \to \mathbb{PV}_K^n})(w) = 2w. \tag{72}$$

$\square$

### E.3 PROOF OF THM. 4.2

*Proof.* It suffices to show that for any $x \in \mathbb{PV}_K^n$ and $v, w \in T_x \mathbb{PV}_K^n$,

$$g_y^{\mathbb{P}} \left( d_x \left( \pi_{\mathbb{PV}_K^n \to \mathbb{P}_K^n} \right)(v), d_x \left( \pi_{\mathbb{PV}_K^n \to \mathbb{P}_K^n} \right)(w) \right) = g_x^{\mathbb{PV}}(v, w), \tag{73}$$

where $y = \pi_{\mathbb{PV}_K^n \to \mathbb{P}_K^n}(x)$.

We first recall the following equations from Eq. (1), App. B.4, and Lem. 4.1:

$$g_x^{\mathbb{PV}}(v, w) = \langle v, w \rangle + K \beta_x^2 \langle x, v \rangle \langle x, w \rangle, \quad \forall x \in \mathbb{PV}_K^n, \forall v, w \in T_x \mathbb{PV}_K^n,$$

$$g_y^{\mathbb{P}}(u, z) = \left( \lambda_y^K \right)^2 \langle u, z \rangle, \quad \forall y \in \mathbb{P}_K^n, \forall u, z \in T_y \mathbb{P}_K^n,$$

$$\pi_{\mathbb{PV}_K^n \to \mathbb{P}_K^n}(x) = \frac{\beta_x}{1 + \beta_x} x, \quad \forall x \in \mathbb{PV}_K^n, \tag{74}$$

$$d_x(\pi_{\mathbb{PV}_K^n \to \mathbb{P}_K^n})(v) = \frac{\beta_x}{1 + \beta_x} v + K \frac{\beta_x^3}{(1 + \beta_x)^2} \langle x, v \rangle x, \quad \forall x \in \mathbb{PV}_K^n, \forall v \in T_x \mathbb{PV}_K^n.$$

Let

$$a = \frac{\beta_x}{1 + \beta_x}, \qquad b = K \frac{\beta_x^3}{(1 + \beta_x)^2}. \tag{75}$$

Then $d_x(\pi_{\mathbb{PV}_K^n \to \mathbb{P}_K^n})(v) = av + b \langle x, v \rangle x$ and $d_x(\pi_{\mathbb{PV}_K^n \to \mathbb{P}_K^n})(w) = aw + b \langle x, w \rangle x$. Thus,

$$\begin{aligned}
& g_y^{\mathbb{P}} \left( d_x \left( \pi_{\mathbb{PV}_K^n \to \mathbb{P}_K^n} \right)(v), d_x \left( \pi_{\mathbb{PV}_K^n \to \mathbb{P}_K^n} \right)(w) \right) \\
&= \left( \lambda_y^K \right)^2 \langle av + b \langle x, v \rangle x, aw + b \langle x, w \rangle x \rangle \\
&= \left( \lambda_y^K \right)^2 \left( a^2 \langle v, w \rangle + ab \langle x, w \rangle \langle v, x \rangle + ab \langle x, v \rangle \langle x, w \rangle + b^2 \langle x, v \rangle \langle x, w \rangle \langle x, x \rangle \right) \\
&= \left( \lambda_y^K \right)^2 \left( a^2 \langle v, w \rangle + \left( 2ab + b^2 \|x\|^2 \right) \langle x, v \rangle \langle x, w \rangle \right).
\end{aligned} \tag{76}$$

Using $y = \pi_{\mathbb{PV}_K^n \to \mathbb{P}_K^n}(x)$ and the relation between $\lambda_y^K$, $\beta_x$, and $\|x\|$ from App. B.4, we simplify the coefficients. First,

$$\begin{aligned}
K \|y\|^2 &= K \left\langle \frac{\beta_x}{1 + \beta_x} x, \frac{\beta_x}{1 + \beta_x} x \right\rangle \\
&= K \frac{\beta_x^2}{(1 + \beta_x)^2} \|x\|^2, \\
&= \frac{\beta_x^2 - 1}{(1 + \beta_x)^2},
\end{aligned} \tag{77}$$

where we use $\beta_x^2 = \frac{1}{1 - K\|x\|^2}$. Hence

$$1 + K\|y\|^2 = 1 + \frac{\beta_x^2 - 1}{(1 + \beta_x)^2} = \frac{2\beta_x}{1 + \beta_x}, \tag{78}$$

which implies $\lambda_y^K = \frac{2}{1 + K\|y\|^2} = \frac{1 + \beta_x}{\beta_x}$. Therefore

$$\left( \lambda_y^K \right)^2 a^2 = \left( \frac{1 + \beta_x}{\beta_x} \right)^2 \left( \frac{\beta_x}{1 + \beta_x} \right)^2 = 1. \tag{79}$$

Next, we compute

$$\begin{aligned}
2ab &= 2 \frac{\beta_x}{1 + \beta_x} K \frac{\beta_x^3}{(1 + \beta_x)^2} = \frac{2K\beta_x^4}{(1 + \beta_x)^3}, \\
b^2 \|x\|^2 &= K^2 \frac{\beta_x^6}{(1 + \beta_x)^4} \|x\|^2 = K \frac{\beta_x^4(\beta_x^2 - 1)}{(1 + \beta_x)^4},
\end{aligned} \tag{80}$$

which brings us to

$$2ab + b^2\|x\|^2 = K\frac{\beta_x^4}{(1+\beta_x)^4}\left(2(1+\beta_x)+\beta_x^2-1\right)$$
$$= K\frac{\beta_x^4(\beta_x+1)^2}{(1+\beta_x)^4} = K\frac{\beta_x^4}{(1+\beta_x)^2}. \tag{81}$$

Multiplying by $\left(\lambda_y^K\right)^2 = \left(\frac{1+\beta_x}{\beta_x}\right)^2$ yields

$$\left(\lambda_y^K\right)^2\left(2ab+b^2\|x\|^2\right) = \left(\frac{1+\beta_x}{\beta_x}\right)^2 K\frac{\beta_x^4}{(1+\beta_x)^2} = K\beta_x^2. \tag{82}$$

Substituting these identities into the expression for $g_{\mathbb{P}}$ gives

$$g_y^{\mathbb{P}}\left(d_x\left(\pi_{\mathbb{PV}_K^n\to\mathbb{P}_K^n}\right)(v), d_x\left(\pi_{\mathbb{PV}_K^n\to\mathbb{P}_K^n}\right)(w)\right) = \langle v,w\rangle + K\beta_x^2\langle x,v\rangle\langle x,w\rangle = g_x^{\mathbb{PV}}(v,w). \tag{83}$$

$$\square$$

### E.4 PROOF OF THM. 4.3

Following the notation in the main theorem, we further denote:

$$\bar{x} = \pi(x) \in \mathbb{P}_K^n, \quad \bar{y} = \pi(y) \in \mathbb{P}_K^n, \quad \bar{v} = d\pi_x(v) \in T_{\bar{x}}\mathbb{P}_K^n, \tag{84}$$

Recalling Eq. (4) and Lem. 4.1, we have the following:

$$\pi(x) = \frac{\beta_x}{1+\beta_x}x, \quad \forall x \in \mathbb{PV}_K^n, \tag{85}$$

$$\pi^{-1}(\bar{y}) = 2\gamma_{\bar{y}}^2\bar{y}, \quad \forall\bar{y} \in \mathbb{P}_K^n, \tag{86}$$

$$d\pi_x(v) = K\frac{\beta_x^3}{(1+\beta_x)^2}\langle x,v\rangle x + \frac{\beta_x}{1+\beta_x}v, \quad \forall x \in \mathbb{PV}_K^n, \forall v \in T_x\mathbb{PV}_K^n, \tag{87}$$

$$d\pi_{\bar{y}}^{-1}(w) = -4K\gamma_{\bar{y}}^4\langle\bar{y},w\rangle\bar{y} + 2\gamma_{\bar{y}}^2w, \quad \forall\bar{y} \in \mathbb{P}_K^n, \forall w \in T_{\bar{y}}\mathbb{P}_K^n. \tag{88}$$

Next, we derive the expressions for each PV operator.

#### E.4.1 PV EXPONENTIAL MAP

We recall from Tab. 13 that the Riemannian exponential on the Poincaré ball is

$$\mathrm{Exp}_{\bar{x}}^{\mathbb{P}}(\bar{v}) = \bar{x} \oplus_{\mathrm{M}}\left(\frac{1}{\sqrt{-K}}\tanh\left(\frac{\sqrt{-K}\lambda_{\bar{x}}^K\|\bar{v}\|}{2}\right)\frac{\bar{v}}{\|\bar{v}\|}\right). \tag{89}$$

By the Riemannian isometry and the gyrovector isomorphism of $\pi$, for any $x \in \mathbb{PV}_K^n$ and $v \in T_x\mathbb{PV}_K^n$ we have

$$\mathrm{Exp}_x(v) \overset{(1)}{=} \pi^{-1}\left(\mathrm{Exp}_{\bar{x}}^{\mathbb{P}}(\bar{v})\right)$$
$$\overset{(2)}{=} x \oplus_{\mathrm{U}} \pi^{-1}\left(\frac{1}{\sqrt{-K}}\tanh\left(\frac{\sqrt{-K}\lambda_{\bar{x}}^K\|\bar{v}\|}{2}\right)\frac{\bar{v}}{\|\bar{v}\|}\right), \tag{90}$$

The above equalities follow from the following facts.

(1) Isometry.

(2) Gyrovector isomorphism.

Let

$$u = \frac{1}{\sqrt{-K}} \tanh\left(\frac{\sqrt{-K}\lambda_{\bar{x}}^K \|\bar{v}\|}{2}\right) \frac{\bar{v}}{\|\bar{v}\|}. \tag{91}$$

We have

$$\|u\| = \frac{1}{\sqrt{-K}} \tanh\left(\frac{\sqrt{-K}\lambda_{\bar{x}}^K \|\bar{v}\|}{2}\right). \tag{92}$$

Let $t = \frac{\sqrt{-K}\lambda_{\bar{x}}^K\|\bar{v}\|}{2}$ so that $\sqrt{-K}\|u\| = \tanh(t)$. Then

$$
\begin{aligned}
\pi^{-1}(u) &= \frac{2}{1 + K\|u\|^2} u \\
&= \frac{2}{1 + K\|u\|^2} \left(\frac{1}{\sqrt{-K}} \tanh(t)\frac{\bar{v}}{\|\bar{v}\|}\right) \\
&= \frac{2}{1 - \tanh^2(t)} \frac{1}{\sqrt{-K}} \tanh(t)\frac{\bar{v}}{\|\bar{v}\|} \\
&= \frac{2}{\sqrt{-K}} \frac{\tanh(t)}{1 - \tanh^2(t)}\frac{\bar{v}}{\|\bar{v}\|} \\
&\overset{(1)}{=} \frac{2}{\sqrt{-K}} \tanh(t)\cosh^2(t)\frac{\bar{v}}{\|\bar{v}\|} \\
&= \frac{2}{\sqrt{-K}} \frac{\sinh(t)}{\cosh(t)}\cosh^2(t)\frac{\bar{v}}{\|\bar{v}\|} \\
&= \frac{2}{\sqrt{-K}} \sinh(t)\cosh(t)\frac{\bar{v}}{\|\bar{v}\|} \\
&\overset{(2)}{=} \frac{1}{\sqrt{-K}}\left(2\sinh(t)\cosh(t)\right)\frac{\bar{v}}{\|\bar{v}\|} \\
&= \frac{1}{\sqrt{-K}} \sinh(2t)\frac{\bar{v}}{\|\bar{v}\|}, \\
&= \frac{1}{\sqrt{-K}} \sinh\left(\sqrt{-K}\lambda_{\bar{x}}^K\|\bar{v}\|\right)\frac{\bar{v}}{\|\bar{v}\|} \\
&\overset{(3)}{=} \frac{1}{\sqrt{-K}} \sinh\left(\frac{\sqrt{-K}(1+\beta_x)}{\beta_x}\|\bar{v}\|\right)\frac{\bar{v}}{\|\bar{v}\|}.
\end{aligned}
\tag{93}
$$

The above equalities use:

(1) $1 - \tanh^2(t) = 1/\cosh^2(t)$;

(2) $\sinh(2t) = 2\sinh(t)\cosh(t)$;

(3) $\lambda_{\bar{x}}^K = \frac{1+\beta_x}{\beta_x}$.

### E.4.2 PV LOGARITHMIC MAP

We recall from Tab. 13 that the Riemannian logarithm on the Poincaré ball is

$$\mathrm{Log}_{\bar{x}}^{\mathbb{P}}(\bar{y}) = \frac{2}{\sqrt{-K}\lambda_{\bar{x}}^K} \frac{\tanh^{-1}\left(\sqrt{-K}\|\bar{z}\|\right)}{\|\bar{z}\|}\bar{z}, \qquad \bar{z} = (-\bar{x}) \oplus_{\mathrm{M}} \bar{y}, \tag{94}$$

where $\lambda_{\bar{x}}^K = \frac{2}{1+K\|\bar{x}\|^2}$. We define

$$z = (-x) \oplus_{\mathrm{U}} y, \quad \bar{z} = \pi(z). \tag{95}$$

By the Riemannian isometry of $\pi$, we have

$$
\begin{aligned}
\mathrm{Log}_x(y) &= d_{\bar{x}}(\pi^{-1})\left(\mathrm{Log}_{\bar{x}}^{\mathbb{P}}(\bar{y})\right) \\
&= d_{\bar{x}}(\pi^{-1})\left(\frac{2}{\sqrt{-K}\lambda_{\bar{x}}^K}\frac{\tanh^{-1}\left(\sqrt{-K}\|\bar{z}\|\right)}{\|\bar{z}\|}\bar{z}\right) \\
&= \alpha(x,y)d_{\bar{x}}(\pi^{-1})\left(\bar{z}\right),
\end{aligned}
\tag{96}
$$

where

$$
\alpha(x,y) = \frac{2}{\sqrt{-K}\lambda_{\bar{x}}^K}\frac{\tanh^{-1}\left(\sqrt{-K}\|\bar{z}\|\right)}{\|\bar{z}\|}.
\tag{97}
$$

The differential of $\pi^{-1}$ at $\bar{x} = \pi(x)$ is

$$
d_{\bar{x}}(\pi^{-1})(h) = -4K\gamma_{\bar{x}}^4\langle\bar{x},h\rangle\bar{x} + 2\gamma_{\bar{x}}^2 h, \quad \forall h \in T_{\bar{x}}\mathbb{P}_K^n,
\tag{98}
$$

where $\gamma_{\bar{x}} = \frac{1}{\sqrt{1+K\|\bar{x}\|^2}}$. Using $\bar{x} = \frac{\beta_x}{1+\beta_x}x$ and the relation $1 - K\|x\|^2 = \frac{1}{\beta_x^2}$, we have

$$
\begin{aligned}
K\|\bar{x}\|^2 &= K\frac{\beta_x^2}{(1+\beta_x)^2}\|x\|^2 = \frac{\beta_x^2}{(1+\beta_x)^2}\left(\frac{\beta_x^2-1}{\beta_x^2}\right) = \frac{\beta_x^2-1}{(1+\beta_x)^2}, \\
1 + K\|\bar{x}\|^2 &= 1 + \frac{\beta_x^2-1}{(1+\beta_x)^2} = \frac{2\beta_x}{1+\beta_x}.
\end{aligned}
\tag{99}
$$

Hence

$$
\gamma_{\bar{x}}^2 = \frac{1}{1+K\|\bar{x}\|^2} = \frac{1+\beta_x}{2\beta_x}, \qquad \gamma_{\bar{x}}^4 = \left(\frac{1+\beta_x}{2\beta_x}\right)^2 = \frac{(1+\beta_x)^2}{4\beta_x^2}.
\tag{100}
$$

Substituting these into $d_{\bar{x}}(\pi^{-1})(h)$ yields

$$
\begin{aligned}
d_{\bar{x}}(\pi^{-1})(h) &= -4K\frac{(1+\beta_x)^2}{4\beta_x^2}\langle\bar{x},h\rangle\bar{x} + 2\frac{1+\beta_x}{2\beta_x}h \\
&= -K\frac{(1+\beta_x)^2}{\beta_x^2}\langle\bar{x},h\rangle\bar{x} + \frac{1+\beta_x}{\beta_x}h \\
&= \frac{1+\beta_x}{\beta_x}h - K\langle x,h\rangle x,
\end{aligned}
\tag{101}
$$

where the last equality uses that $\bar{x} = \frac{\beta_x}{1+\beta_x}x$. Applying Eq. (101) with $h = \bar{z}$ and using that $\bar{z} = \pi(z)$ is collinear with $z = (-x) \oplus_U y$, we obtain

$$
\begin{aligned}
\mathrm{Log}_x(y) &= \alpha(x,y)(d\pi_x)^{-1}(\bar{z}) \\
&= \alpha(x,y)\left(\frac{1+\beta_x}{\beta_x}\bar{z} - K\langle x,\bar{z}\rangle x\right).
\end{aligned}
\tag{102}
$$

Since $\bar{z} = \pi(z)$ and $\pi$ is given by Eq. (4), $z$ and $\bar{z}$ are collinear and

$$
\bar{z} = \rho z, \qquad \rho = \frac{\beta_z}{1+\beta_z},
\tag{103}
$$

which also implies $\langle x,\bar{z}\rangle = \rho\langle x,z\rangle$. Substituting these into Eq. (102) yields

$$
\begin{aligned}
\mathrm{Log}_x(y) &= \alpha(x,y)\left(\frac{1+\beta_x}{\beta_x}\rho z - K\rho\langle x,z\rangle x\right) \\
&= \underbrace{\alpha(x,y)\frac{1+\beta_x}{\beta_x}\rho}_{\sigma(x,y)}z + \underbrace{\left(-K\alpha(x,y)\rho\right)}_{\tau(x,y)}\langle x,z\rangle x.
\end{aligned}
\tag{104}
$$

Using the definition of $\alpha(x, y)$ in Eq. (97) together with $\lambda_{\bar{x}}^K = \frac{1+\beta_x}{\beta_x}$ and $\rho = \frac{\beta_z}{1+\beta_z}$, a straightforward simplification yields

$$\sigma(x, y) = \frac{2}{\sqrt{-K}} \frac{\tanh^{-1}\left(\sqrt{-K}\|\bar{z}\|\right)}{\|z\|}, \quad \tau(x, y) = \frac{2\beta_x}{1 + \beta_x} \frac{\sqrt{-K}\tanh^{-1}\left(\sqrt{-K}\|\bar{z}\|\right)}{\|z\|}. \quad (105)$$

Thus,

$$\mathrm{Log}_x(y) = \sigma(x, y)z + \tau(x, y)\langle x, z\rangle x. \quad (106)$$

### E.4.3 PV PARALLEL TRANSPORT

We recall from Tab. 13 that the parallel transport on the Poincaré ball is

$$\mathrm{PT}_{\bar{x}\to\bar{y}}^{\mathbb{P}}(w) = \frac{\lambda_{\bar{x}}^K}{\lambda_{\bar{y}}^K}\,\mathrm{gyr}_{\mathrm{M}}[\bar{y}, -\bar{x}](w), \quad \text{with } w \in T_{\bar{x}}\mathbb{P}_K^n. \quad (107)$$

We have

$$
\begin{aligned}
&\mathrm{PT}_{x\to y}(v)\\
&\overset{(1)}{=} d_{\bar{y}}(\pi^{-1})\left(\mathrm{PT}_{\bar{x}\to\bar{y}}^{\mathbb{P}}\left(d\pi_x(v)\right)\right)\\
&= d_{\bar{y}}(\pi^{-1})\left(\frac{\lambda_{\bar{x}}^K}{\lambda_{\bar{y}}^K}\,\mathrm{gyr}_{\mathrm{M}}[\bar{y}, -\bar{x}]\left(d\pi_x(v)\right)\right)\\
&\overset{(2)}{=} \frac{\lambda_{\bar{x}}^K}{\lambda_{\bar{y}}^K} d_{\bar{y}}(\pi^{-1})\left(\mathrm{gyr}_{\mathrm{M}}[\bar{y}, -\bar{x}]\left(d\pi_x(v)\right)\right)\\
&\overset{(3)}{=} \frac{\lambda_{\bar{x}}^K}{\lambda_{\bar{y}}^K}\left(\frac{1+\beta_y}{\beta_y}\,\mathrm{gyr}_{\mathrm{M}}[\bar{y}, -\bar{x}]\left(d\pi_x(v)\right) - K\left\langle y, \mathrm{gyr}_{\mathrm{M}}[\bar{y}, -\bar{x}]\left(d\pi_x(v)\right)\right\rangle y\right)\\
&\overset{(4)}{=} \frac{(1+\beta_x)\beta_y}{(1+\beta_y)\beta_x}\left(\frac{1+\beta_y}{\beta_y}\,\mathrm{gyr}_{\mathrm{M}}[\bar{y}, -\bar{x}]\left(d\pi_x(v)\right) - K\left\langle y, \mathrm{gyr}_{\mathrm{M}}[\bar{y}, -\bar{x}]\left(d\pi_x(v)\right)\right\rangle y\right)\\
&= \frac{1+\beta_x}{\beta_x}\,\mathrm{gyr}_{\mathrm{M}}[\bar{y}, -\bar{x}]\left(d\pi_x(v)\right) - K\frac{(1+\beta_x)\beta_y}{(1+\beta_y)\beta_x}\left\langle y, \mathrm{gyr}_{\mathrm{M}}[\bar{y}, -\bar{x}]\left(d\pi_x(v)\right)\right\rangle y.
\end{aligned}
\quad (108)
$$

The above equalities use:

(1) the isometry property of $\pi$;

(2) linearity of $d_{\bar{y}}(\pi^{-1})$;

(3) Eq. (101).

(4) Using the relation between $\lambda_{\bar{x}}^K$ and $\beta_x$ in the proof of Thm. 4.2,

$$\lambda_{\bar{x}}^K = \frac{1+\beta_x}{\beta_x} \quad \lambda_{\bar{y}}^K = \frac{1+\beta_y}{\beta_y}. \quad (109)$$

### E.4.4 PV GEODESIC DISTANCE

We recall from Tab. 13 that the geodesic distance on the Poincaré ball $\mathbb{P}_K^n$ is

$$= \mathrm{d}^{\mathbb{P}}(y_1, y_2) = \frac{2}{\sqrt{-K}}\tanh^{-1}\left(\sqrt{-K}\|(-y_1)\oplus_{\mathrm{M}} y_2\|\right), \quad y_1, y_2 \in \mathbb{P}_K^n. \quad (110)$$

By isometry and isomorphism, the PV geodesic distance is

$$
\begin{aligned}
\mathrm{d}(x, y) &= \mathrm{d}^{\mathbb{P}}\left(\pi(x), \pi(y)\right)\\
&= \frac{2}{\sqrt{-K}}\tanh^{-1}\left(\sqrt{-K}\|(-\pi(x))\oplus_{\mathrm{M}} \pi(y)\|\right)\\
&= \frac{2}{\sqrt{-K}}\tanh^{-1}\left(\sqrt{-K}\|\pi(-x\oplus_{\mathrm{U}} y)\|\right),
\end{aligned}
\quad (111)
$$

### E.4.5  SPECIAL CASES AT THE IDENTITY

**Exponential map at the identity.**

$$\text{Exp}_{\mathbf{0}}(v) \overset{(1)}{=} \frac{1}{\sqrt{-K}} \sinh\left(\frac{\sqrt{-K}(1+\beta_x)}{\beta_x}\|\bar{v}\|\right) \frac{\bar{v}}{\|\bar{v}\|}$$

$$\overset{(2)}{=} \frac{1}{\sqrt{-K}} \sinh\left(\sqrt{-K}\|v\|\right) \frac{v}{\|v\|}. \tag{112}$$

The above comes from the following.

    (1) $\mathbf{0}$ is the gyro identity;

    (2) $\beta_{\mathbf{0}} = 1$ and $d\pi_{\mathbf{0}}(v) = \frac{1}{2}v$.

**Logarithmic map at the identity.** As $z = -\mathbf{0} \oplus_U y = y$, we have

$$\text{Log}_{\mathbf{0}}(y) = \sigma(\mathbf{0}, y)z + \tau(\mathbf{0}, y)\langle \mathbf{0}, z\rangle \mathbf{0}$$

$$= \sigma(\mathbf{0}, y)y$$

$$= \frac{2}{\sqrt{-K}} \frac{\tanh^{-1}\left(\sqrt{-K}\|\pi(y)\|\right)}{\|y\|} y. \tag{113}$$

Using $\pi(y) = \frac{\beta_y}{1+\beta_y}y$, we obtain From $\pi(y) = \frac{\beta_y}{1+\beta_y}y$ and $\beta_y = \frac{1}{\sqrt{1-K\|y\|^2}}$, we obtain

$$\sqrt{-K}\|\pi(y)\| = \frac{\beta_y}{1+\beta_y}\sqrt{-K}\|y\|. \tag{114}$$

Let $t = \sqrt{-K}\|y\|$ and $s = \sqrt{1+t^2}$, so that $\beta_y = \frac{1}{\sqrt{1-K\|y\|^2}} = \frac{1}{s}$. Define

$$a = \frac{\beta_y}{1+\beta_y}t = \frac{t}{s+1}. \tag{115}$$

Using the hyperbolic double-angle identity, we have

$$\tanh\left(2\tanh^{-1}(a)\right) = \frac{2a}{1+a^2}$$

$$= \frac{2t/(s+1)}{1+t^2/(s+1)^2}$$

$$= \frac{2t(s+1)}{(s+1)^2 + t^2}$$

$$= \frac{2t(s+1)}{s^2 + 2s + 1 + t^2}$$

$$= \frac{2t(s+1)}{2(1+t^2+s)}$$

$$= \frac{t(s+1)}{1+t^2+s}$$

$$= \frac{t(s+1)}{s^2+s}$$

$$= \frac{t}{s} = \frac{t}{\sqrt{1+t^2}}. \tag{116}$$

Denoting $u = \sinh^{-1}(t)$, we have

$$\cosh(u) = \sqrt{1+\sinh^2(u)} = \sqrt{1+t^2}. \tag{117}$$

Therefore,

$$\tanh\left(\sinh^{-1}(t)\right) = \tanh(u) = \frac{\sinh(u)}{\cosh(u)} = \frac{t}{\sqrt{1+t^2}}. \tag{118}$$

Since $\tanh$ is strictly increasing on $\mathbb{R}$, this implies that

$$2\tanh^{-1}\left(\frac{\beta_y}{1+\beta_y}t\right) = \sinh^{-1}(t). \tag{119}$$

Substituting this identity back gives

$$\sigma(\mathbf{0}, y) = \frac{1}{\sqrt{-K}}\frac{\sinh^{-1}\left(\sqrt{-K}\|y\|\right)}{\|y\|}, \tag{120}$$

and therefore

$$\text{Log}_{\mathbf{0}}(y) = \frac{1}{\sqrt{-K}}\sinh^{-1}\left(\sqrt{-K}\|y\|\right)\frac{y}{\|y\|}. \tag{121}$$

**Parallel transport from the identity.** The gyration satisfies $\text{gyr}_{\text{M}}[\mathbf{0}, \bar{y}] = \text{gyr}_{\text{M}}[\bar{y}, \mathbf{0}] = \mathbb{I}$. Substituting this into Eq. (108) gives

$$\begin{aligned}
\text{PT}_{\mathbf{0}\to y}(v) &= \frac{1+\beta_{\mathbf{0}}}{\beta_{\mathbf{0}}}d\pi_{\mathbf{0}}(v) - K\frac{(1+\beta_{\mathbf{0}})\beta_y}{(1+\beta_y)\beta_{\mathbf{0}}}\langle y, d\pi_{\mathbf{0}}(v)\rangle y \\
&= 2\cdot\frac{1}{2}v - K\frac{2\beta_y}{1+\beta_y}\cdot\frac{1}{2}\langle y, v\rangle y \\
&= v - K\frac{\beta_y}{1+\beta_y}\langle y, v\rangle y.
\end{aligned} \tag{122}$$

**Parallel transport to the identity.** Taking $y = \mathbf{0}$ in Eq. (108) and using $\text{gyr}_{\text{M}}[\mathbf{0}, -\bar{x}] = \mathbb{I}$ yields

$$\begin{aligned}
\text{PT}_{x\to\mathbf{0}}(v) &= \frac{1+\beta_x}{\beta_x}d\pi_x(v) \\
&= \frac{1+\beta_x}{\beta_x}\left(K\frac{\beta_x^3}{(1+\beta_x)^2}\langle x, v\rangle x + \frac{\beta_x}{1+\beta_x}v\right) \\
&= v + K\frac{\beta_x^2}{1+\beta_x}\langle x, v\rangle x.
\end{aligned} \tag{123}$$

**Distance from the identity.** This can be directly obtained by gyro identity.

$$\text{d}(\mathbf{0}, y) = \frac{2}{\sqrt{-K}}\tanh^{-1}\left(\sqrt{-K}\|\pi(y)\|\right). \tag{124}$$

Using the same identity as above with $t = \sqrt{-K}\|y\|$ yields

$$\text{d}(\mathbf{0}, y) = \frac{1}{\sqrt{-K}}\sinh^{-1}\left(\sqrt{-K}\|y\|\right). \tag{125}$$

### E.5 PROOF OF THM. 4.4

*Proof.* As shown by (Chen et al., 2025c, Prop. 21), the Möbius gyroaddition and gyromultiplication can be written by the Riemannian operators. Besides, the isometry $\pi_{\mathbb{PV}_K^n\to\mathbb{P}_K^n}$ preserves the identity: $\pi_{\mathbb{PV}_K^n\to\mathbb{P}_K^n}(\mathbf{0}) = \mathbf{0}$. By Nguyen & Yang (2023, Lems 2.1-2.2), one can directly obtain the results. □

### E.6 PROOF OF THM. 5.1

We first establish the PV hyperplane equivalence and then derive the distance formula.

### E.6.1 EQUIVALENT CHARACTERIZATION OF THE PV HYPERPLANE

We first review a useful lemma from Chen et al. (2025b, Lem. J.1).

**Lemma E.1.** *We assume that the manifold $\mathcal{M}$ admits a gyrogroup defined by*

$$x \oplus y = \mathrm{Exp}_x \left( \mathrm{PT}_{e \to x} \left( \mathrm{Log}_e \left( y \right) \right) \right), \forall x, y \in \mathcal{M}. \tag{126}$$

*where $e \in \mathcal{M}$ is the origin of the manifold. Then, we have the following*

$$\left\langle \mathrm{Log}_p(x), a \right\rangle_p = \left\langle \mathrm{Log}_e(\ominus p \oplus x), \mathrm{PT}_{p \to e}(a) \right\rangle_e, \quad \forall x, p \in \mathcal{M} \text{ and } \forall a \in T_p\mathcal{M}. \tag{127}$$

Now, we are ready to prove Thm. 5.1.

*Proof of PV hyperplane.* Thm. 4.4 indicates that the assumption of Lem. E.1 holds with $\mathcal{M} = \mathbb{PV}_K^n$, $\oplus = \oplus_{\mathrm{U}}$ and $e = \mathbf{0}$. Then, the PV hyperplane

$$H_{a,p} = \left\{ x \in \mathbb{PV}_K^n \mid \left\langle \mathrm{Log}_p(x), a \right\rangle_p = 0 \right\} \tag{128}$$

can be rewritten as

$$H_{a,p} = \left\{ x \in \mathbb{PV}_K^n \mid \left\langle \mathrm{Log}_{\mathbf{0}}(-p \oplus_{\mathrm{U}} x), \mathrm{PT}_{p \to \mathbf{0}}(a) \right\rangle_{\mathbf{0}} = 0 \right\}. \tag{129}$$

Using the explicit PV operators in Thm. 4.3 and the PV metric in Eq. (1), we have

$$\begin{aligned} \mathrm{Log}_{\mathbf{0}}(-p \oplus_{\mathrm{U}} x) &= \alpha(-p \oplus_{\mathrm{U}} x), \quad \text{for some scalar } \alpha \geq 0, \\ \mathrm{PT}_{p \to \mathbf{0}}(a) &= \beta d\pi_p(a), \quad \text{for some scalar } \beta > 0, \\ g_{\mathbf{0}}(u, v) &= \langle u, v \rangle. \end{aligned} \tag{130}$$

As $\alpha = 0$ is trivial, we only consider the case $\alpha > 0$:

$$\left\langle \mathrm{Log}_{\mathbf{0}}(-p \oplus_{\mathrm{U}} x), \mathrm{PT}_{p \to \mathbf{0}}(a) \right\rangle_{\mathbf{0}} = 0 \quad \Longleftrightarrow \quad \left\langle -p \oplus_{\mathrm{U}} x, d_p \pi(a) \right\rangle = 0. \tag{131}$$

$\square$

### E.6.2 PV POINT-TO-HYPERPLANE DISTANCE

We first prove a lemma on the isometry and point-to-hyperplane distance, which will be used to derive the PV point-to-hyperplane distance.

**Lemma E.2** (Isometry and point-to-hyperplane distance)**.** *Let $(\mathcal{M}, g)$ and $(\bar{\mathcal{M}}, \bar{g})$ be Riemannian manifolds and let $\phi : \mathcal{M} \to \bar{\mathcal{M}}$ be a Riemannian isometry. For $p \in \mathcal{M}$ and $a \in T_p\mathcal{M}$, define the hyperplane*

$$H_{a,p} = \left\{ x \in \mathcal{M} \mid g_p \left( \mathrm{Log}_p(x), a \right) = 0 \right\}. \tag{132}$$

*Let $\bar{p} = \phi(p)$ and $\bar{a} = d_p\phi(a) \in T_{\bar{p}}\bar{\mathcal{M}}$, and define the corresponding hyperplane on $\bar{\mathcal{M}}$ by*

$$\bar{H}_{\bar{a},\bar{p}} = \left\{ \bar{x} \in \bar{\mathcal{M}} \mid \bar{g}_{\bar{p}} \left( \bar{\mathrm{Log}}_{\bar{p}}(\bar{x}), \bar{a} \right) = 0 \right\}. \tag{133}$$

*Then $\phi$ maps $H_{a,p}$ onto $\bar{H}_{\bar{a},\bar{p}}$, that is,*

$$\phi \left( H_{a,p} \right) = \bar{H}_{\bar{a},\bar{p}}. \tag{134}$$

*Moreover, for every $x \in \mathcal{M}$ we have*

$$\mathrm{d}_{\mathcal{M}} \left( x, H_{a,p} \right) = \mathrm{d}_{\bar{\mathcal{M}}} \left( \phi(x), \bar{H}_{\bar{a},\bar{p}} \right), \tag{135}$$

*when the point-to-hyperplane distance exists. Here, $\mathrm{d}_{\mathcal{M}}$ and $\mathrm{d}_{\bar{\mathcal{M}}}$ denote the Riemannian distances on $\mathcal{M}$ and $\bar{\mathcal{M}}$, respectively.*

*Proof.* Since $\phi$ is a Riemannian isometry, we have

$$g_p\left(\mathrm{Log}_p(x), a\right) = \bar{g}_{\bar{p}}\left(d_p\phi\left(\mathrm{Log}_p(x)\right), d_p\phi(a)\right) = \bar{g}_{\bar{p}}\left(\bar{\mathrm{Log}}_{\bar{p}}\left(\phi(x)\right), \bar{a}\right). \tag{136}$$

Therefore,

$$g_p\left(\mathrm{Log}_p(x), a\right) = 0 \quad \Longleftrightarrow \quad \bar{g}_{\bar{p}}\left(\bar{\mathrm{Log}}_{\bar{p}}\left(\phi(x)\right), \bar{a}\right) = 0, \tag{137}$$

which shows that $x \in H_{a,p}$ if and only if $\phi(x) \in \bar{H}_{\bar{a},\bar{p}}$, and hence $\phi\left(H_{a,p}\right) = \bar{H}_{\bar{a},\bar{p}}$.

For the point-to-hyperplane distance, recall that for a subset $S \subset \mathcal{M}$ the distance from $x$ to $S$ is

$$\mathrm{d}_{\mathcal{M}}(x, S) = \inf_{z \in S} \mathrm{d}_{\mathcal{M}}(x, z). \tag{138}$$

For the point-to-hyperplane distance, we have

$$\begin{aligned}
\mathrm{d}_{\mathcal{M}}\left(x, H_{a,p}\right) &= \inf_{z \in H_{a,p}} \mathrm{d}_{\mathcal{M}}(x, z) \\
&= \inf_{z \in H_{a,p}} \mathrm{d}_{\bar{\mathcal{M}}}\left(\phi(x), \phi(z)\right) \\
&= \inf_{\bar{z} \in \bar{H}_{\bar{a},\bar{p}}} \mathrm{d}_{\bar{\mathcal{M}}}\left(\phi(x), \bar{z}\right) \\
&= \mathrm{d}_{\bar{\mathcal{M}}}\left(\phi(x), \bar{H}_{\bar{a},\bar{p}}\right).
\end{aligned} \tag{139}$$

$\square$

Next, we review the Poincaré hyperplane and point-to-hyperplane distance (Ganea et al., 2018, Sec. 3.1).

**Poincaré point-to-hyperplane distance.** For a point $p \in \mathbb{P}_K^n$ and a normal vector $a \in T_p\mathbb{P}_K^n$, the Poincaré point-to-hyperplane distance is given by Ganea et al. (2018, Thm. 5):

$$H_{a,p}^{\mathbb{P}} = \left\{ x \in \mathbb{P}_K^n \mid \left\langle \mathrm{Log}_p^{\mathbb{P}}(x), a \right\rangle_p = 0 \right\} = \left\{ x \in \mathbb{P}_K^n \mid \langle -p \oplus_{\mathrm{M}} x, a \rangle = 0 \right\}, \tag{140}$$

$$d^{\mathbb{P}}(y, H_{a,p}^{\mathbb{P}}) = \frac{1}{\sqrt{-K}} \sinh^{-1}\left( \frac{2\sqrt{-K}\,|\langle -p \oplus_{\mathrm{M}} y, a \rangle|}{\left(1 + K\|-p \oplus_{\mathrm{M}} y\|^2\right)\|a\|} \right). \tag{141}$$

*Proof of the PV point-to-hyperplane distance.* Let

$$\bar{p} = \pi(p) \in \mathbb{P}_K^n, \quad \bar{a} = d_p\pi(a) \in T_{\bar{p}}\mathbb{P}_K^n, \quad \bar{y} = \pi(y) \in \mathbb{P}_K^n. \tag{142}$$

By Lem. E.2, the point-to-hyperplane distances satisfy

$$\mathrm{d}^{\mathbb{PV}}\left(y, H_{a,p}\right) = \mathrm{d}^{\mathbb{P}}\left(\bar{y}, \bar{H}_{\bar{a},\bar{p}}\right). \tag{143}$$

Applying the Poincaré distance formula (141) with $p = \bar{p}$, $a = \bar{a}$, and $y = \bar{y}$ gives

$$\mathrm{d}^{\mathbb{P}}\left(\bar{y}, \bar{H}_{\bar{a},\bar{p}}\right) = \frac{1}{\sqrt{-K}} \sinh^{-1}\left( \frac{2\sqrt{-K}\,|\langle -\bar{p} \oplus_{\mathrm{M}} \bar{y}, \bar{a} \rangle|}{\left(1 + K\|-\bar{p} \oplus_{\mathrm{M}} \bar{y}\|^2\right)\|\bar{a}\|} \right). \tag{144}$$

The gyrovector isomorphism $\pi$ implies

$$-\bar{p} \oplus_{\mathrm{M}} \bar{y} = \pi(-p \oplus_{\mathrm{U}} y). \tag{145}$$

Denote $z = -p \oplus_{\mathrm{U}} y$. From Sec. 4 we have the explicit expression

$$\pi(z) = \frac{\beta_z}{1 + \beta_z} z \tag{146}$$

with $\beta_z > 0$. Since $\beta_z = \frac{1}{\sqrt{1-K\|z\|^2}}$, we obtain

$$
\begin{aligned}
1 + K\big\|\pi(z)\big\|^2 &= 1 + K\left(\frac{\beta_z}{1+\beta_z}\right)^2 \|z\|^2 \\
&= 1 + \frac{K\beta_z^2\|z\|^2}{(1+\beta_z)^2} \\
&= 1 + \frac{\beta_z^2(1 - \beta_z^{-2})}{(1+\beta_z)^2} \\
&= 1 + \frac{\beta_z^2 - 1}{(1+\beta_z)^2} \\
&= 1 + \frac{\beta_z - 1}{1+\beta_z} \\
&= \frac{2\beta_z}{1+\beta_z}
\end{aligned}
\tag{147}
$$

The above yields

$$
\frac{2\sqrt{-K}\left|\big\langle \pi(z), \bar{a}\big\rangle\right|}{\left(1 + K\big\|\pi(z)\big\|^2\right)\|\bar{a}\|} = \frac{\sqrt{-K}\left|\langle z, \bar{a}\rangle\right|}{\|\bar{a}\|}.
\tag{148}
$$

Therefore,

$$
d\big(y, H_{a,p}\big) = \frac{1}{\sqrt{-K}} \sinh^{-1}\left(\frac{\sqrt{-K}\left|\big\langle -p \oplus_{\mathrm{U}} y, d_p\pi(a)\big\rangle\right|}{\|d_p\pi(a)\|}\right).
\tag{149}
$$

$\square$

### E.7 Proof of Thm. 5.2

*Proof of PV MLR.* For clarity, we fix a class index $k$ and omit $k$ in the notation whenever possible. We denote $\pi = \pi_{\mathbb{PV}_K^n \to \mathbb{P}_K^n}$ as in Thm. 5.1.

**Step 1: From hyperplane distance to a signed score.** The PV MLR in Eq. (18) associated with parameters $(p, a)$ for $x \in \mathbb{PV}_K^n$ is

$$
\begin{aligned}
&v_k(x) \\
&= \mathrm{sign}\left(\big\langle -p_k \oplus_{\mathrm{U}} x, d_{p_k}\pi(a_k)\big\rangle\right) \|a_k\|_{p_k} d\big(x, H_{a_k,p_k}\big) \\
&\overset{(1)}{=} \frac{\|a_k\|_{p_k}}{\sqrt{-K}} \mathrm{sign}\left(\big\langle -p_k \oplus_{\mathrm{U}} x, d_{p_k}\pi(a_k)\big\rangle\right) \sinh^{-1}\left(\frac{\sqrt{-K}\left|\big\langle -p_k \oplus_{\mathrm{U}} x, d_{p_k}\pi(a_k)\big\rangle\right|}{\|d_{p_k}\pi(a_k)\|}\right) \\
&\overset{(2)}{=} \frac{\|a_k\|_{p_k}}{\sqrt{-K}} \sinh^{-1}\left(\frac{\sqrt{-K}\big\langle -p_k \oplus_{\mathrm{U}} x, d_{p_k}\pi(a_k)\big\rangle}{\|d_{p_k}\pi(a_k)\|}\right).
\end{aligned}
\tag{150}
$$

The above comes from the following.

(1) Thm. 5.1;

(2) $\sinh^{-1}$ is odd and strictly increasing.

**Step 2: Trivialization and reduction to a single direction.** We adopt the unidirectional parameterization in Sec. 5.1:

$$
p_k = \mathrm{Exp}_{\mathbf{0}}\big(r_k[z_k]\big), \qquad a_k = \mathrm{PT}_{\mathbf{0} \to p_k}(z_k), \qquad [z_k] = \frac{z_k}{\|z_k\|},
\tag{151}
$$

with $z_k \in T_{\mathbf{0}}\mathbb{PV}_K^n \cong \mathbb{R}^n$ and $r_k \in \mathbb{R}$.

As parallel transport is an isometry, we have

$$\|a_k\|_{p_k} = \|z_k\|_{\mathbf{0}} = \|z_k\|. \tag{152}$$

Moreover, $p_k$ and $z_k$ are collinear, because $\text{Exp}_{\mathbf{0}}$ in Thm. 4.3 preserves directions at the origin. Using the explicit expression of $\text{PT}_{\mathbf{0} \to y}$ at the origin in Thm. 4.3, we see that $\text{PT}_{\mathbf{0} \to p_k}$ maps $z_k$ to a linear combination of $z_k$ and $p_k$. Therefore, $a_k$ is also collinear with $z_k$.

The differential $d_{p_k}\pi$ in Lem. 4.1 has the form

$$d_{p_k}\pi(v) = \alpha_k v + \beta_k \langle p_k, v \rangle p_k, \quad \alpha_k > 0, \ \beta_k \in \mathbb{R}, \tag{153}$$

so $d_{p_k}\pi$ maps any vector in $\text{span}\{z_k\}$ into the same one-dimensional subspace. Consequently, there exists a scalar $\lambda_k > 0$ such that

$$d_{p_k}\pi(a_k) = \lambda_k z_k. \tag{154}$$

The sign of $\lambda_k$ can be absorbed into $z_k$ by redefining $z_k \leftarrow -z_k$ if necessary. Without loss of generality we may assume $\lambda_k > 0$. Putting Eq. (151), Eq. (152), Eq. (154) and $\|d_{p_k}\pi(a_k)\| = \lambda_k\|z_k\|$ into Eq. (150) yields

$$v_k(x) = \frac{\|z_k\|}{\sqrt{-K}} \sinh^{-1}\left( \frac{\sqrt{-K}}{\|z_k\|} \langle -p_k \oplus_{\mathrm{U}} x, z_k \rangle \right). \tag{155}$$

**Step 3: Eliminating the gyroaddition.** The remaining task is to expand the gyro-additive term in Eq. (155). From Sec. 3, PV gyroaddition is given by

$$u \oplus_{\mathrm{U}} v = u + v + \left( \frac{1 - \beta_v}{\beta_v} - K \frac{\beta_u}{1 + \beta_u} \langle u, v \rangle \right) u, \qquad \beta_w = \frac{1}{\sqrt{1 - K\|w\|^2}}. \tag{156}$$

Setting $u = -p_k$ and $v = x$ yields

$$-p_k \oplus_{\mathrm{U}} x = -p_k + x + \left( \frac{1 - \beta_x}{\beta_x} - K \frac{\beta_{p_k}}{1 + \beta_{p_k}} \langle -p_k, x \rangle \right)(-p_k). \tag{157}$$

Taking the inner product with $z_k$ gives

$$\begin{aligned}
\langle -p_k \oplus_{\mathrm{U}} x, z_k \rangle &= \langle -p_k, z_k \rangle + \langle x, z_k \rangle + \left( \frac{1 - \beta_x}{\beta_x} - K \frac{\beta_{p_k}}{1 + \beta_{p_k}} \langle -p_k, x \rangle \right) \langle -p_k, z_k \rangle \\
&= \langle x, z_k \rangle + \left( 1 + \frac{1 - \beta_x}{\beta_x} - K \frac{\beta_{p_k}}{1 + \beta_{p_k}} \langle -p_k, x \rangle \right) \langle -p_k, z_k \rangle.
\end{aligned} \tag{158}$$

Next, we rewrite the above expression using the unidirectional parameterization of $p_k$. From Eq. (151) and the explicit PV exponential at the origin in Thm. 4.3, we have

$$p_k = \text{Exp}_{\mathbf{0}}\left( r_k[z_k] \right) = \frac{1}{\sqrt{-K}} \sinh\left( \sqrt{-K} r_k \right) \frac{z_k}{\|z_k\|}. \tag{159}$$

Thus,

$$\langle -p_k, z_k \rangle = -\frac{1}{\sqrt{-K}} \sinh\left( \sqrt{-K} r_k \right) \|z_k\|. \tag{160}$$

Moreover, since $p_k$ and $z_k$ share the same direction, any $x$ admits the decomposition

$$x = x_\| + x_\perp, \qquad x_\| = \frac{\langle x, z_k \rangle}{\|z_k\|^2} z_k, \qquad \langle x_\perp, z_k \rangle = 0, \tag{161}$$

which implies

$$\langle -p_k, x \rangle = \left\langle -p_k, x_\| \right\rangle = \frac{\langle x, z_k \rangle}{\|z_k\|^2} \langle -p_k, z_k \rangle = -\frac{1}{\sqrt{-K}} \sinh\left( \sqrt{-K} r_k \right) \frac{\langle x, z_k \rangle}{\|z_k\|}. \tag{162}$$

The beta factor at $p_k$ is

$$\beta_{p_k} = \frac{1}{\sqrt{1 - K\|p_k\|^2}} = \text{sech}\left( \sqrt{-K} r_k \right), \tag{163}$$

where we used $\|p_k\|^2 = -\frac{1}{K}\sinh^2\left(\sqrt{-K}r_k\right)$ and the identity $1 + \sinh^2(t) = \cosh^2(t)$.

Using $\langle -p_k, z_k \rangle$, $\langle -p_k, x \rangle$, and $\beta_{p_k}$, we have

$$
\begin{aligned}
&\langle -p_k \oplus_{\mathrm{U}} x, z_k \rangle \\
&= \langle x, z_k \rangle + \left( 1 + \frac{1 - \beta_x}{\beta_x} - K\frac{\beta_{p_k}}{1 + \beta_{p_k}} \langle -p_k, x \rangle \right) \langle -p_k, z_k \rangle \\
&= \langle x, z_k \rangle + \left( \frac{1}{\beta_x} - K\frac{\beta_{p_k}}{1 + \beta_{p_k}} \langle -p_k, x \rangle \right) \langle -p_k, z_k \rangle \\
&= \langle x, z_k \rangle + \frac{1}{\beta_x} \left( -\frac{\sinh\left(\sqrt{-K}r_k\right)}{\sqrt{-K}} \|z_k\| \right) \\
&\quad - K\frac{\beta_{p_k}}{1 + \beta_{p_k}} \left( -\frac{\sinh\left(\sqrt{-K}r_k\right)}{\sqrt{-K}} \frac{\langle x, z_k \rangle}{\|z_k\|} \right) \left( -\frac{\sinh\left(\sqrt{-K}r_k\right)}{\sqrt{-K}} \|z_k\| \right) \\
&= \langle x, z_k \rangle - \frac{\sinh\left(\sqrt{-K}r_k\right)}{\sqrt{-K}} \frac{\|z_k\|}{\beta_x} + \frac{\beta_{p_k}\sinh^2\left(\sqrt{-K}r_k\right)}{1 + \beta_{p_k}} \langle x, z_k \rangle \\
&= \left( 1 + \frac{\beta_{p_k}\sinh^2\left(\sqrt{-K}r_k\right)}{1 + \beta_{p_k}} \right) \langle x, z_k \rangle - \frac{\sinh\left(\sqrt{-K}r_k\right)}{\sqrt{-K}} \frac{\|z_k\|}{\beta_x}.
\end{aligned}
\tag{164}
$$

Since $\beta_{p_k} = \mathrm{sech}\left(\sqrt{-K}r_k\right)$ and $1 + \sinh^2\left(\sqrt{-K}r_k\right) = \cosh^2\left(\sqrt{-K}r_k\right) = 1/\beta_{p_k}^2$, we have

$$
\begin{aligned}
1 + \frac{\beta_{p_k}\sinh^2\left(\sqrt{-K}r_k\right)}{1 + \beta_{p_k}} &= \frac{1 + \beta_{p_k} + \beta_{p_k}\sinh^2\left(\sqrt{-K}r_k\right)}{1 + \beta_{p_k}} \\
&= \frac{1 + \beta_{p_k}\cosh^2\left(\sqrt{-K}r_k\right)}{1 + \beta_{p_k}} \\
&= \frac{1 + 1/\beta_{p_k}}{1 + \beta_{p_k}} = \frac{1}{\beta_{p_k}} = \cosh\left(\sqrt{-K}r_k\right),
\end{aligned}
\tag{165}
$$

which implies

$$
\langle -p_k \oplus_{\mathrm{U}} x, z_k \rangle = \cosh\left(\sqrt{-K}r_k\right) \langle x, z_k \rangle - \frac{\sinh\left(\sqrt{-K}r_k\right)}{\sqrt{-K}} \frac{\|z_k\|}{\beta_x}.
\tag{166}
$$

Recalling that $\beta_x = 1/\sqrt{1 - K\|x\|^2}$, we obtain

$$
\langle -p_k \oplus_{\mathrm{U}} x, z_k \rangle = \cosh\left(\sqrt{-K}r_k\right) \langle x, z_k \rangle - \frac{\sinh\left(\sqrt{-K}r_k\right)}{\sqrt{-K}} \|z_k\|\sqrt{1 - K\|x\|^2}.
\tag{167}
$$

Substituting Eq. (167) into Eq. (155), we arrive at

$$
v_k(x) = \frac{\|z_k\|}{\sqrt{-K}} \sinh^{-1}\left( \cosh\left(\sqrt{-K}r_k\right) \frac{\sqrt{-K}}{\|z_k\|} \langle x, z_k \rangle - \sinh\left(\sqrt{-K}r_k\right)\sqrt{1 - K\|x\|^2} \right).
\tag{168}
$$

$\square$

*Proof of PV MLR limits.* By Taylor expansions, we have

$$\cosh\left(\sqrt{-K}r_k\right) = 1 - \frac{Kr_k^2}{2} + \mathcal{O}(K^2),$$
$$\sinh\left(\sqrt{-K}r_k\right) = \sqrt{-K}r_k + \mathcal{O}\left((-K)^{3/2}\right) \tag{169}$$
$$\sqrt{1 - K\|x\|^2} = 1 - \frac{K\|x\|^2}{2} + \mathcal{O}(K^2).$$

The argument of $\sinh^{-1}(\cdot)$ in Eq. (168) can be simplified as

$$\sinh^{-1}\left\{\cosh\left(\sqrt{-K}r_k\right)\frac{\sqrt{-K}}{\|z_k\|}\langle x, z_k\rangle - \sinh\left(\sqrt{-K}r_k\right)\sqrt{1 - K\|x\|^2}\right\}$$

$$= \sinh^{-1}\left\{\left(1 - \frac{Kr_k^2}{2} + \mathcal{O}(K^2)\right)\frac{\sqrt{-K}}{\|z_k\|}\langle x, z_k\rangle\right.$$

$$\left. - \left(\sqrt{-K}r_k + \mathcal{O}\left((-K)^{3/2}\right)\right)\left(1 - \frac{K\|x\|^2}{2} + \mathcal{O}(K^2)\right)\right\} \tag{170}$$

$$= \sinh^{-1}\left\{\sqrt{-K}\left(\frac{\langle x, z_k\rangle}{\|z_k\|} - r_k\right) + \mathcal{O}\left((-K)^{3/2}\right)\right\}$$

$$= \sqrt{-K}\left(\frac{\langle x, z_k\rangle}{\|z_k\|} - r_k\right) + \mathcal{O}\left((-K)^{3/2}\right).$$

Substituting this into Eq. (168) gives

$$v_k(x) = \frac{\|z_k\|}{\sqrt{-K}}\left(\sqrt{-K}\left(\frac{\langle x, z_k\rangle}{\|z_k\|} - r_k\right) + \mathcal{O}\left((-K)^{3/2}\right)\right)$$

$$= \|z_k\|\left(\frac{\langle x, z_k\rangle}{\|z_k\|} - r_k\right) + \mathcal{O}(-K) \tag{171}$$

$$= \langle x, z_k\rangle - r_k\|z_k\| + \mathcal{O}(-K),$$

$$\xrightarrow{K\to 0^-} \langle x, z_k\rangle - r_k\|z_k\|.$$

$\square$

### E.8 PROOF OF THM. 5.3

*Proof of PV FC layer.* Specializing Thm. 5.1 to $p = \mathbf{0}$ and $a = e_k$ and using that $-\mathbf{0} \oplus_U y = y$ gives the LHS

$$\text{sign}\left(\langle d_{\mathbf{0}_k}\pi(e_k), -\mathbf{0} \oplus_U x\rangle\right)d\left(y, H_{e_k,\mathbf{0}}\right) = \frac{1}{\sqrt{-K}}\sinh^{-1}\left(\sqrt{-K}y_k\right), \tag{172}$$

with $y_k = \langle y, e_k\rangle$. Then, we obtain

$$y_k = \frac{1}{\sqrt{-K}}\sinh\left(\sqrt{-K}v_k(x)\right), \quad k = 1, \ldots, m. \tag{173}$$

$\square$

*Proof of PV FC limits.* By Thm. 5.2, as $K \to 0^-$ we have

$$v_k(x) \to \langle x, z_k\rangle + b_k, \quad b_k = -r_k\|z_k\|, \tag{174}$$

For $K < 0$ and $v_k(x) \neq 0$, we can rewrite $y_k$ as

$$y_k = v_k(x)\frac{\sinh\left(\sqrt{-K}v_k(x)\right)}{\sqrt{-K}v_k(x)}, \tag{175}$$

and we define the fraction to be 1 when $v_k(x) = 0$. Since $\sqrt{-K} \to 0$ and $v_k(x)$ converges to a finite limit, we have $\sqrt{-K}v_k(x) \to 0$. Using the standard limit $\lim_{u \to 0} \sinh(u)/u = 1$, it follows that

$$\frac{\sinh\left(\sqrt{-K}v_k(x)\right)}{\sqrt{-K}v_k(x)} \to 1 \quad \text{as} \quad K \to 0^-. \tag{176}$$

Combining the above limits yields

$$\lim_{K \to 0^-} y_k = \lim_{K \to 0^-} v_k(x) = \langle x, z_k \rangle + b_k. \tag{177}$$

$\square$

### E.9 PROOF OF THM. 5.4

*Proof.* The result is first established in the Poincaré ball model (Chen et al., 2025c, Thms. 14 and 16). Since $\pi : \mathbb{PV}to\mathbb{P}$ is a Riemannian isometry, it intertwines the key geometric operators used in the proof:

$$d\pi_x \circ \operatorname{Exp}_x^{\mathbb{PV}} = \operatorname{Exp}_{\pi(x)}^{\mathbb{P}} \circ d\pi_x, \qquad d\pi_x \circ \operatorname{Log}_x^{\mathbb{PV}} = \operatorname{Log}_{\pi(x)}^{\mathbb{P}} \circ d\pi_x, \tag{178}$$

$$\pi\left(x \oplus_t^{\mathbb{PV}} y\right) = \pi(x) \oplus_t^{\mathbb{P}} \pi(y). \tag{179}$$

Hence the homogeneity identities, written purely in terms of Exp, Log and the gyroaddition $\oplus_t$, are preserved under $\pi$. Therefore the same theorem holds for the PV model (Chen et al., 2025c, Lem. 11). $\square$

### E.10 PROOF OF PROP. D.1

*Proof.* We first recall the isometries between the Poincaré ball and the hyperboloid (Skopek et al., 2020, Sec. 2.1):

$$\pi_{\mathbb{H}_K^n \to \mathbb{P}_K^n}(x) = \frac{x_s}{1 + \sqrt{|K|}x_t}, \tag{180}$$

$$\pi_{\mathbb{P}_K^n \to \mathbb{H}_K^n}(y) = \begin{bmatrix} \dfrac{1}{\sqrt{|K|}} \dfrac{1 - K\|y\|^2}{1 + K\|y\|^2} \\ \dfrac{2y}{1 + K\|y\|^2} \end{bmatrix}. \tag{181}$$

Hence, the following are Riemannian isometries:

$$\pi_{\mathbb{H}_K^n \to \mathbb{PV}_K^n} = \pi_{\mathbb{P}_K^n \to \mathbb{PV}_K^n} \circ \pi_{\mathbb{H}_K^n \to \mathbb{P}_K^n}, \qquad \pi_{\mathbb{PV}_K^n \to \mathbb{H}_K^n} = \pi_{\mathbb{P}_K^n \to \mathbb{H}_K^n} \circ \pi_{\mathbb{PV}_K^n \to \mathbb{P}_K^n}. \tag{182}$$

It remains to derive the explicit formulas.

For $x = [x_t, x_s^\top]^\top \in \mathbb{H}_K^n$, we first map to the Poincaré ball:

$$y = \pi_{\mathbb{H}_K^n \to \mathbb{P}_K^n}(x) = \frac{x_s}{1 + \sqrt{|K|}x_t}. \tag{183}$$

Applying $\pi_{\mathbb{P}_K^n \to \mathbb{PV}_K^n}$ from Eq. (4) yields

$$\pi_{\mathbb{H}_K^n \to \mathbb{PV}_K^n}(x) = \pi_{\mathbb{P}_K^n \to \mathbb{PV}_K^n}(y) = 2\gamma_y^2 y, \qquad \gamma_y = \frac{1}{\sqrt{1 + K\|y\|^2}}. \tag{184}$$

Using $y = x_s/(1 + \sqrt{|K|}x_t)$, we compute

$$1 + K\|y\|^2 = 1 + K\frac{\|x_s\|^2}{\left(1 + \sqrt{|K|}x_t\right)^2} = \frac{\left(1 + \sqrt{|K|}x_t\right)^2 + K\|x_s\|^2}{\left(1 + \sqrt{|K|}x_t\right)^2}. \tag{185}$$

Since $x \in \mathbb{H}_K^n$ satisfies $\langle x, x \rangle_{\mathcal{L}} = 1/K$, we have

$$\langle x, x \rangle_{\mathcal{L}} = -x_t^2 + \|x_s\|^2 = \frac{1}{K} \quad \Rightarrow \quad \|x_s\|^2 = x_t^2 + \frac{1}{K}. \tag{186}$$

Substituting this into the numerator gives

$$\left(1 + \sqrt{|K|}x_t\right)^2 + K\|x_s\|^2 = \left(1 + \sqrt{|K|}x_t\right)^2 + K\left(x_t^2 + \frac{1}{K}\right)$$
$$= \left(1 + \sqrt{|K|}x_t\right)^2 + Kx_t^2 + 1 \tag{187}$$
$$= 1 + 2\sqrt{|K|}x_t + |K|x_t^2 + Kx_t^2 + 1$$
$$= 2\left(1 + \sqrt{|K|}x_t\right).$$

Therefore,

$$1 + K\|y\|^2 = \frac{2\left(1 + \sqrt{|K|}x_t\right)}{\left(1 + \sqrt{|K|}x_t\right)^2} = \frac{2}{1 + \sqrt{|K|}x_t}, \tag{188}$$

and hence

$$\gamma_y^2 = \frac{1}{1 + K\|y\|^2} = \frac{1 + \sqrt{|K|}x_t}{2}. \tag{189}$$

Finally,

$$\pi_{\mathbb{H}_K^n \to \mathbb{PV}_K^n}(x) = 2\gamma_y^2 y = 2 \cdot \frac{1 + \sqrt{|K|}x_t}{2} \cdot \frac{x_s}{1 + \sqrt{|K|}x_t} = x_s. \tag{190}$$

For $\pi_{\mathbb{PV}_K^n \to \mathbb{H}_K^n}$, take $x \in \mathbb{PV}_K^n$ and map to the Poincaré ball by Eq. (4):

$$y = \pi_{\mathbb{PV}_K^n \to \mathbb{P}_K^n}(x) = \frac{\beta_x}{1 + \beta_x}x, \qquad \beta_x = \frac{1}{\sqrt{1 - K\|x\|^2}}. \tag{191}$$

Applying $\pi_{\mathbb{P}_K^n \to \mathbb{H}_K^n}$, we obtain

$$\pi_{\mathbb{PV}_K^n \to \mathbb{H}_K^n}(x) = \pi_{\mathbb{P}_K^n \to \mathbb{H}_K^n}(y) = \begin{bmatrix} \frac{1}{\sqrt{|K|}} \frac{1 - K\|y\|^2}{1 + K\|y\|^2} \\ \frac{2y}{1 + K\|y\|^2} \end{bmatrix}. \tag{192}$$

We now simplify the spatial and temporal components separately. We write

$$y = \frac{\beta_x}{1 + \beta_x}x, \qquad \|y\|^2 = \left(\frac{\beta_x}{1 + \beta_x}\right)^2 \|x\|^2. \tag{193}$$

Using $\beta_x^2 = 1/(1 - K\|x\|^2)$, we obtain

$$K\|y\|^2 = K\|x\|^2 \frac{\beta_x^2}{(1 + \beta_x)^2} = \frac{\beta_x^2 - 1}{(1 + \beta_x)^2} = \frac{\beta_x - 1}{(1 + \beta_x)}$$
$$\Rightarrow 1 + K\|y\|^2 = \frac{2\beta_x}{1 + \beta_x}, \quad 1 - K\|y\|^2 = \frac{2}{1 + \beta_x}. \tag{194}$$

The spatial component of $\pi_{\mathbb{PV}_K^n \to \mathbb{H}_K^n}(x)$ is

$$\frac{2y}{1 + K\|y\|^2} = \frac{2\frac{\beta_x}{1 + \beta_x}x}{\frac{2\beta_x}{1 + \beta_x}} = x, \tag{195}$$

and the temporal component is

$$\frac{1}{\sqrt{|K|}} \frac{1 - K\|y\|^2}{1 + K\|y\|^2} = \frac{1}{\sqrt{|K|}} \cdot \frac{\frac{2}{1 + \beta_x}}{\frac{2\beta_x}{1 + \beta_x}} = \frac{1}{\sqrt{|K|}\beta_x} = \frac{\sqrt{1 - K\|x\|^2}}{\sqrt{|K|}} = \sqrt{\|x\|^2 - \frac{1}{K}}. \tag{196}$$

Thus,

$$\pi_{\mathbb{PV}_K^n \to \mathbb{H}_K^n}(x) = \begin{bmatrix} \sqrt{\|x\|^2 - \frac{1}{K}} \\ x \end{bmatrix}. \tag{197}$$

$\square$

