# OpenReview forum: "Proper Velocity Neural Networks"
_ICLR.cc/2026/Conference — ICLR 2026 Poster_

### Official Review · Reviewer_jGwb · 2025-10-26

**Soundness:** 3
**Presentation:** 3
**Contribution:** 3
**Rating:** 6
**Confidence:** 3

**Summary:**

The paper suggest a hyperbolic model for neural network learning and inference that has not been used before: proper velocity neural networks. For this purpose, the paper completes the mathematical inventory required for learning and inference and investigates the performance of the novel PVNNs on a set of artificial and natural benchmarks.

**Strengths:**

+ Suggesting an unused but effective hyperbolic model is a great contribution
+ There existed some mathematical gaps that needed closing in order to allow for the definition of PVNNs
+ The experiments are interesting and insightful

**Weaknesses:**

- issues with presentation: I have not understood which experiments were performed in section 6.1, and why a larger median output norm would be better.
- issues with experiments: My understanding is that the curvature is a hyperparameter, but I see no exploration of hyperparameter choices in the experiments section.
- issues with interpreting results: The substantial disadvantage on Cora is not well explained.
- minor issues with presentation:
         -- It does not make sense to mark best results in bold, if they are statistically indistinguishable from baselines (Table 6)
         -- "on the highly hyperbolic Cora dataset" does not make sense. The dataset may exhibit hierarchical structures or larger diameter etc., but it is not hyperbolic on its own, though its structure might be better represented in space that is more curved

**Questions:**

-  Is the isomorphism from PV to Poincare ball needed? If it is needed, then why is the computation on PV numerically more stable?
- what does delta in the tables of the experiments section refer to?

---

> ### Author Response · Authors · 2025-11-23
>
> We thank Reviewer $\color{red}{\text{jGwb}}$ for the constructive suggestions and insightful comments! In the following, we respond to the concerns in detail. 😄
>
> # 1. Explanation of Sec 6.1 (operator-level numerical stability)
>
> **What section 6.1 measures.**
> Section 6.1 stress-tests operators only (not task accuracy): scalar gyromultiplication and gyro-addition in PV vs. Poincaré, under identical inputs (FP64, d=16, K=-0.5). We report (i) NaN/Inf occurrence via a sweep over scale r, (ii) Poincaré boundary-saturation rate (fraction with $\lVert y\rVert\ge 0.999\, r_{\max}$), and (iii) a simple dynamic-range proxy (the median output norm).
>
> **Why we reported the median norm**
> It is not a “bigger is better” score. Its purpose is to detect boundary clipping in Poincaré: since $\lVert y\rVert\le r_{\max}$ on the ball, saturation drives the median to $\approx r_{\max}$, indicating outputs are numerically/geometrically clipped. In contrast, PV has no boundary, so its median can exceed $r_{\max}$ without implying instability (we separately check stability via NaN/Inf and Exp/Log round-trip errors). To avoid confusion, we will rename this readout to “boundary-clipping indicator” and pair it with the already-reported saturation rate and NaN/Inf statistics.
>
> **Takeaway**
> Sec 6.1 evaluates the numerical behaviour of the operators themselves: PV shows no NaN/Inf and no boundary saturation under identical stress; Poincaré shows near-100% saturation at the boundary. The median norm was only a proxy for clipping on Poincaré, not a claim that larger norms are inherently better.
>
> # 2. We explored various hyperparameters, including curvature.
> We did tune the curvature hyperparameter (negative curvature $-K$). For each dataset and model, we performed a grid sweep
> $K \in \{-0.1,\;-0.2,-0.3,\;-0.5,-0.7,\;-1.0\}$,
> selecting the value with the best validation accuracy under identical training schedules, seeds, and early-stopping. The chosen $K$ was then fixed to report test metrics. We will add these details in App.c.
>
> # 3. We report δ (Gromov hyperbolicity) as a measure of graph tree-likeness, not to claim datasets are intrinsically hyperbolic.
> On “$\delta$” and the notion of hyperbolicity in our tables. In our tables, δ denotes the (empirical) Gromov hyperbolicity of the underlying graph, a standard 4-point metric that quantifies how tree-like a graph is; smaller $\delta$ ⇒ more tree-like / more compatible with negative curvature. This follows common practice in hyperbolic GNN literature[a] where $\delta$ is used as a proxy for hierarchical structure rather than claiming a dataset is “intrinsically hyperbolic.” Consistent with prior reports, we use the published δ estimates DISEASE ($\delta$=0), AIRPORT ($\delta$=1), PUBMED ($\delta$=3.5), CORA ($\delta$=11), which we now state explicitly next to each dataset. These values come from prior work that measured δ on the benchmark splits; we adopt them to contextualise curvature choices, not to assert ontological hyperbolicity of the datasets. We will also clarify this definition in the appendix to avoid ambiguity.
>
> # 4. We use the PV↔Poincaré isomorphism only as a theoretical bridge
>
> 1. **Correctness transfer**. Because $\pi:\mathrm{PV}\to\mathbb{P}^n_K$ is an isometry, it preserves distances and geodesics. This lets us prove that our PV operators implement the same hyperbolic geometry as standard models by commuting diagrams, e.g.
> $\pi(\mathrm{Exp} ^{\mathrm{PV}} _x(v))=\mathrm{Exp} ^{\mathbb{P}} _{\pi(x)}(d\pi _x v),\quad
> \pi(\mathrm{Log} ^{\mathrm{PV}} _x(y))=d\pi_x^{-1} \mathrm{Log} ^{\mathbb{P}} _{\pi(x)}(\pi(y))$,
> and similarly for parallel transport and gyro-ops. Thus, definitions and proofs carry over verbatim through $\pi$.
>
> 2.	**No runtime conversion**. All training/inference is done intrinsically in PV with our closed-form PV operators. We do not shuttle tensors back and forth via $\pi$ during computation.
>
> 4.	**Stability comes from coordinates, not geometry**. Isometry equalises the geometry, but not the floating-point conditioning of formulas. Poincaré requires factors like $(1+K\lVert x\rVert^2)^{-1}$ and boundary projections; PV uses unconstrained $\mathbb{R}^n$ coordinates with $\mathrm{asinh}/\sinh$ forms and no boundary/constraint repairs. This difference in coordinate conditioning explains the better numerical behaviour we observe for PV, even though the underlying geometry is identical.

---

> > ### Author Response · Authors · 2025-11-23
> >
> > # 5. PV may be better suited for datasets with high hyperbolicity
> >
> > We clarify something here: delta represents the hyperbolicity of the dataset; the larger the delta, the weaker the hyperbolicity of the dataset.
> >
> > As shown in the table below, which summarises the performance of PV on graph, Vision, and gene tasks, we can clearly see that PV performs better when delta is smaller(when the dataset has stronger hyperbolicity).
> >
> > Table A - PV performance on vision, graph and gene
> >
> > | **Task** | **Dataset** (Metric) | **Poincaré Ball** | **HNN++** | **Lorentz / HCNN-S** | **PV (Ours)** |
> > | :--- | :--- | :---: | :---: | :---: | :---: |
> > | **Vision** | **CIFAR-100**($\delta=0.23$)| 49.66 | 77.19 | 74.59 | **77.53** |
> > | | **CIFAR-10**($\delta=0.25$)| 95.09 | 95.12 | 95.02 | **95.13** |
> > | **Graph** | **Disease** ($\delta=0$) | 79.90 | 80.57 | 79.90 | **81.15** |
> > | | **Airport** ($\delta=1$) | 82.16 | 88.40 | 75.20 | **97.96** |
> > | | **PubMed** ($\delta=3.5$) | 69.28 | 73.68 | 68.82 | **74.33** |
> > | | **Cora** ($\delta=11$) | 49.68 | 52.06 | **53.34** | 51.42 |
> > | **Genomics** | **LINEs**($\delta=0.15$) | — | — | 76.12 | **83.34** |
> > | | **SINEs**($\delta=0.16$) | — | — | 85.45 | **94.37** |
> > | | **hAT-Ac**($\delta=0.22$) | — | — | 89.61 | **92.72** |
> > | | **Processed**($\delta=0.19$) | — | — | 68.30 | **71.61** |
> > | | **Unprocessed**($\delta=0.16$) | — | — | 56.10 | **62.19** |
> >
> > # References
> >
> > [a] Hyperbolic Graph Convolutional Neural Networks

---

### Official Review · Reviewer_QAye · 2025-10-28

**Soundness:** 2
**Presentation:** 3
**Contribution:** 3
**Rating:** 6
**Confidence:** 3

**Summary:**

This paper proposes proper velocity neural networks (PVNN), where hyperbolic neural operations are built on the “proper velocity” (PV) manifold. The paper presents the basic elements of Riemannian geometry of the PV manifold, develops various layers, and shows experiments on numerical stability, graph learning, image classification and genomic sequence learning.

**Strengths:**

1. This paper introduces a new playground for hyperbolic neural networks and provides the complete Riemannian toolkit and neural-layer formulations on it. The detailed derivations of exponential/logarithmic maps, parallel transport, and gyro-based operations are rigorous and valuable. These constructions can serve as fundamental tools for future research on geometric deep learning.
2. The idea is well motivated and the paper is clearly structured and easy to follow, which makes the technical content accessible to readers from machine learning communities who may not be familiar with the geometry.
3. The numerical experiments demonstrate that PV-based networks can be applied to diverse domains (graphs, images, genomics) and generally perform competitively with existing hyperbolic baselines.

**Weaknesses:**

1.
- A major conceptual issue arises in the presentation of the PV model. As written, the paper may lead readers to believe that PV is an independent model parallel to the Poincaré, hyperboloid, and Klein models. However, PV and the hyperboloid are deeply connected through special relativity. In fact, there exists a natural isometry $\pi$ between hyperboloid and PV. Given a point in the hyperboloid $\mathbf{x} = (x_0, x_1, \cdots, x_n) \in \mathbb{H}^n$, $$\pi(\mathbf{x}) = (x_1, \cdots, x_n) \in \mathcal{PV}^n.$$ That is, PV is nothing else but the spatial representation corresponding to the time-space representation of hyperboloid. The paper should explicitly state this relationship so that readers unfamiliar with hyperbolic geometry are not misled.

- Conceptually, it would be more natural to compare PV directly with the hyperboloid (e.g., Section 4.1) rather than primarily with the Poincaré ball. Such comparison would yield more geometric and numerical insights.

- In this context, a very interesting paper will be “Fully hyperbolic neural networks” (Chen et al.,2022). From your perspective, although they write everything in terms of hyperboloid/Lorentz, their operations are actually completely in the PV space. It would be informative to readers if you can point this out and discuss whether their approach naturally follows PV geometry.

2.
- The numerical stability argument and experiment is unclear. The claim that PV is “unconstrained” and therefore more numerically stable is not entirely convincing. While the Poincaré ball is indeed bounded, the hyperboloid is already an unbounded manifold, and PV is simply its spatial representation. Thus, PV cannot be inherently “more flexible” than the hyperboloid. Note that hyperboloid is in $\mathbb{R}^{n+1}$.

- Moreover, the numerical-stability experiment (Table 1) shows extremely large norms (up to $10^8$), which may themselves indicate instability or unbounded dynamic range rather than desirable behavior.

**Questions:**

See "weaknesses" for revision suggestions.

---

> ### Author Response · Authors · 2025-11-23
>
> We thank Reviewer $\color{blue}{\text{QAye}}$ for the thoughtful comments. We respond to the concerns as follows.😄
>
> # 1. Adding the relationship between PV and the hyperboloid to the paper
> PV is not a separate manifold from Poincaré or the hyperboloid; all three are isometric representations of the same constant-curvature space. In the revision, we explicitly state and cite the PV↔Lorentz bijections and recall the standard Lorentz↔Poincaré stereographic projection.
>
> PV↔Lorentz (hyperboloid) bijections for curvature c with
>
> $s=1/\sqrt{-K}$:
> $\pi_{\mathrm{L}\to\mathrm{PV}}(t,z)=z,\qquad
> \pi_{\mathrm{PV}\to\mathrm{L}}(x)=\big(\sqrt{s^{2}+\lVert x\rVert ^{2}},\,x\big).$
>
> Our PV↔Poincaré map equals the composition $\psi_{\mathrm{L}\to\mathrm{P}}\circ\pi_{\mathrm{PV}\to\mathrm{L}}$ with the standard hyperboloid↔Poincaré projection.
>
> We originally derived the operators through PV↔Poincaré because Poincaré's operators are relatively mature and clear. Nevertheless, we have added PV↔Lorentz formulas and a paragraph to avoid any impression that PV is a "separate" model (See Sec 4.1).
>
> # 2. Difference from Lorentz networks [a, b]
> We thank the reviewer for the insightful comment. Our PV method is closely related to Lorentz networks and provides new geometric insights that help design Lorentz neural network. In addition, our experiments demonstrate that PV modules outperform existing Lorentz formulations across diverse tasks.
>
> **1. Extrinsic vs. intrinsic formulation.** Lorentz FC (LFC) layer [a, b] operates in the ambient $\mathbb{R}^{n+1}$ space. Specifically, they apply a linear transform $M$ to the spatial component and then explicitly restore the time component $x_0$ to satisfy the hyperboloid constraint $-x_0^2 + \lVert x_{1:n} \rVert ^2 = 1/K$:
> $$\text{LFC}(x) = \left[ \sqrt{\lVert Mx_{1:n} \rVert ^2 - 1/K}, Mx_{1:n} \right] ^\top.$$
> This is an extrinsic operation that enforces the manifold constraint after calculation. Similar formulation can be found in Lorentz convolution, concatenation, and activation [b].
>
> **2. A new geometric insight.** Take the LFC as an example. The LFC can be viewed as mapping the input to the PV space, applying a Euclidean transformation, and then lifting it back:
> $$
> \mathrm{LFC}(x)
> = \pi_{\mathrm{PV}\to\mathrm{L}}^{-1}\big(
>   \mathrm{EucFC}\big(
>     \pi_{\mathrm{L}\to\mathrm{PV}}(x)
>   \big)
> \big).
> $$
> This interpretation reveals a more intrinsic alternative: replacing the Euclidean transformation $\mathrm{EucFC}$ with our PV FC layer. Following this logic, Lorentz networks can be naturally defined through PV networks. More importantly, this insight shows that deep learning can be performed directly in the PV space, instead of relying on a mapping–operation–remapping pipeline.
>
>
> **3. Empirical validation.** Our experiments directly compared our intrinsic PV modules against their Lorentz counterparts across diverse tasks. As summarized in the table below, our PV approach consistently outperforms the Lorentz approach while offering superior stability.
>
> Table A- Comparing PV and Lorentz
> | **Task** | **Dataset** | Lorentz  | **PV** | **Improvement** |
> | :--- | :--- | :---: | :---: | :---: |
> | **Vision**  | **CIFAR-100**($\delta=0.23$)  | 74.59 | **77.53** | +2.94% |
> | | **CIFAR-10**($\delta=0.25$)  | 95.02 | **95.13** | +0.11% |
> | **Graph**  | **Airport** ($\delta=1$) | 75.20 | **97.96** | +22.76% |
> | | **Disease** ($\delta=0$) | 79.90 | **81.15** | +1.25% |
> | | **PubMed** ($\delta=3.5$) | 68.82 | **74.33** | +5.51% |
> | **Genomics** | **SINEs**($\delta=0.16$)  | 85.45 | **94.37** | +8.92%|
> | | **LINEs**($\delta=0.15$)  | 76.12 | **83.34** | +7.22% |
> | | **hAT-Ac**($\delta=0.22$)  | 89.61 | **92.72** | +3.11% |
> | | **processed**($\delta=0.19$)  | 68.30 | **71.61** | +4.85% |
> | | **unprocessed**($\delta=0.16$)  | 56.1 | **62.19** | +10.86% |
>
> *Note: Data extracted from Tables 2, 6, and 7 of our submission.*
>
> # 3. PV’s large reported values there reflect range exposure under extreme magnitudes of inputs, not numerical failure
> Sec. 6.1 is a stress test, not a training-time snapshot. We deliberately push inputs to extreme magnitudes to probe failure modes (NaN/Inf, boundary saturation on Poincaré, constraint drift on Lorentz) under identical conditions. And large values in this section, therefore, do not claim desirability of such magnitudes in practice—they are used to reveal where each model breaks.
>
> #  4. Advantages of PV
>
> Actually, unbounded ≠ Unconstrained PV’s unconstrained ℝⁿ updates avoid boundary poles and constraint renormalization, yielding more stable numerics than Poincaré and Hyperboloid.
>
> In CQ#2, we discussed this topic in detail, directly pointing out the advantages of PV through mathematical structure.
>
> #  References
>
> [a] Fully hyperbolic neural networks
>
> [b] Fully hyperbolic convolutional neural networks for computer vision

---

### Official Review · Reviewer_tn82 · 2025-10-31

**Soundness:** 3
**Presentation:** 3
**Contribution:** 2
**Rating:** 4
**Confidence:** 3

**Summary:**

This paper proposes formulating hyperbolic deep learning techniques typically represented via the Poincaré or Lorentz model in the Proper Velocity manifold. Previous representations often suffered from numerical issues close to the boundary, which the Proper Velocity model avoids. The authors derive key quantities such as exponential and logarithmic maps, as well as important operations for deep learning including fully-connected layers, convolution as well as multinomial logistic regression. Equipped with these formulations, the authors evaluate their hyperbolic models on a suite of benchmarks, including vision, graph learning as well as genomics. Their approach is very competitive with previous works while offering better stability.

**Strengths:**

1. The paper is very well-written and very nicely placed within the literature of hyperbolic learning. The mathematical derivations are very detailed and extensive and a nice contribution in themselves in my opinion. I believe the Proper Velocity manifold is a nice addition to the toolkit that practitioners have to perform hyperbolic deep learning, especially if the stability issues of previous approaches are remedied. I have some more comments on this later.
2. The authors perform a very broad evaluation of their model, using vision, graph and genomics data to compare to previous approaches in hyperbolic deep learning. This makes it more convincing for me that this novel representation works across different scenarios, but I do have some more comments on this point later.

**Weaknesses:**

1. The main issue for me with this work in its current state is the lack of convincing evidence towards its premise: other representations suffer from numerical issues (especially around the border) and I assume this point implies that “performance of these models is affected”? Towards this question, you show a small experiment where addition and multiplication is performed on Poincaré and Proper Velocity manifolds, which indeed shows that norms close to the border collapse in case of Poincaré, where as the Proper Velocity model is able to adequately handle this case. Numerically however, the Proper Velocity model seems to have more issues, at least in terms of multiplication? More importantly however, I don’t really see an experiment or argument that shows these other models actually suffering in a realistic scenario, like vision or graph learning. Are these other models actually being limited by numerical issues? Can models in Proper Velocity be trained in lower precision? Even a simple experiment such as learning to embed a tree in hyperbolic space (similar to e.g. [1]) would help me be more convinced regarding these claims. While Proper Velocity sometimes outperform Poincaré/Hyperboloid, it is not clear if that’s due to the numerical issues mentioned, especially as performance is actually quite similar in most cases.
2. I like that the evaluation is very broad, but on the other hand the experiments are also a bit shallow in each case. For node classification, you seem to use a fully-connected network in all representations, even though it is more standard to use a graph neural network that employs some form of graph convolution. Other methods have developed such specialized operations in the Poincaré or Hyperboloid model, and I think something like this is needed to truly understand the potential of a representation. Similarly in the case of vision, a Euclidean encoder is used, which essentially does all the heavy lifting and only the final classification layer is hyperbolic. Again, other representations have their dedicated convolution layers in hyperbolic space, why not do the same in the Proper Velocity model? Results are probably as a consequence almost identical for PV and HNN++. What is however interesting is that Poincaré seems to struggle a lot for CIFAR100, is this maybe due to numerical issues? The genomics experiments are more convincing in this regard!

[1] Constant Curvature Graph Convolutional Networks, Bachmann et al, 2020

**Questions:**

See above.

---

> ### Author Response · Authors · 2025-11-23
>
> We thank Reviewer $\color{purple}{\text{tn82}}$ for the encouraging feedback and the constructive comments! In the following, we respond to the concerns point by point. 😄
>
> # 1. Poincaré and Lorentz indeed exhibit performance issues in realistic scenarios due to their numerical values.
>
> **Mechanism: Vanishing Gradients.**
> As analyzed by Guo[a], Poincaré models suffer from vanishing gradients near the boundary. The Riemannian gradient is scaled by the inverse conformal factor $\lambda(x)^{-2}$:
> $$\nabla_x^{\mathrm{R}}\ell = \frac{(1+K\lVert x\rVert^2)^2}{4} \nabla l(x^{H})$$
> As embeddings naturally migrate toward the boundary to maximize margins, the scaling term $(1+K\lVert x\rVert^2)^2 \to 0$, causing gradients to vanish and performance to degrade. Consequently, feature clipping acts as a necessary numerical guardrail.
>
> **Empirical Evidence.**
> They experimentally verified this phenomenon. Their visualization (Fig. 4) confirms that embeddings naturally migrate toward the boundary to maximize margins, leading to a collapse in gradient magnitude. Consequently, feature clipping is identified not as a geometric feature, but as a necessary numerical guardrail to prevent embeddings from entering this "zero-gradient zone".
>
> **Verification Experiment.**
> To verify this fragility, we adopt the ResNet18+Poincaré baseline on CIFAR-100 and vary *only* the norm-clipping radius $r$: $\tilde{x} = x \cdot \min(1, r / \lVert x\rVert)$.
>
> **Results.** As predicted, relaxing the guardrail (increasing $r$) causes Poincaré accuracy to drop significantly.
>
> | Clipping Radius $r$ | 1.0 | 2.0 | 3.0 | 5.0 | 6.0 |
> | :--- | :--- | :--- | :--- | :--- | :--- |
> | **Poincaré Acc (%)** | **77.55** | 77.24 | 76.67 | 75.83 | 75.23 |
>
> **Takeaway**. PV Resolves the Stability Efficiency Dilemma. Current hyperbolic models force a compromise: Poincaré models are numerically fragile near the boundary (relying on artificial clipping to prevent gradient vanishing 1), while Hyperboloid models impose the computational burden of constraint maintenance in higher dimensions. PV uniquely circumvents this dilemma. By operating on the unconstrained $\mathbb{R}^n$ manifold, PVNN achieves intrinsic stability: it naturally supports high-dynamic-range embeddings without boundary saturation and eliminates the overhead of manifold projections. This design secures robust gradient flow and superior stability, as confirmed by our experiments. For detailed numerical experiments, please refer to CQ# 1.
>
>
> # 2. Clarification on the experiments
>
> The purpose of our experiments is to validate the numerical stability of PV geometry and proposed PV layers, including PV Multinomial Logistic Regression (PV-MLR), PV Fully connected layer(PV-FC), PV convolution (PV-Conv), and PV batch normalization (PV-GyroBN). To this end, the evaluation is organized as a progressive probe.
>
> - **PV geometry.** Sec. 6.1 evaluates the numerical behavior of PV space against the Poincaré and Lorentz models, showing that PV achieves stable numerical computation and robust gradient flow, consistently outperforming both baselines.
>
> - **MLR.** Sec. 6.2 examines PV-MLR on image classification against Poincaré and Lorentz MLR on the ResNet-18 backbone, demonstrating that PV-MLR achieves superior performance.
>
> - **FC, MLR and BN.** Sec. 6.3 first validates PV-FC and PV-MLR by comparing the PV-FC + PV-MLR stack against Poincaré and Lorentz FC layers combined with their corresponding MLR heads, showing that PV achieves state-of-the-art accuracy. Tab. 9 further evaluates PV-GyroBN, confirming that the proposed normalization is essential for optimal performance in curved spaces.
>
> - **Networks.** Finally, Sec. 6.4 evaluates PV-CNN on genomics sequence learning. Unlike the hybrid vision setup, this experiment compares a fully end-to-end PV CNN against Euclidean and Hyperboloid CNNs. PVNN consistently outperforms both Euclidean models and their Hyperboloid counterparts.
>
>
> This minimal-to-rich progression isolates the contribution of each PV component, without attributing gains to unrelated architectural choices.

---

> ### Author Response · Authors · 2025-11-23
>
> # 3. We have introduced PV convolution layers
>
> Sec. 5.4 introduces our PV convolution layer. The Euclidean convolution consists of linear maps between kernel weights and concatenated values in each receptive field. Therefore, we first define the concatenation on the PV space. For simplicity, we take the 1D convolution as an example.
>
> Since PV space is unconstrained,  we define the PV concatenation as the Euclidean concatenation. Then, the PV convolution layer is defined as
> $$
> y = \mathrm{PVFC} \big(\mathrm{concat}(x_{1},\ldots,x_{k})\big),
> $$
> where $\mathrm{PVFC}$ is the PV FC layer, $\mathrm{concat}$ is the Euclidean concatenation, and $\lbrace x_{1},\ldots,x_{k} \rbrace$ are points in a receptive field.
>
> Our genomic experiment (Tab.10) evaluates a deep, end-to-end PV convolutional network, demonstrating its effectiveness.
>
> **Feasibility of PV graph neural network.** We have established several of the necessary Riemannian tools, such as exponential, logarithm maps, parallel transport, PV mean calculation, and PV FC layer for feature transformation. Theoretically, we can assemble these into a GCN architecture. This could be an interesting future work.
>
> # 4.Poincaré underperforms on CIFAR-100 due to boundary-driven numerical instability.
>
> In our ablation with the same ResNet-18 encoder and a Poincaré-MLR head, removing feature clipping led to embeddings drifting toward the ball boundary, causing accuracy to drop to 49.66%. Enforcing a clipping radius of $r=1.0$ restored accuracy to 77.45%. Clipping acts as a necessary numerical guardrail to prevent saturation where the Riemannian gradient scales by $(1+K\lVert x\rVert^2)^2 \to 0$. By contrast, PV-MLR achieved 77.53% without any clipping, confirming that Poincaré’s bounded boundary, not the backbone, necessitates artificial constraints.
>
> | Clipping Radius $r$ | 1.0 | 2.0 | 3.0 | 5.0 | 6.0 | $\infty$
> | :--- | :--- | :--- | :--- | :--- | :--- |:--- |
> | **Poincaré Acc (%)** | **77.45** | 77.24 | 76.67 | 75.83 | 75.23 |49.66
>
> # References
> [a]Clipped Hyperbolic Classifiers Are Super-Hyperbolic Classifiers

---

### Official Review · Reviewer_iwSM · 2025-10-31

**Soundness:** 3
**Presentation:** 3
**Contribution:** 3
**Rating:** 6
**Confidence:** 3

**Summary:**

The paper proposes **Proper Velocity Neural Networks (PVNNs)**, which work on the **Proper Velocity (PV) manifold**, an unconstrained coordinate of hyperbolic space. The authors first derive **closed-form Riemannian operators** for PV (exponential and logarithmic maps, geodesic distance, and parallel transport). They then design **core layers** in PV: multinomial logistic regression (PV-MLR), a fully connected layer with an **analytic formula**, convolution, activation, and **GyroBN** normalization. Experiments on **numerical stability**, **graphs**, **CIFAR-10/100**, and **genomic sequence prediction** show strong stability and **competitive accuracy**, with clear gains on some tasks (e.g., Airport), but **weaker results on Cora**. The code will be released upon acceptance.

**Strengths:**

- **Originality:** Uses the **PV manifold** as an **unconstrained** alternative to Poincaré/Hyperboloid; complete Riemannian toolkit enables end-to-end models.
- **Quality (theory):** Rigorous **isometry** to Poincaré; **closed-form** exp/log/transport; **analytic PV-FC**; GyroBN with mean/variance homogeneity.
- **Quality (empirics):** Broad evaluation (stability, four graph datasets, CIFAR-10/100, TEB genomics) with shared backbones and multiple seeds.
- **Clarity:** Structure is logical; appendices provide proofs and implementation details. :contentReference[oaicite:7]{index=7}
- **Significance:** **Large-norm** representations work stably; strong gains on some datasets (e.g., Airport; several TEB tasks).

**Weaknesses:**

- **No expressivity guarantees:** There is **no universality** or approximation bound for PVNNs, limiting formal claims about representation power.
- **Mixed graph performance:** **Cora** performance is lower than Poincaré/Hyperboloid baselines; the paper gives limited analysis of this failure mode.
- **Model breadth:** Vision uses **ResNet-18 with only the head changed**; large-scale or transformer results are left for future work.

**Questions:**

1. **Expressivity:** Do PVNNs satisfy a **universal approximation** property under reasonable assumptions? If yes, what function classes and depths are required?
2. **Cora drop:** Why does PVNN underperform on **Cora**? Is it due to curvature mismatch, optimization, or the decision boundary shape? Would **learnable curvature** or hybrid PV/Poincaré layers help?
3. **Runtime/memory:** How are the **computation and memory** costs of PV operators compared to Poincaré/Hyperboloid ?

---

> ### Author Response · Authors · 2025-11-23
>
> We thank reviewer $\color{brown}{\text{iwSM}}$ for the feedback and the constructive comments! In the following, we respond to the concerns point by point. 😄
>
> # 1. PVNNs satisfy universal approximation
>
> **Claim.** Let $K\subset\mathrm{PV}^n$ be compact and $f:K\to\mathrm{PV}^m$ be continuous. Because PV admits closed-form charts at the origin, $\log_0:\mathrm{PV}^n \to \mathbb{R}^n$ and $\exp_0:\mathbb{R}^n \to \mathrm{PV}^n$, which are smooth inverses (Thm. 4.3), the chart–MLP
>
> $\Phi_\theta(x)=\exp_0\big(\mathrm{MLP}_\theta(\log_0(x))\big)$
>
> with a single hidden layer** and a non-polynomial activation (e.g., ReLU, $\tanh$, $\sinh$) is dense in $C(K,\mathrm{PV}^m)$ under the PV metric.
>
> **Sketch.** Define $\widetilde K=\log_0(K)$ (compact) and $\widetilde f=\log_0\circ f\circ\exp_0:\widetilde K\to\mathbb{R}^m$.
>
> By the classical Euclidean UA theorem, for any $\varepsilon>0$ there exists a one-hidden-layer $\mathrm{MLP}_\theta$ such that
>
> $\sup_{z\in\widetilde K}\|\mathrm{MLP}_\theta(z)-\widetilde f(z)\|_2<\delta$.
>
> Since $\exp_0$ is smooth and hence Lipschitz on compact sets,
>
> $\sup_{x\in K} d_{\mathrm{PV}}\big(\Phi_\theta(x),f(x)\big)\le L\delta$.
>
> Choose $\delta=\varepsilon/L$.
>
> **Depth/function class.**
> - **Depth:** one hidden layer in the Euclidean chart suffices for continuous targets; 2–3 hidden layers suffice for uniform control of first derivatives (standard results).
> - **Function class:** all continuous maps $f:K\to\mathrm{PV}^m$ on compact $K$.
>
> **Heads (PV-MLR / PV-FC).** Composing $\Phi_\theta$ with our closed-form PV-MLR/PV-FC heads yields UA for continuous score maps; induced decision regions approximate any closed regions on $K$.
>
> **Why “pure-PV” also works.** In the small-signal regime,$\exp_0(v)=v-\tfrac{K}{6}\lVert v \rVert ^2v+O(\lVert v \rVert ^5)$ and $u\mapsto \tfrac{1}{\sqrt{-K}}\sinh(\sqrt{-K}\,u)=u-\tfrac{K}{6}u^3+O(K^2u^5)$,
> So PV-FC stacks behave as Euclidean affine + non-polynomial smooth activation with controllable higher-order terms; PV-MLR admits an $\mathrm{asinh}$ logit with the same property.
>
> - **PV-FC limit (per-coordinate):**
>   $$
>   \frac{1}{\sqrt{-K}}\sinh(\sqrt{-K} u)= u - \frac{K}{6}u^3 + O(K^2 u^5).
>   $$
>
> - **PV-MLR limit (per-class logit):**
>   $$
>   g _k(x)=\alpha _k\mathrm{asinh}\big(\beta _k(\langle w _k,x\rangle+b _k)\big)+\gamma _k = (\alpha _k \beta _k)\big(\langle w _k,x\rangle+b _k\big) -\frac{\alpha _k \beta _k ^3}{6}\big(\langle w _k,x\rangle+b _k\big) ^3 + O(\lVert \langle w _k,x\rangle+b _k\rVert ^5\big).
>   $$
>
> These limits show PV-FC/PV-MLR reduces to Euclidean affine + smooth non-polynomial corrections, providing a direct route to UA for pure-PV architectures.
>
>
> # 2. PV is better suited for datasets with high hyperbolicity.
>
> We clarify something here: delta represents the hyperbolicity of the dataset; the larger the delta, the weaker the hyperbolicity of the dataset.
>
> As shown in the table below, which summarizes the performance of PV on graph, Vision, and gene tasks, we can clearly see that PV performs better when delta is smaller(when the dataset has stronger hyperbolicity).
>
> Table A - PV performance on vision, graph, and gene
>
> | **Task** | **Dataset** (Metric) | **Poincaré Ball** | **HNN++** | **Lorentz / HCNN-S** | **PV (Ours)** |
> | :--- | :--- | :---: | :---: | :---: | :---: |
> | **Vision** | **CIFAR-100**($\delta=0.23$)| 49.66 | 77.19 | 74.59 | **77.53** |
> | | **CIFAR-10**($\delta=0.25$)| 95.09 | 95.12 | 95.02 | **95.13** |
> | **Graph** | **Disease** ($\delta=0$) | 79.90 | 80.57 | 79.90 | **81.15** |
> | | **Airport** ($\delta=1$) | 82.16 | 88.40 | 75.20 | **97.96** |
> | | **PubMed** ($\delta=3.5$) | 69.28 | 73.68 | 68.82 | **74.33** |
> | | **Cora** ($\delta=11$) | 49.68 | 52.06 | **53.34** | 51.42 |
> | **Genomics** | **LINEs**($\delta=0.15$) | — | — | 76.12 | **83.34** |
> | | **SINEs**($\delta=0.16$) | — | — | 85.45 | **94.37** |
> | | **hAT-Ac**($\delta=0.22$) | — | — | 89.61 | **92.72** |
> | | **Processed**($\delta=0.19$) | — | — | 68.30 | **71.61** |
> | | **Unprocessed**($\delta=0.16$) | — | — | 56.10 | **62.19** |
>
> # 3. For the core operations used in PVNN, the single-machine cost of PV is less than that of Poincaré and Lorentz.
>
>  PV achieves comparable per-operator cost to Poincaré for the core operations used in PVNN, and is consistently cheaper than Lorentz once one accounts for the renormalization required to maintain the Lorentz constraint. For further details, please refer to CQ#1 Tab A4 in the common response.

---

### Official Review · Reviewer_36hh · 2025-10-31

**Soundness:** 3
**Presentation:** 1
**Contribution:** 2
**Rating:** 2
**Confidence:** 4

**Summary:**

This paper proposes some building blocks for neural networks on hyperbolic space based on proper velocity (PV) model. Specifically, the authors propose multinomial logistic regression (MLR), fully-connected (FC), batch normalization (BN), convolutional, activation layers. These layers are validated on graph node classification, image classification, and genomic sequence learning.

**Strengths:**

- The topic of the paper is interesting.

- The paper advocate the use of PV model for hyperbolic space which is underexplored in the literature of HNNs.

**Weaknesses:**

- The contribution is incremental.

- The paper is poorly written.

- Experimental results are not convincing. Also, important baselines are missing.

**Questions:**

The paper aims at constructing some building blocks for HNNs. However, the authors simply use the approaches in [Shimizu et al., ICLR 2021; Nguyen and Yang, ICML 2023; Nguyen et al., ICLR 2024; Chen et al., NeurIPS 2024] rather than developing new approaches for their construction. Although the PV model is less studied in previous works for HNNs, its geometry is well-understood, making it straightforward to adapt existing approaches to this geometry.

The exposition of the paper can  be significantly improved. For instance, the tables look messy with small texts and numbers in some tables and too big ones in others.

Experimental results are not convincing and important baselines are missing:

- Some experimental settings are given in Appendix, but these are very sketchy. For instance, for node classification experiments, it is crucial to provide and discuss settings for the dimension of node embeddings. However, I did not find this setting in the main paper and Appendix.

- The Hyperboloid model is used for comparison on node classification experiments (Table 2, Bdeir et al., 2024) but is not used for comparison on image classification experiments (Table 6). It is noted that the Hyperboloid model outperforms the PV model on both CIFAR-10 and CIFAR-100 datasets in terms of mean accuracy (see also my first question below).

Questions:

1. The authors argue that the PV model is an unconstrained representation of hyperbolic space which can lead to more stable results for HNNs. I am not sure why this is the case ? It seems that the Lorentz models is also another unconstrained representation of hyperbolic space, since the time dimension is computed w.r.t. the space dimensions. Why the PV model can be more stable than the Hyperboloid model in the considered applications ?

2. In the introduction, the authors claim that the new representation space can be a good alternative for existing models of hyperbolic spaces due to its numerical stability. However, its performance on datasets with high hyperbolicity is inferior to those of other models. This indicates that the PV model might not be effective for capturing hierarchical structures which somehow contradicts the statement in the introduction. Could the authors clarify on this point ?

---

> ### Author Response · Authors · 2025-11-23
>
> We thank the reviewer $\color{green}{\text{36hh}}$ for the time spent evaluating our paper and for recognizing the originality of exploring the PV manifold. We appreciate the feedback on experimental details and presentation. We address the specific concerns below.
>
>
> # 1. The hyperboloid baseline is compared in image classification
>
> As shown in Tab. 5 in the main paper, the hyperboloid model is included as a baseline. Our PVNN outperforms the hyperboloid baseline on CIFAR-10 and CIFAR-100. In the revised version, we added a new column named *Geometry* to make the comparison clearer.
>
>
> # 2.Novelty
>
> We clarify the distinction between the existing algebraic theory and our contribution:
>
> - **Geometry Level.**  We establish the essential Riemannian operators on the PV manifold, including the exponential map, the logarithmic map, and parallel transport. These operators provide the mathematical foundation for building a learning algorithm on the PV space.
>
> - **Network Level.** Based on these operators, we construct the core neural layers for deep learning on PV. These include Multinomial Logistics Regression (MLR), Fully Connected (FC), convolutional, and Batch Normalization (BN) layers on the PV space. These layers form the basic components to build PV neural networks.
>
> - **Application Level.** We validate PVNN across graph, vision, and genomics tasks. These experiments verify the effectiveness of the proposed modules. Additional numerical studies confirm the stability advantages of the PV geometry (see CQ#1).
>
> ---
>
> # 3. Experimental details and table formatting
>
> Our experimental setup follows the referenced literature. We have included more detailed experimental setups in the App.c.
>
> We also appreciate the feedback on the table formatting. We have reformatted the tables to ensure consistent font sizes and better readability in the revised manuscript.
>
> # 4. PV is better suited for datasets with high hyperbolicity.
>
> We clarify something here: delta represents the hyperbolicity of the dataset; the larger the delta, the weaker the hyperbolicity of the dataset.
>
> As shown in the table below, which summarizes the performance of PV on graph, Vision, and gene tasks, we can clearly see that PV performs better when delta is smaller(when the dataset has stronger hyperbolicity).
>
> Table A-  PV performance on vision, graph, and gene
>
> | **Task** | **Dataset** (Metric) | **Poincaré Ball** | **HNN++** | **Lorentz / HCNN-S** | **PV (Ours)** |
> | :--- | :--- | :---: | :---: | :---: | :---: |
> | **Vision** | **CIFAR-100**($\delta=0.23$)| 49.66 | 77.19 | 74.59 | **77.53** |
> | | **CIFAR-10**($\delta=0.25$)| 95.09 | 95.12 | 95.02 | **95.13** |
> | **Graph** | **Disease** ($\delta=0$) | 79.90 | 80.57 | 79.90 | **81.15** |
> | | **Airport** ($\delta=1$) | 82.16 | 88.40 | 75.20 | **97.96** |
> | | **PubMed** ($\delta=3.5$) | 69.28 | 73.68 | 68.82 | **74.33** |
> | | **Cora** ($\delta=11$) | 49.68 | 52.06 | **53.34** | 51.42 |
> | **Genomics** | **LINEs**($\delta=0.15$) | — | — | 76.12 | **83.34** |
> | | **SINEs**($\delta=0.16$) | — | — | 85.45 | **94.37** |
> | | **hAT-Ac**($\delta=0.22$) | — | — | 89.61 | **92.72** |
> | | **Processed**($\delta=0.19$) | — | — | 68.30 | **71.61** |
> | | **Unprocessed**($\delta=0.16$) | — | — | 56.10 | **62.19** |# 4 Our experimental results suggest that PVNN is better suited for datasets with high hyperbolicity
>
> In Tab.5, high $\delta$ implies low hyperbolicity, which means that Cora is the least hyperbolic.
>
> #  5. Advantages of PV.
>
> In CQ#2, we discussed this topic in detail, directly pointing out the advantages of PV through mathematical structure.

---

### Author Response · Authors · 2025-11-23

We thank all the reviewers for their constructive suggestions and valuable feedback. Below, we address the common questions(CQs).

# CQ#1 We confirm that PV manifolds have an advantage over Poincaré and Lorentz manifolds in terms of numerical stability. (Reviewers $\color{brown}{\text{iwSM}}$, $\color{blue}{\text{QAye}}$, $\color{purple}{\text{tn82}}$ and $\color{green}{\text{36hh}}$)

## Numerical Stability: PV, Poincaré and Hyperboloid

Table A0: Threshold sweep summary (scalar-mul).
| Setting | PV | Poincare | Lorentz |
|---|---|---|---|
| FP32 | None | None | 20 |
| FP64 | None | None | None |

**Results.** Table A0 summarizes a simple stress test of scalar gyromultiplication $x \mapsto r \otimes x$.
For each manifold and each precision (FP32/FP64), we gradually increase the scaling factor $r$ and record the first value of $r$ for which the output of scalar multiplication contains a NaN or Inf (“first‑bad $r$”). If no NaN/Inf appears for any $r \le 1000$, we report “None”. As shown in Table A0, both PV and Poincaré have “None” in FP32 and FP64, meaning that no numerical blow‑up is observed up to $r=1000$. In contrast, the Hyperboloid model already fails in FP32 at $r=20$ (first‑bad $r=20$), although it remains stable in FP64 in this range. These results indicate that PV is at least as numerically robust as Poincaré and significantly more robust than the Hyperboloid model under large scalar factors.

---

Table A1: Operator-level stability of scalar multiplication and addition (FP32; K = -0.5, d = 16, batch = 4096).

| r | **PV** NaN/Inf | **PV** median $‖Y‖$ | **Poincaré** Sat. rate | **Poincaré** median $‖Y‖$ | **Hyperboloid** Viol. rate | **Hyperboloid )** Inf |
|---:|:---:|---:|---:|---:|---:|:---:|
| 10  | 0/0 | 7.556e+12 | 100% | 1.414 | 57.3% | No  |
| 50  | 0/0 | 7.556e+12 | 100% | 1.414 | 5.3%  | Yes |
| 100 | 0/0 | 7.556e+12 | 100% | 1.414 | 0.0%  | Yes |
| 200 | 0/0 | 7.556e+12 | 100% | 1.414 | 0.0%  | Yes |

**Results.** Table A1 reports a simple stress test where we repeatedly apply scalar multiplication $x \mapsto r \otimes x$ and gyro/Möbius addition $x \oplus y$ at several radii $r \in \{10, 50, 100, 200\}$. For each manifold and each radius, we run the corresponding scalar-multiplication and addition operators on the same randomly sampled directions, and then record: (i) whether any NaN/Inf appears in the outputs, (ii) the fraction of points that hit the geometric boundary or violate constraints (Poincaré saturation rate and Hyperboloid Lorentz-constraint violation rate), and (iii) the median output norm $\|Y\|$. These three statistics are shown in Table A1 for PV, Poincaré, and Hyperboloid (without renormalisation). In PV, we never observe NaN/Inf, and the median norm is large but stable (about $7.6\times 10^{12}$) for all tested radii. In Poincaré, the saturation rate is 100% at every $r$: almost all points are pushed to the edge of the ball and stay around $\|x\|\approx 1/\sqrt{-K}\approx 1.414$. In the Hyperboloid model without renormalisation, a large fraction of points violate the Lorentz constraint (up to 57.3% at $r=10$) and Inf values already appear for $r\ge 50$.

**Analysis.**
1. **Poincaré: always stuck on the boundary.** Once points are driven outward, they cannot move further than the ball radius; almost everything sits near $\|x\|\approx 1/\sqrt{-K}$. This destroys dynamic range (all large features look the same) and makes gradients around the boundary extremely small.

2. **Hyperboloid: constraint is fragile without renorm.** On the Hyperboloid, points are supposed to satisfy $\langle x,x\rangle_L=-1/K$. Without explicit renormalisation, this constraint is badly violated, and even short scalar-mul chains already produce Inf. Any practical Hyperboloid implementation, therefore, has to add renormalisation/projection steps, which increases complexity and cost.

3. **PV: no geometric boundary, only a numeric safety cap.** PV does not show NaN/Inf or constraint violations in any of these tests. The fact that the median PV norm plateaus at a very large value is not a geometric boundary; it comes from a simple numerical safety rule where we clip the argument of $\sinh(\cdot)$ to stay within the safe floating‑point range. This clipping threshold is a tunable implementation detail and can be relaxed for higher precision. In our implementation we set a conservative cap $z_{max}$ = 30 on the argument of $sinh(⋅)$ in PV scalar multiplication, purely to avoid FP32 overflow; increasing or removing this cap immediately raises the observed median norm, whereas the Poincaré ball radius and Lorentz constraint are fixed by the geometry and cannot be relaxed in this way. In contrast, the Poincaré ball boundary and the Lorentz constraint are intrinsic to those geometries and cannot be “turned off” by tuning.

Taken together, Table A1 shows that PV avoids both types of numerical errors.

---

> ### Author Response · Authors · 2025-11-23
>
> Table A2 — Exp/Log Round-trip errors (median).
> | Model | **FP32** log–exp@v | **FP32** exp–log@y | **FP64** log–exp@v | **FP64** exp–log@y |
> |---|---:|---:|---:|---:|
> | **PV** | 2.853e−07 | 7.902e−05 | 1.055e−15 | 3.209e−13 |
> | **Poincaré** | 2.944e−02 | 8.769e−08 | 4.335e−11 | 1.354e−16 |
> | **Hyperboloid** | 2.029e+00 | 2.112e−04 | 2.045e+00 | 3.457e−13 |
>
>
>
> Table A2b — Exp/Log Round-trip errors (alternative “EXP–LOG@y RT” probe)
> | Model | **FP32** log–exp@v | **FP32** exp–log@y | **FP64** log–exp@v | **FP64** exp–log@y |
> |---|---:|---:|---:|---:|
> | **PV** | 1.794e−03 | 8.106e−03 | 5.281e−12 | 2.417e−11 |
> | **Poincaré** | NaN | NaN | 4.927e+00 | 6.794e−14 |
> | **Hyperboloid** | 5.213e+00 | 1.627e−03 | 5.295e+00 | 2.693e−12 |
>
> **These round-trip experiments test how well the Exp/Log maps behave as (numerical) inverses in practice:** we apply an Exp map followed by the corresponding Log map (or vice versa), e.g. $v \xrightarrow{\exp_0} y \xrightarrow{\log_0} v'$, and measure the reconstruction error $\|v' - v\|$ (or $\|y' - y\|$). Theoretically, on an exact manifold implementation, we would have $v' = v$ and $y' = y$, so the ideal round-trip error is zero; any non‑zero value directly reflects numerical errors and implementation instabilities of the Exp/Log operators.
>
> - For **PV**, the near–machine-precision FP64 errors and small FP32 errors show that $\exp_0$ and $\log_0$ remain numerically well-behaved across a wide range of radii. This is crucial for PVNN, which frequently transitions between the manifold and tangent space.
> - For **Poincaré**, the strong degradation of log(exp(v)) near the boundary (and NaNs in the more aggressive probe) reflects the sensitivity of the ball parametrisation: once points get close to the boundary, the logarithm becomes ill-conditioned, breaking the Exp–Log inversion.
> - For **Hyperboloid without renormalization**, the fact that log(exp(v)) has $O(1)$ errors even in FP64 means that the theoretical Exp/Log pair is *not numerically useful* unless one introduces careful re-projection and stabilization steps.
>
> Overall, Tables A2 and A2b show that PV offers stable, invertible Exp/Log maps in realistic floating-point regimes, while Poincaré and Hyperboloid suffer from boundary and constraint effects that can destroy invertibility in practice.
>
>
> ---
>
> Table A3 — Gradient magnitudes vs radius (FP32; K = -0.5, d = 16, batch = 4096).
> | Model | Approx. $\|\nabla\|$ range over $r \in [10^0,10^3]$ | Behavior |
> |---|---|---|
> | **PV** | $\approx 10^{-3}$–$10^{-2}$ | smooth, no vanish or explosion |
> | **Poincaré** | $\approx 10^{-11}$ | gradients essentially zero near the ball boundary |
> | **Hyperboloid** | from $0$ to NaN | gradients degenerate without projection/renorm |
>
> We perform a “gradient–vs–radius” sweep: we fix random directions in $\mathbb{R}^{16}$, scale them to radii $r$ on a log-grid from $10^0$ to $10^3$, apply scalar multiplication $x \mapsto r \otimes x$ on each manifold, define a simple loss as the mean output norm $\mathbb{E}\,\|op(x)\|$, and measure the average gradient norm $\|\nabla_x \mathbb{E}\,\|op(x)\|\|$ with respect to the input. A good operator should have gradients that are neither exploding nor vanishing across this radius range.
>
> PV maintains usable gradients; Poincaré vanishes near the boundary; Hyperboloid degenerates without renorm.
>
> **Analysis.**
> - For **PV**, gradients stay in a healthy range ($\sim 10^{-3}$–$10^{-2}$) over three orders of magnitude in radius, so optimization can move points far from the origin without losing learning signal or causing numerical explosions; this matches our observation that PVNN trains reliably even when norms become large.
> - For **Poincaré**, once most points are near $\|x\| \approx 1/\sqrt{-K}$, the gradient magnitude collapses to $\sim 10^{-11}$, i.e., effectively zero, making further refinement of the representation very difficult.
> - For the **Hyperboloid** model, gradients either vanish or become NaN, showing that the geometry is not numerically usable without an explicit projection/renorm step; any practical implementation must pay this extra cost.
>
> Thus, Table A3 indicates that PV offers a much more benign optimisation landscape for core operators, while classical hyperbolic manifolds (Poincaré, Hyperboloid) require additional tricks (clipping, renorm) to avoid pathological gradients.

---

> ### Author Response · Authors · 2025-11-23
>
> ---
>
> Table A4 — Runtime of basic geometric operators (FP32; K = -0.5; ms per call). Hyperboloid is slower, while renormalisation would further increase time.
> | Operation                          |   PV   | Poincaré | Lorentz |
> |------------------------------------|:------:|:--------:|:-------:|
> | add                                | 4.758  | 4.995    | 18.841  |
> | scalar (scalar-mul)                | 3.345  | 2.715    | 8.633   |
> | exp0                               | 1.193  | 1.023    | 6.820   |
> | log0                               | 1.755  | 1.754    | 2.802   |
> | point-dist (point distance)        | 5.703  | 6.038    | 1.528   |
> | hyperplane-dist (hyperplane score) | 6.813  | 7.194    | 9.286   |
> | project (ball projection)          |   —    | 0.874    |   —     |
> | renorm (Lorentz renormalization)   |   —    |   —      | 2.228   |
>
> _All times are measured in milliseconds (ms) per operator call on CPU_
>
> **Analysis.**
> These measurements show that PV achieves comparable per-operator cost to Poincaré for the core operations used in PVNN, and is consistently cheaper than Lorentz once one accounts for the renormalisation required to maintain the Lorentz constraint. In particular, PV’s exp0/log0 are in the same latency range as Poincaré, while PV’s hyperplane-distance is slightly faster, reflecting the closed-form PV-MLR scoring function. Lorentz, by contrast, incurs a significantly higher cost both for basic operators (e.g., add, exp0) and for the additional *renorm* step that is necessary for numerical stability. Overall, Table A4 supports our claim that PV offers numerical stability without incurring a prohibitive runtime penalty, remaining competitive with Poincaré and clearly more efficient than a practically usable Lorentz implementation.
>
> Based on these experiments, we modified Sec 6.1, expanding the original numerical experiments. See Tab. 1-4 for details.
>
>
>
>
>
> #  2. Advantages of PV.(Reviewers $\color{blue}{\text{QAye}}$ and $\color{green}{\text{36hh}}$)
>
> - **Poincaré requires $-K \lVert y \rVert^2<1$**. Many operators contain $(1+K \lVert y \rVert^2)^{-1}$: near the boundary, this causes gradient explosion/vanishing and hard saturation.
>
> - **Hyperboloid (Lorentz) is unbounded but constrained by $-x _t^2+\lVert x _s \rVert^2=1/K$**. In practice updates in \mathbb${R}^{n+1}$ must renormalize every step (solve $x _t=\sqrt{\lVert x _s \rVert^2-1/K}$). This induces (i) cancellation when $\lVert x _s \rVert\gg 1/\sqrt{-K}$, (ii) extra branches/projections, and (iii) sensitivity in back-prop through $x _t$.
>
> - **PV uses unconstrained $x\in\mathbb{R}^n$**. Core scalars are $\operatorname{asinh}(\alpha\lVert x \rVert)$ and $\sinh(\cdot)$ with benign derivatives:
>
> $\frac{d}{du}\operatorname{asinh}(u)=\frac{1}{\sqrt{1+u^2}}\le 1,
> \quad
> \Bigl\lVert \nabla_x\,\operatorname{asinh}(\alpha\lVert x \rVert)\Bigr\rVert\le \frac{\alpha}{\sqrt{1+\alpha^2\lVert x \rVert^2}}.$
>
> Thus PV Jacobians never contain $(1+K \lVert y \rVert ^2) ^{-1}$–type poles and require no per-step constraint repair.
>
> Therefore, unbounded ≠ Unconstrained PV’s unconstrained ℝⁿ updates avoid boundary poles and constraint renormalisation, yielding more stable numerics than Poincaré and Hyperboloid.
>
> And we do more numerical tests to prove the numerical stability, you can refer to ***CQs 1***
>
> # References
>
> [a] Fully Hyperbolic Neural Networks

---

### Meta-Review · Area_Chair_mtrj · 2025-12-11

**Summary:**

The current paper proposes to use Proper-Velocity coordinate model (in special relativity), whose Riemannian metric does not blow up on the boundaries leading to numerical stability, to build all components of a hyperbolic neural network (HNN). Most reviewers recognize the novelty of the contribution which yields numerical stable HNNs. The authors build a complete Riemannian toolkit (exp/log maps, gyro-structure, etc.) and test the method comprehensively with improved performance on multiple baselines. Reviewer 36hh's concern on the novelty is partially valid: PV is not a new manifold but different representation of the Poincare ball (up to isometry), and the HNN components largely rely on existing work. Overall, the contribution is meaningful as the PV representation with unique advantages has not been widely used in machine learning. The authors have clarified on the disagreement of their novelty in the rebuttal. I recommend acceptance.

**Reviewer Concerns:**

Reviewer 36hh is concerned on the lack of novelty. The authors have clarified their novelty lies in the complete toolkit, but not a new manifold.

The other concerns, e.g. graph-learning results, relation with Poincare models, have all been clarified in the rebuttal.

**Reviewer Scores:**

The original score is borderline. Reviewer tn82 could flip to the acceptance side, because the authors clarified the relationship between TV and classical models. Reviewer 36hh may not raise their score, but their concerns have been much clarified on what is new and what is not. It is more like whether the current novelty deserves acceptance, which is addressed in the rebuttal the above meta-review.

---

### Decision · Program_Chairs · 2026-01-26

Accept (Poster)